# Microbial peptides activate tumour-infiltrating lymphocytes in glioblastoma

Reza Naghavian[1,2], Wolfgang Faigle[1,2,3], Pietro Oldrati[1], Jian Wang[1,4], Nora C. Toussaint[5,6], Yuhan Qiu[1], Gioele Medici[7], Marcel Wacker[8,9,10,11], Lena K. Freudenmann[9,10,11], Pierre-Emmanuel Bonté[3], Michael Weller[12], Luca Regli[7], Sebastian Amigorena[3], Hans-Georg Rammensee[9,10,11], Juliane S. Walz[8,9,10,11,13], Silvio D. Brugger[14], Malte Mohme[15], Yingdong Zhao[16], Mireia Sospedra[1,2,20], Marian C. Neidert[7,17,20] & Roland Martin[1,2,18,19 ✉]

Microbial organisms have key roles in numerous physiological processes in the human body and have recently been shown to modify the response to immune checkpoint inhibitors[1,2]. Here we aim to address the role of microbial organisms and their potential role in immune reactivity against glioblastoma. We demonstrate that HLA molecules of both glioblastoma tissues and tumour cell lines present bacteria-specific peptides. This finding prompted us to examine whether tumour-infiltrating lymphocytes (TILs) recognize tumour-derived bacterial peptides. Bacterial peptides eluted from HLA class II molecules are recognized by TILs, albeit very weakly. Using an unbiased antigen discovery approach to probe the specificity of a TIL CD4+ T cell clone, we show that it recognizes a broad spectrum of peptides from pathogenic bacteria, commensal gut microbiota and also glioblastoma-related tumour antigens. These peptides were also strongly stimulatory for bulk TILs and peripheral blood memory cells, which then respond to tumour-derived target peptides. Our data hint at how bacterial pathogens and bacterial gut microbiota can be involved in specific immune recognition of tumour antigens. The unbiased identification of microbial target antigens for TILs holds promise for future personalized tumour vaccination approaches.

The treatment of cancer includes the three classical 'pillars': surgery, radiotherapy and chemotherapy. With cancer immunotherapy, a fourth pillar emerged[3]. The introduction of immune checkpoint inhibition led to advances in survival rates in several cancers and demonstrated that unleashing immune effector mechanisms can result in the elimination of tumour cells[4]. In parallel, numerous approaches are being explored to go a step further and vaccinate patients with tumour neoantigens to induce tumour-specific T cells that mount an effective immune response against the tumour[5]. Several lines of evidence indicate that tumour vaccination will be feasible, particularly in 'hot' tumours with high numbers of mutations and strong immune cell infiltrates[6,7]. However, numerous hurdles still need to be overcome including the relatively low immunogenicity of many tumour antigens, which, if not mutated, do not elicit strong antitumour responses, as T cells with reactivity against self-antigens are eliminated by thymic central tolerance[8]. Hence, the more foreign an antigen 'looks' to T cells, the more likely it will induce a strong immune response. In agreement with this, reports about patients, whose tumour either shrank substantially during an infection with a bacterial[9] or viral pathogen[10] or disappeared, indicated that protective immune responses against pathogens may also target the tumour, most likely by T cell cross-reactivity against tumour-derived and pathogen-derived antigens[11]. Supporting this notion, several studies have demonstrated improved responses to immune checkpoint inhibitors in the presence of certain gut bacteria[1,2]. Adoptive transfer

[1]Neuroimmunology and MS Research Section (NIMS), Neurology Clinic, University of Zurich, University Hospital Zurich, Zurich, Switzerland. [2]Cellerys AG, Schlieren, Switzerland. [3]Immunity and Cancer, Institut Curie, PSL University, INSERM U932, Paris, France. [4]School of Basic Medical Sciences, Division of Life Sciences and Medicine, University of Science and Technology of China, Hefei, China. [5]NEXUS Personalized Health Technologies, ETH Zurich, Schlieren, Switzerland. [6]Swiss Institute of Bioinformatics, Zurich, Switzerland. [7]Clinical Neuroscience Center, Department of Neurosurgery, University Hospital Zurich, University of Zurich, Zurich, Switzerland. [8]Department of Peptide-based Immunotherapy, University of Tübingen, University Hospital Tübingen, Tübingen, Germany. [9]Institute for Cell Biology, Department of Immunology, University of Tübingen, Tübingen, Germany. [10]German Cancer Consortium (DKTK) and German Cancer Research Center (DKFZ), partner site Tübingen, Tübingen, Germany. [11]Cluster of Excellence iFIT (EXC 2180) 'Image-Guided and Functionally Instructed Tumor Therapies', University of Tübingen, Tübingen, Germany. [12]Laboratory of Molecular Neuro-Oncology, Department of Neurology and Clinical Neuroscience, University Hospital Zurich, University of Zurich, Zurich, Switzerland. [13]Clinical Collaboration Unit Translational Immunology, German Cancer Consortium (DKTK), Department of Internal Medicine, University Hospital Tübingen, Tübingen, Germany. [14]Department of Infectious Diseases and Hospital Epidemiology, University Hospital Zurich, University of Zurich, Zurich, Switzerland. [15]Department of Neurosurgery, University Hospital Hamburg Eppendorf, University of Hamburg, Hamburg, Germany. [16]Computational and Systems Biology Branch, Biometric Research Program, Division of Cancer Treatment and Diagnosis, NCI, NIH, Rockville, MD, USA. [17]Department of Neurosurgery, Cantonal Hospital St. Gallen, St. Gallen, Switzerland. [18]Institute of Experimental Immunology, University of Zurich, Zurich, Switzerland. [19]Therapeutic Immune Design Unit, Center for Molecular Medicine, Department of Clinical Neurosciences, Karolinska Institutet, Stockholm, Sweden. [20]These authors contributed equally: Mireia Sospedra, Marian C. Neidert. ✉e-mail: roland.martin@uzh.ch

of *Bacteroides fragilis*-specific T cells to germ-free mice restored the response to anti-CTLA4 (ref. 2). Indeed, antitumour responses can be elicited by cross-reactivity between commensal bacteria and tumour antigens[12]. Furthermore, CD8[+] T cells can cross-recognize a peptide from *Enterococcus hirae* bacteriophage and a tumour antigen by molecular mimicry[13]. One approved therapy for bladder cancer, an extract of the *Mycobacterium bovis* strain Bacille Calmette Guerin, which is locally instilled into the bladder, has been shown to induce T helper 1 ($T_H1$) CD4[+] T cell responses against the tumour and to provide long-term protection in mice[14,15]. The importance of CD4[+] T cells in driving antitumour responses are highlighted by showing that melanoma-derived neoepitopes are recognized by CD4[+] T cells[16], and neoantigen vaccinations in melanoma and glioblastoma primarily activate CD4[+] T cells[17–19].

As another line of evidence hinting at a possible involvement of pathogens or commensal bacteria in antitumour responses, several tumours harbour bacteria[20]. Melanoma metastases even present bacterial peptides on their HLA class I and class II molecules[21], indicating that these are available for T cell recognition and thus could be used for tumour vaccination. Intratumoural bacteria may influence tumour outcome positively and negatively in several ways, including direct effects on transcriptional pathways of inflammation, tumour growth and immune cell infiltration[22,23]. Regarding antitumour immunity, microbiota participate in shaping the adaptive immune repertoire by stimulation via pathogen-associated molecular patterns, by immune-modulating metabolites and, most interestingly in the context of tumour vaccination, by inducing cross-reactive T cell responses[11].

Here we pursued several experimental paths to address whether and how microbial antigens may contribute to immune reactivity against glioblastoma. We first examined the glioblastoma immunopeptidomes of 19 patients for HLA class II-bound bacterial peptides, and next whether TILs and TIL-derived CD4[+] T cell clones (TCCs) in one of the patients react against HLA-bound bacterial peptides and tumour antigens. We then used an unbiased antigen discovery approach to identify targets derived from viruses, bacteria and bacterial gut microbiota, which we refer to as gut microbiota from now on. TCC88, which is directed to a glioblastoma neoantigen, not only recognizes multiple other glioblastoma-derived peptides but also responds strongly to a broad spectrum of bacterial-derived and gut microbiota-derived targets, which also elicit cross-reactive T cell responses against tumour targets in bulk TILs and even peripheral blood memory T cells.

## Microbial peptides in HLA class II of glioblastoma

A study has reported that tumours contain distinct microbial communities including glioblastoma[20]. Furthermore and supporting that bacterial antigens within tumours could in principle elicit intratumoural immune responses, another study has found bacterial peptides in the HLA class II immunopeptidome of melanoma cells[21]. Although it appears easier that bacterial peptides can be presented by tumour cells or local antigen-presenting cells (APCs) in the skin, we examined this topic here in glioblastoma. We first analysed the HLA class I-associated and HLA class II-associated immunopeptidomes of 19 primary or recurrent glioblastomas (Supplementary Tables 1 and 2) and annotated these for human proteins, tumour-specific antigens (that is, mutated tumour antigens) and also for sequences from bacteria (UniProtKB/Swiss-Prot protein database) and viruses with tropism for humans (Swiss-Prot-20210305) (Fig. 1a). We identified not only multiple tumour-associated antigens, that is, self-proteins overexpressed in the tumour, and tumour-specific antigens on both HLA class I and class II molecules (data not shown) but also between 5 and 54 unique HLA class II-bound bacterial sequences per patient and tumour, but no viral peptides (Fig. 1b, left, and Supplementary Table 3). As controls, we analysed the immunopeptidomes from brain lesions of patients with multiple sclerosis (MS) and healthy brain tissue[24] (dataset identifier PXD019643). We found between 10 and 45 bacterial hits from brain tissues of three

patients with MS and six healthy donors (Extended Data Fig. 1a and Supplementary Table 4). Seven peptides from the control cohorts exactly matched peptides in glioblastoma samples, and we therefore subtracted these from the bacterial peptides in glioblastoma. Tumour-derived bacterial peptides originated from 255 bacterial proteins and 199 unique bacteria mainly from the phyla Proteobacteria and Firmicutes, with Enterobacteriaceae, Bacillaceae and Mycobacteriaceae among the top families from each phylum (Fig. 1b, right, and Supplementary Table 3). The composition and number of bacterial peptides differed between patients, although several shared HLA alleles (Supplementary Table 2). Furthermore, we generated tumour cell lines from six of the patients and analysed the immunopeptidomes (Supplementary Table 1). The tumour cell lines presented similar numbers of HLA class II-bound bacterial peptides, which we refer to here as immunopeptidome-derived bacterial peptides (IPdBPs). These observations indicate that bacterial peptides exist in the tumour cells, are processed and loaded onto HLA molecules and form part of the IPdBPs. However, the number of IPdBPs of both tumour tissue and cell lines was low (Supplementary Table 3). In addition, both tumour and isolated cell lines showed normal expression of HLA class II in patient 1635WI (Extended Data Fig. 2). We can therefore not rule out that at least part of the IPdBPs are presented by APCs in the tumour. We further conducted 16S rRNA gene sequencing on fresh frozen tumour tissues of ten patients with glioblastoma. Similar to immunopeptidome analyses, bacterial loads in the tumours were low and Proteobacteria accounted for the majority of them (Extended Data Fig. 1b and Supplementary Tables 5 and 6). We then compared the 16S rRNA results with bacterial sources of the IPdBPs in each patient. We found at least one exact match for bacterial species in five out of ten patients (Supplementary Table 7). As the resolution at the species level is limited due to high similarities of 16S rRNA sequences between certain bacteria[21], we examined the data at the next taxonomic level, bacterial genus, and found matches in seven patients. These data suggest the presence of both bacterial DNA as well as peptides in glioblastoma. Next, we assessed the possible immune recognition of IPdBPs in detail in patient 1635WI, whom we had previously treated with a personalized neoantigen vaccination and checkpoint inhibitors in addition to standard therapy[25]. We identified 18 and 23 unique IPdBPs from primary and recurrent tumour, respectively (Fig. 1c, top, and Supplementary Tables 8 and 9), and four in both tumours. Peptides of the primary and recurrent tumours were derived from 15 and 21 bacterial proteins (Supplementary Table 8). Consistent with the immunopeptidome data of the entire group of patients, Proteobacteria and Firmicutes were most common (Fig. 1c, bottom). Furthermore, we compared the deduced core binding motifs and HLA-DR anchor positions between bacterial and self-antigen or tumour-antigen-derived peptides within this patient (Extended Data Fig. 1c). Amino acid preferences at HLA-DR anchor positions were similar between bacterial and self/tumour peptides, but different at non-anchor positions (Extended Data Fig. 1c).

We further wanted to know whether IPdBPs are involved in the intratumoural immune response and therefore tested bulk TILs from the recurrent glioblastoma (patient 1635WI) with all 37 unique IPdBPs. Compared with a tumour neoantigen derived from the paired amphipathic helix protein SIN3A, peptide SINA3A* 300–312 (QPVEFNHAIHYVN; '*' marks all mutated proteins throughout the paper), which was part of the tumour vaccine, the proliferative response of TILs to the 37 IPdBPs was very weak (Fig. 1d and Supplementary Table 10); however, they secreted pro-inflammatory cytokines including IFNγ, TNF and GM-CSF upon stimulation with several IPdBPs (Fig. 1e).

## Tumour-reactive CD4[+] TCCs

The above data indicate that bacteria-derived peptides may be involved in intratumoural immune responses. As bulk TIL responses to IPdBPs were weak, we isolated TCCs from the TILs that recognize a tumour antigen with the goal of testing them for possible recognition of IPdBPs.

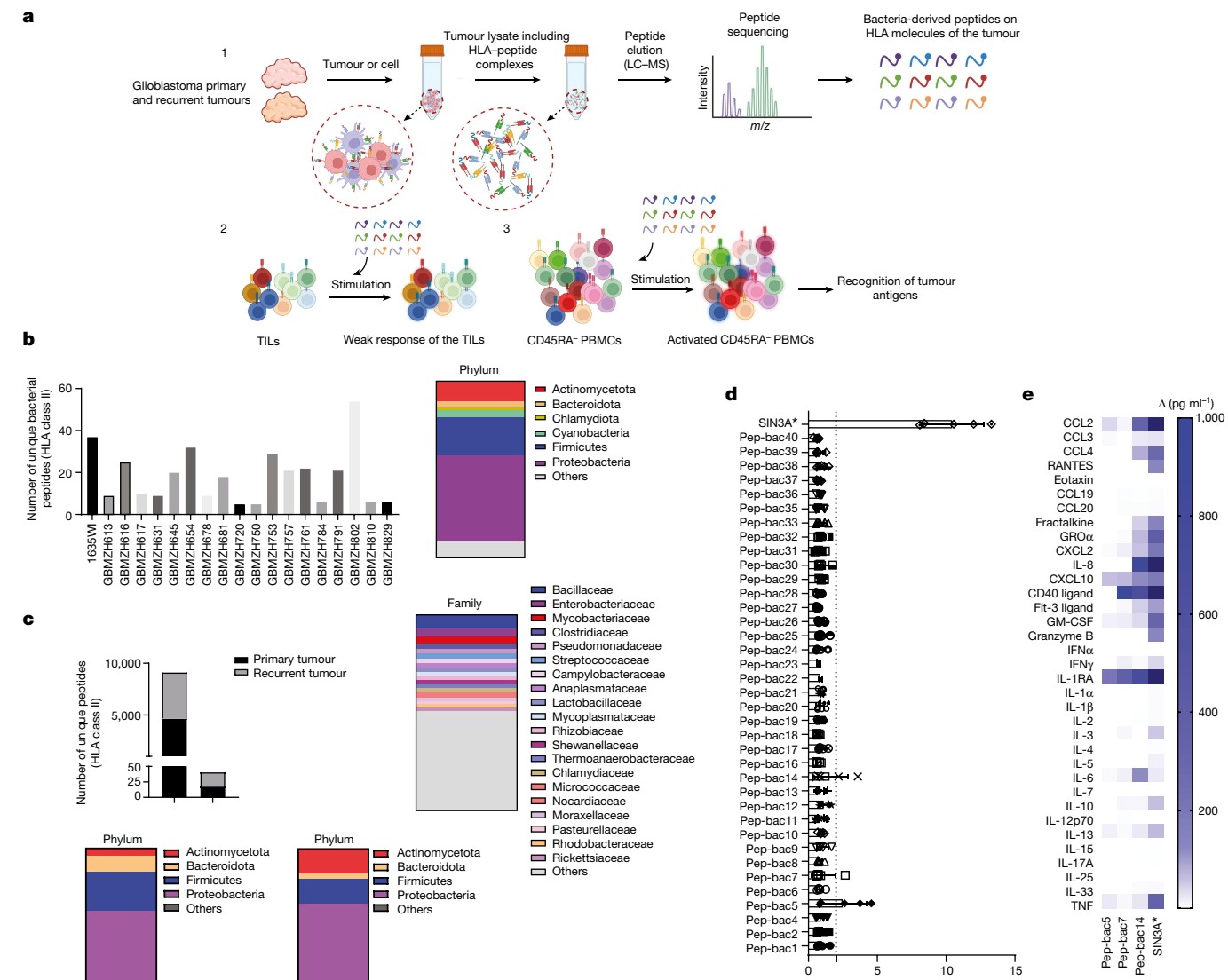

**Fig. 1 | Elution of HLA class II-presented bacterial peptides from 19 glioblastomas and the stimulatory effect of IPdBPs on bulk TILs. a**, Schematic of the immunopeptidome analyses of glioblastoma tissue, from which eluted peptides were annotated using the UniProt bacteria database (1). TILs or peripheral memory T cells were stimulated with IPdBPs (2 and 3). LC–MS, liquid chromatography–mass spectrometry; PBMC, peripheral blood mononuclear cell. The images in panel **a** were created using BioRender (https://biorender. com). **b**, Number of unique IPdBPs in patients with glioblastoma (*n* = 19) (data from patient 1635WI contains both primary and recurrent tumours together), with 344 unique IPdBPs identified (left). These peptides originated from 255 and 199 unique proteins and bacteria, respectively (3 technical replicates). The taxonomy of these bacteria shows that they primarily stem from Proteobacteria and Firmicutes at the phylum level (right). **c**, IPdBPs from primary and recurrent tumours (patient 1635WI) (top). As a control, peptidome data were also annotated using human protein database to identify eluted self-peptides. As shown, most

of HLA class II-bound peptides are self-(human) peptides. IPdBPs of primary (bottom left, *n* = 18) and recurrent (bottom right, *n* = 23) tumours were separately annotated to their source bacteria at the phylum level. **d**, Bulk TILs (patient 1635WI) were stimulated with IPdBPs (See Supplementary Table 8), each in five replicate wells, and proliferation was measured after 5 days via [3]H-thymidine incorporation. Proliferation is shown as a stimulatory index (SI), and peptides with SI ≥ 2 are considered stimulatory (the dotted line shows SI = 2 throughout the paper). The data are expressed as the mean value of all wells ± s.e.m. SIN3A* peptide was used as a positive control as TILs recognize this tumour antigen. T cells with APCs without any stimuli were used as a negative control. **e**, Heatmap of cytokine and chemokine secretion of TILs stimulated with three IPdBPs, which weakly stimulated proliferation. The secretion of each cytokine and chemokine was calculated by subtracting the corresponding cytokine and chemokine amount (pg ml[−1]) of the negative control. TILs with APCs without any stimuli were used as a negative control. Also see Extended Data Fig. 1.

On the basis of the response of bulk TILs to a strongly stimulatory peptide from the personalized vaccine cocktail[25], we used a SIN3A* peptide to stimulate bulk TILs (Fig. 2a and Extended Data Fig. 3a). TCCs were then generated from proliferating CD4[+] (carboxyfluorescein succinimidyl ester (CFSE)[dim]) T cells (Fig. 2a). Of CD4[+] TCCs, 70 out of 124 responded to the SIN3A* peptide by proliferation or production of IFNγ (Fig. 2b). TCCs were characterized further for TCR expression and functional phenotypes. Most showed a polyfunctional $T_H1/T_H2$ phenotype with production of IFNγ, IL-5 and IL-13 after stimulation via

anti-CD2, anti-CD3 and anti-CD28 antibodies (Fig. 2c). Next, we tested several of these tumour-reactive TCCs with IPdBPs. TCC57 and TCC2D3 responded to one and several IPdBPs; however, again relatively weakly compared with the SIN3A* peptide (Fig. 2d).

## TCCs cross-recognize bacterial peptides

We next investigated the recognition of foreign antigens by intratumoural lymphocytes from a different angle, that is, not starting from

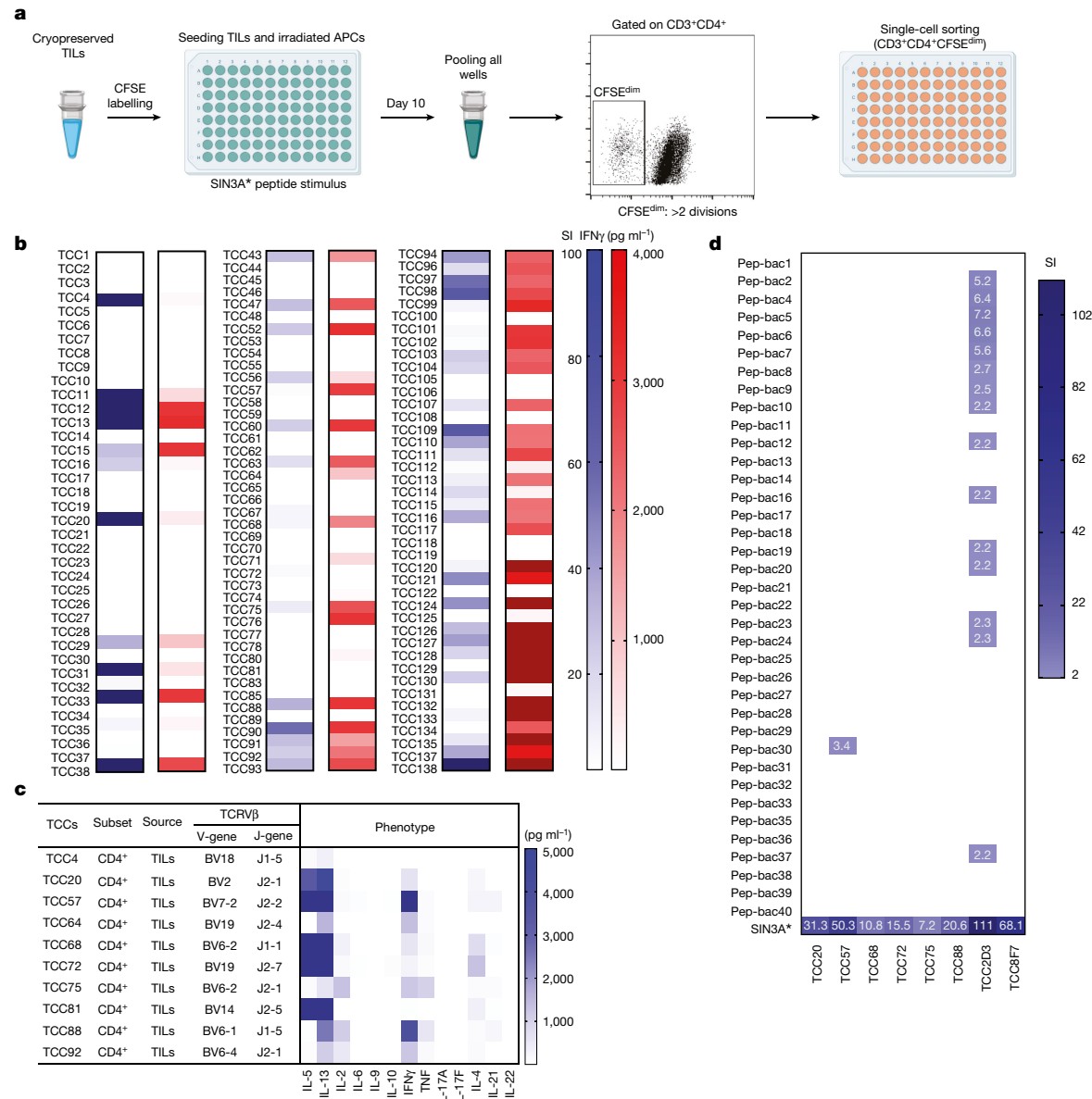

**Fig. 2 | Isolation and characterization of tumour-specific TCCs by stimulating TILs with SIN3A* peptide. a**, Bulk TILs were labelled with CFSE and stimulated with irradiated autologous PBMCs as APCs pulsed with SIN3A* peptide. After 10 days, 60 replicate wells were pooled and CFSE^dimCD3+CD4+ T cells were single-cell sorted and expanded using PHA and human IL-2. The images in panel **a** were created using BioRender (https://biorender.com). **b**, All isolated TCCs were tested against the cognate antigen (SIN3A* peptide) for proliferation and IFNγ production. After 3 days, proliferation was measured using 3H-thymidine incorporation (SIs), and ELISA was used to measure IFNγ

secretion. The response of each TCC was compared with the stimulation of the same TCC with APCs without peptide. **c**, TCRB V-gene and J-gene and functional phenotypes of several of the TCCs. **d**, Several TIL-derived TCCs with reactivity to tumour antigen (SIN3A*) were tested for reactivity against IPdBPs. TCCs were stimulated with peptides and irradiated PBMCs as APCs for 3 days. Proliferation was measured using 3H-thymidine incorporation. SIN3A* peptide served as the positive control. Mean SIs are shown in a heatmap format. SI < 2 (non-stimulatory) are shown with an empty field, and only SI ≥ 2 for stimulatory peptides are shown. Also see Extended Data Fig. 3.

the immunopeptidome, but with an unbiased antigen discovery methodology (Extended Data Fig. 3b), which we had previously used to identify the targets for autoreactive TCCs from the peripheral blood, cerebrospinal fluid and brain tissue of patients with MS[26,27], and in viral and bacterial infections[28,29]. It involves testing TCCs with synthetic combinatorial peptide libraries in the positional scanning format (ps-SCL). Individual ps-SCL mixtures are characterized by one defined amino acid in a determined position of a decamer, for example, L-alanine in position 1, whereas the remaining nine positions are randomized (Extended Data Fig. 3b). As a preparatory step[30], we characterized the HLA-DR restriction of several TCCs using irradiated Epstein–Barr virus (EBV)-transformed bare lymphocyte syndrome (BLS) B cells transfected

with single autologous HLA-DR molecules including DRB1*03:01, DRB1*04:02, DRB3*02:02 or DRB4*01:01. Most CD4+ TCCs robustly recognized the SIN3A* peptide when presented by HLA-DRB3*02:02 (Extended Data Fig. 4a). As previously shown for virus-specific and autoantigen-specific TCCs[31,32], TCC88 showed cross-restriction by responding to the cognate peptide presented on all four HLA-DR molecules but responded strongest to SIN3A*–HLA-DRB3*02:02 (half-maximal effective concentration (EC50) = 0.013 µM) (Extended Data Fig. 4a). After testing a TCC with the 200 ps-SCLs, the experimental data (proliferation or cytokine release) were transformed into a scoring matrix that depicts the stimulatory score for each L-amino acid in each position of a decamer peptide[29] (Extended Data Fig. 3b). On the

basis of the assumption that each amino acid contributes independently and additively to T cell recognition, the scoring matrix and a dedicated search algorithm allow scanning of any database to identify target peptides for the respective TCC in an unbiased way[29]. Peptides with predicted high stimulatory scores, for example, with higher than 80% of the maximal score, are then selected, synthesized and tested. Previously, for virus-specific, bacteria-specific and autoreactive TCCs, we had demonstrated that there is a strong relationship between the predicted stimulatory score and T cell recognition[29,33]. Among the above SIN3A*-specific TCCs, TCC88 responded well to the ps-SCL when these were presented by BLS cells transduced with HLA-DRA1*01:01 and HLA-DRB3*02:02 using IFNγ production as T cell activation readout (Extended Data Fig. 4b). Most of the other TCCs did not respond to ps-SCL (data not shown).

We next created a scoring matrix[29] using the median IFNγ values of three independent experiments and transforming these to $\log_{10}$ values (Extended Data Fig. 4c). Following the in silico predictions, over 300 peptides in the range between 60% and 100% of the maximum theoretical score (%) were synthesized and tested to assess and reconfirm the accuracy of the predictions. Of these, TCC88 recognized more than 80% of peptides within the score range of 90–100% of the maximum score (Extended Data Fig. 4d).

For the identification of putative targets of TCC88, we searched several large databases including all human proteins, the translated transcriptome of the primary and recurrent autologous glioblastoma, all viruses, which are annotated for infecting humans, as well as UniProt-deposited bacteria and human gut microbiota[34] (see Methods) (Fig. 3a). The numbers of unique 10-mers within each of these databases are shown in Fig. 3a. Combined, these represent more than $5 \times 10^9$ decamer peptides. We synthesized 542 peptides (see Methods) with high predicted stimulatory scores (Fig. 3a) and tested them with TCC88. As previously shown for other TCCs[26–29,32], TCC88 recognized multiple peptides from different sources including the autologous tumour transcriptome, the human proteome, viruses, bacteria and gut microbiota (Fig. 3a and Supplementary Table 10). Interestingly, 57.7% of all stimulatory peptides were derived from bacteria and gut microbiota, and 29.3% from tumour antigens containing tumour-associated antigens derived from the autologous glioblastoma, whereas peptides derived from other human proteins and viruses represented only minor fractions (Fig. 3a). Following our above observation, we were particularly interested in the bacteria-derived and microbiota-derived peptides that stimulated TCC88. We tested 126 bacteria-derived and 100 gut microbiota-derived peptides. The TIL-derived, SIN3A*-specific TCC88 recognized 21 bacterial and 42 gut microbiota-derived peptides (Fig. 3a,b) and 34 of these equally well or even better at a single-antigen concentration than the mutated tumour peptide, whereas peptides from other sources did not show this effect (Fig. 3b,c). The recognition with high functional avidity by TCC88 was confirmed by dose titration of several bacteria-derived and microbiota-derived peptides in the context of HLA-DRB3*02:02 ($EC_{50}$ ~ 0.017, 0.018 and 0.019 μM for the peptides HGM62 (*Sedimentibacter*), HGM3 (*Victivallis*) and HGM27 (*Prevotella*), respectively), which is similar to SIN3A* ($EC_{50}$ ~ 0.018 μM) (Fig. 3d). Despite the remarkable cross-recognition of multiple tumour-derived, bacteria-derived and microbiota-derived peptides, TCC88 did not respond to a large number of peptides (260 peptides) covering the whole-protein sequence of several brain and myelin, pancreatic and skin autoantigens (Extended Data Fig. 4e and Supplementary Table 11). The response of TCC88 to the unmutated SIN3A peptide confirmed that it weakly recognizes the wild-type antigen (Extended Data Fig. 4f). Upon activation with cognate antigen or bacteria-derived and microbiota-derived peptides, TCC88 shows a pro-inflammatory phenotype and secretes IFNγ, GM-CSF, TNF and granzyme B (Fig. 3e). The release of CCL3 (also known as MIP1α) and CCL4 (also known as MIP1β) indicates that activated TCC88 may recruit other immune cells such as CD8+ T cells[35] (Fig. 3e). Although the cytokine and chemokine

responses of TCC88 to different antigens were overall comparable, differences were seen, that is, higher IL-2 and fractalkine (also known as CX3CL1) levels upon stimulation with SIN3A* and increased levels of IL-10 and CXCL10 (also known as IP10) with bacteria-derived and microbiota-derived peptides (Fig. 3e). Furthermore, cytotoxic activity against autologous glioblastoma cells was comparable after stimulation with SIN3A* or HGM3 peptides (Fig. 3f). Owing to slow in vitro growth of patient-derived tumour cells, only HGM3 peptide was tested as it has the same $EC_{50}$ as the SIN3A* peptide for the clone. Hence, TCC88 strongly cross-recognizes tumour antigens as well as bacteria-derived and microbiota-derived peptides, which are mainly derived from the phyla Firmicutes, Proteobacteria and Bacteroidota (Extended Data Fig. 5a). In line with previous data and other clones[26,32], we were not able to discern common amino acid patterns in stimulatory peptides when compared with non-stimulatory peptides (Extended Data Fig. 5b). Among stimulatory peptides are peptides from several species including *B. fragilis* (HGM32 peptide), which had previously been shown to be important for immune checkpoint inhibitor therapies[2] (Supplementary Table 10). Finally, stimulating TCC88 with both HGM3 and SIN3A* peptides did not have synergistic effects (Fig. 3g).

## TILs recognize bacterial peptides

The above observation raised the question whether the reactivity of TCC88 against tumour antigens is unique for this clone and whether bacteria-derived and microbiota-derived peptides could also be recognized by bulk TILs. We therefore tested the proliferative and cytokine responses of bulk TILs against the bacteria-derived and microbiota-derived peptides that had been identified as targets for TCC88. Indeed, bulk TILs also recognized 23 bacteria-derived and microbiota-derived peptides (Fig. 4a and Extended Data Fig. 5c). To exclude that the response of bulk TILs is due to the fact that they contain TCC88, we tested additional TIL-derived TCCs that had been established by stimulation with SIN3A* peptide against the bacteria-derived and microbiota-derived peptides. TCC68 and TCC75 also recognized both SIN3A* peptide and various bacteria-derived and microbiota-derived peptides (Extended Data Fig. 5d). However, not all SIN3A* peptide-responsive TCCs showed this pattern. As an example, TCC8F7 strongly responds to the SIN3A*-derived and several other glioblastoma-derived peptides, but not any of the bacterial antigens (Extended Data Fig. 5d). Figure 4b summarizes the responses of TILs and TCCs to bacteria-derived and microbiota-derived peptides. Their reactivity against the peptides that had been selected based on the unbiased antigen search for TCC88 differed, but both TCC68 and TCC75 recognized several bacteria-derived and microbiota-derived peptides with different strengths (Fig. 4b). The comparison of the reactivity of TCC88, bulk TILs and two other TCCs shows that TILs contain multiple TCCs with different reactivity patterns but clear responses to the bacteria-derived and microbiota-derived peptides. To corroborate this point, we compared the TCR expression of the TCCs. TCC88 expresses TCR beta variable 6.1 (TCRBV6.1), and TCC68 and TCC75 express TCRBV6.2. We further stained TILs using a panel of 22 fluorescently labelled TCRBV antibodies that can detect 27 different TCRBVs (Supplementary Table 12), excluding TCRBV6.1 and TCRBV6.2. Stained CD4+ TILs were then sorted and rested for a day before testing them with peptides (Extended Data Fig. 6a). The antibody panel stained approximately half of the TCRVβs in the TILs and not only excluded TCRBV6.1 and TCRBV6.2 but also those for which no antibody was available. These depleted TILs still responded to bacterial peptides (shown with red arrows in Extended Data Fig. 6b), supporting that bulk TILs contain several T cells that recognize bacteria-derived and microbiota-derived peptides. Moreover, the cytokine and chemokine profile of TCC75 shows that it also secretes pro-inflammatory cytokines upon activation by either a tumour neoantigen or *Prevotella* spp. (HGM27 peptide). Of note, TCC75 secretes higher levels of IL-10 upon stimulation with

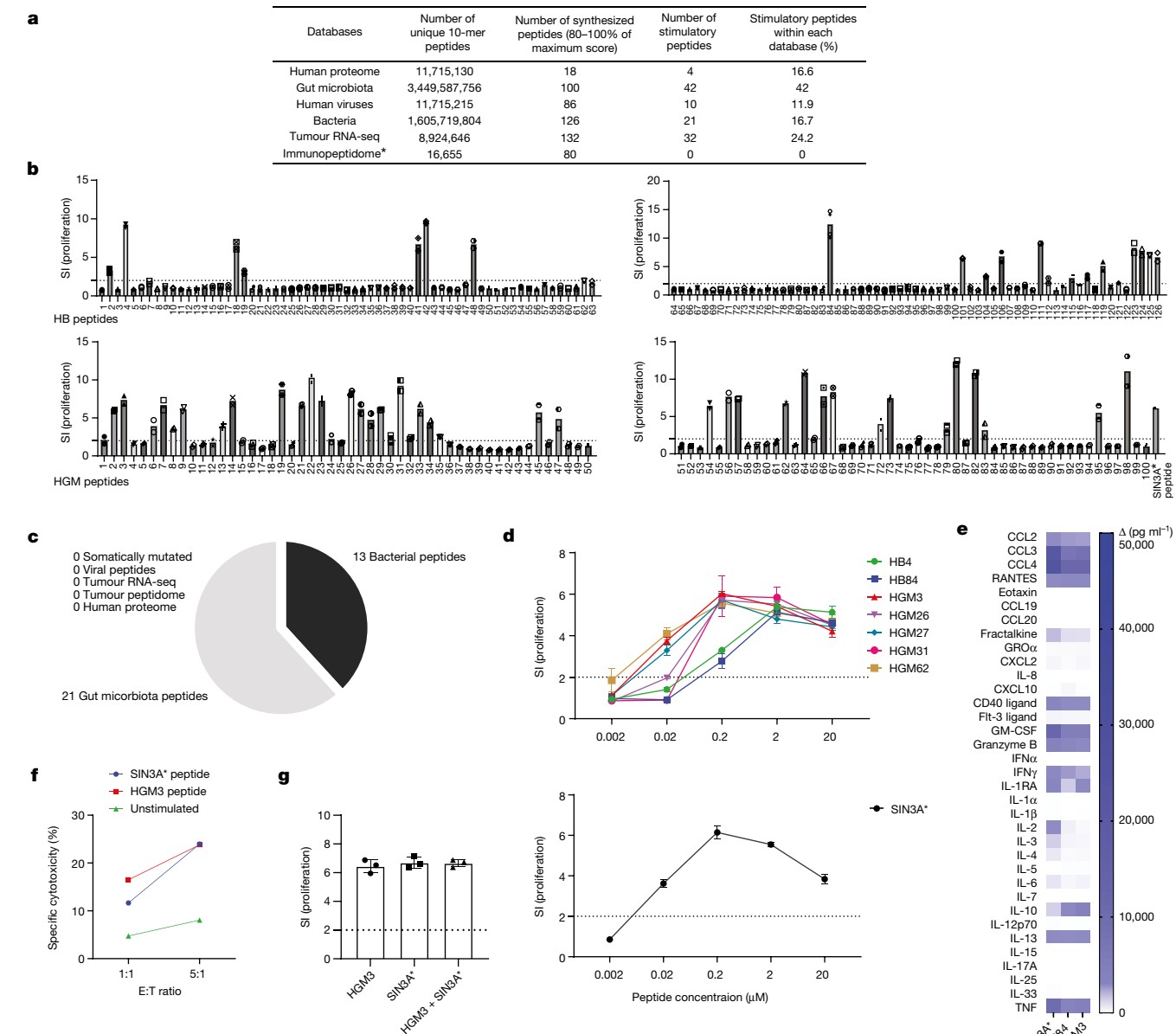

**Fig. 3 | Bacteria-derived and microbiota-derived peptides can stimulate TCC88 as well as the tumour antigen. a**, Peptides with the highest predicted stimulatory scores were identified by searching various databases (see Methods). Numbers of unique 10-mer peptides within each database, synthesized peptides, TCC88 stimulatory peptides and the corresponding percentage of stimulatory peptides within each database are shown. Proliferation was measured using $^3$H-thymidine incorporation (see Supplementary Table 10). *The immunopeptidome is annotated using human proteins in which 11 peptides score over 80%. Hence, the rest of the synthesized peptides were chosen from 60–80% of maximum score. RNA-seq, RNA sequencing. **b**, TCC88 was stimulated with bacteria-derived (top) and microbiota-derived (bottom) peptides for 3 days and proliferation was measured by $^3$H-thymidine incorporation (data represent the mean SI value of two replicate wells). HB, UniProt bacteria database; HGM, human gut microbiota[34]. **c**, Pie chart displaying the number of peptides from all databases that stimulated TCC88 equally well or better than the cognate antigen (SIN3A* tumour antigen) at a concentration of 10 μM.

**d**, TCC88 was tested with several strongly stimulatory bacteria-derived and microbiota-derived and SIN3A* peptides at decreasing concentrations including 20, 2, 0.2, 0.02 and 0.002 μM. Irradiated BLS-DRB3*02:02 was pulsed with the peptides and used to stimulate TCC88 (data represent the mean SI value of five replicate wells ± s.e.m.). **e**, Supernatants were collected from TCC88 after stimulation with SIN3A*, HGM3 and HB84 peptides. Cytokine and chemokine secretion of TCC88 was examined and compared upon stimulation with these three different antigens. A specific amount of each cytokine and chemokine (pg ml$^{-1}$) was calculated by subtracting the cytokines in the negative control cultures (no peptide control). **f**, Cytotoxicity of TCC88 against the autologous tumour cell line after stimulation with 2 μM of SIN3A* and HGM3 (data represent the mean percentage of two replicate wells). E:T, effector to target. **g**, TCC88 was tested with HGM3 (10 μM) or SIN3A* (10 μM) peptides or both together (each 10 μM), and $^3$H-thymidine incorporation was measured after 3 days (data represent the mean SI value of three replicate wells ± s.e.m.). Also see Extended Data Figs. 3 and 4.

tumour antigen, showing that different TCCs may secrete different cytokines and chemokines upon stimulation with a similar antigen (Extended Data Fig. 5e). Next, we compared the bacteria-derived and microbiota-derived peptides that we identified for TIL-derived TCC88

with the IPdBPs of the tumour, but did not find shared sequences. However, some peptides derived from the same source bacterium including *Bacillus* sp. and *Bacteroides ovatus*, indicating that TCC88 responded to peptides from different proteins of these bacteria (Supplementary

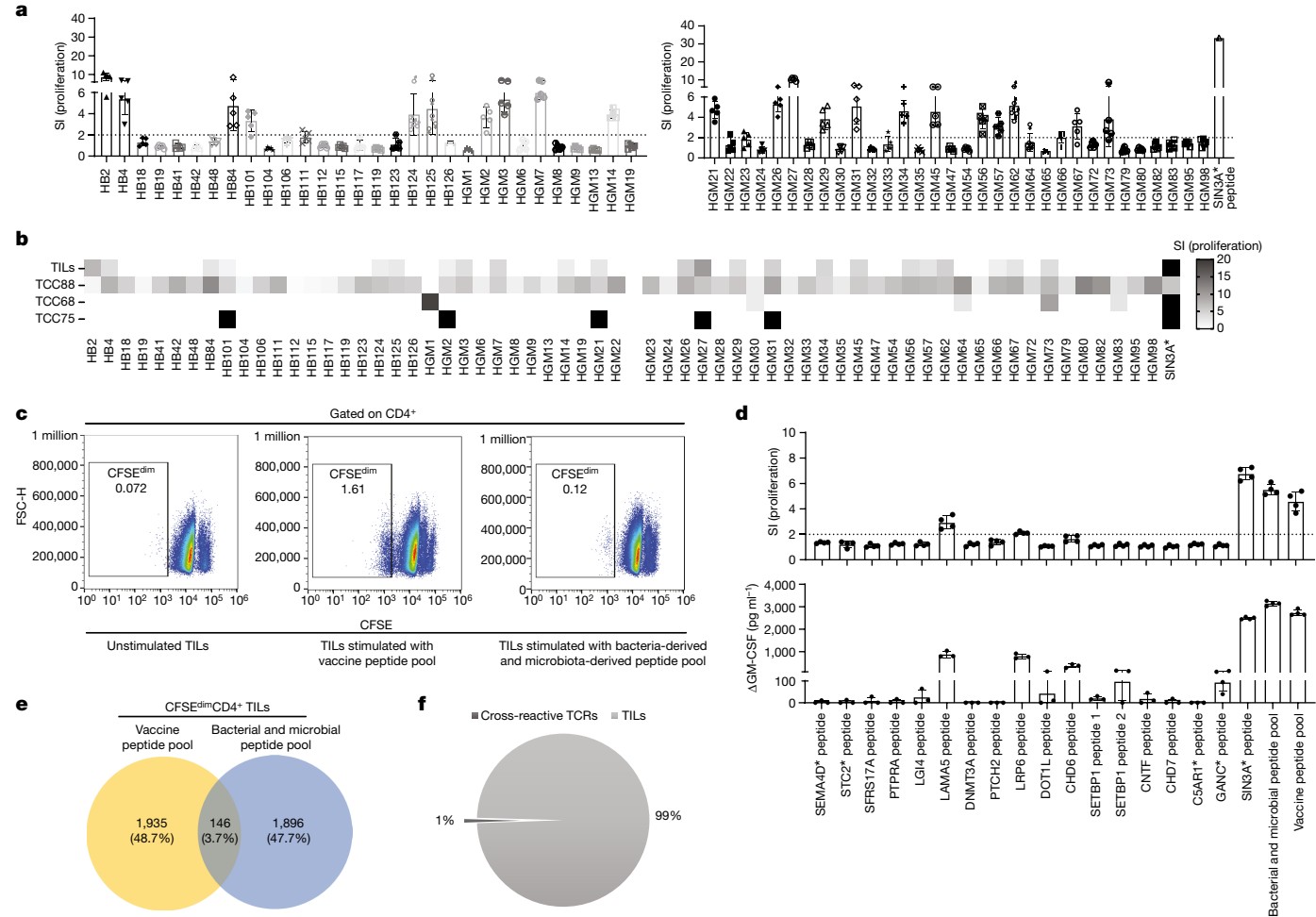

**Fig. 4 | Response of bulk TILs to bacteria-derived and microbiota-derived peptides. a**, Bulk TILs were stimulated with bacteria-derived and microbiota-derived peptides and irradiated autologous PBMCs as APCs. Proliferation was measured by [3]H-thymidine incorporation after 5 days (data represent the mean value of five replicate wells ± s.e.m.). **b**, Overview of proliferative responses of several TCCs and bulk TILs to bacteria-derived and microbiota-derived peptides shown in a heatmap. SI < 2 are shown in white and SI ≥ 2 are shown in varying shades of grey. **c**, Bulk TILs were labelled with CFSE and separately stimulated with a pool of bacteria-derived and microbiota-derived peptides and a pool of vaccine or tumour peptides that includes peptides from somatic mutations and genes that were overexpressed in the autologous tumour (see Supplementary Table 13). Peptide-pulsed, irradiated autologous PBMCs were used as APCs, and bulk TILs were incubated for 10 days. Sixty replicate wells were pooled and proliferating CD4[+] T cells (CFSE[dim]) were sorted. FSC-H, forward scatter height. **d**, Expanded CFSE[dim]CD4[+] TILs that responded to the bacteria-derived and microbiota-derived peptide pool were stimulated with several tumour antigens, a bacteria-derived and microbiota-derived peptide pool and a pool of vaccine peptides. After 5 days, supernatants were harvested and proliferation was measured using [3]H-thymidine incorporation (data represent the mean SI value of four replicate wells ± s.e.m.). GM-CSF was measured by ELISA and is depicted in (pg ml[−1]) after subtraction of the negative control. **e**, Part of the CFSE[dim]CD4[+] TILs that responded to a bacteria-derived and microbiota-derived peptide pool and a vaccine peptide pool were used for TCRVβ sequencing. The percentage and frequency of TCR overlap of unique productive TCRVβ sequences of CFSE[dim] TILs responding to the vaccine peptide pool and TCRVβ sequences of CFSE[dim] TILs responding to the bacteria-derived and microbiota-derived peptide pool are shown. **f**, TCR sequence overlap of cross-reactive T cells in TILs recognizing that both the vaccine-derived peptide pool and the bacteria-derived and microbiota-derived peptide pool comprise 1% of entire TILs. Also see Extended Data Figs. 5 and 6.

Table 15). These results demonstrate that unbiased epitope discovery for a single TIL-derived TCC was informative regarding the reactivity of bulk T cells, in that TILs were able to respond to various bacteria-derived and microbiota-derived peptides.

## Cross-reactivity in bulk TILs

As several TIL-derived TCCs cross-recognize a tumour antigen and bacteria-derived and microbiota-derived peptides, we addressed whether bulk TILs responding to bacteria-derived and microbiota-derived peptides also recognize tumour antigens. CFSE-labelled TILs were stimulated with a pool of 22 peptides that had been applied in the vaccine of the patient and included somatic mutations and peptides derived from genes that were overexpressed in the autologous

tumour[25] (Fig. 4c and Supplementary Table 13). CFSE-labelled TILs were also separately stimulated with a pool of 23 bacteria-derived and microbiota-derived peptides that were recognized by bulk TILs (Fig. 4c and Supplementary Table 13). Proliferative (CFSE[dim]) CD4[+] T cells were sorted, and part of these were short-term expanded with phytohaemagglutinin (PHA) and IL-2 and the remainder used for TCR sequencing (Fig. 4c). A higher number of T cells within the bulk TILs responded to the vaccine peptide pool than to the bacteria-derived and microbiota-derived peptide pool (Fig. 4c). However, CD4[+] TILs that had been stimulated with the bacteria-derived and microbiota-derived peptide pool also recognized several tumour antigens including laminin subunit α5 (LAMA5), low-density lipoprotein receptor-related protein 6 (LRP6) and SIN3A* peptides, as well as the pools of tumour antigens and bacteria-derived and microbiota-derived peptides by

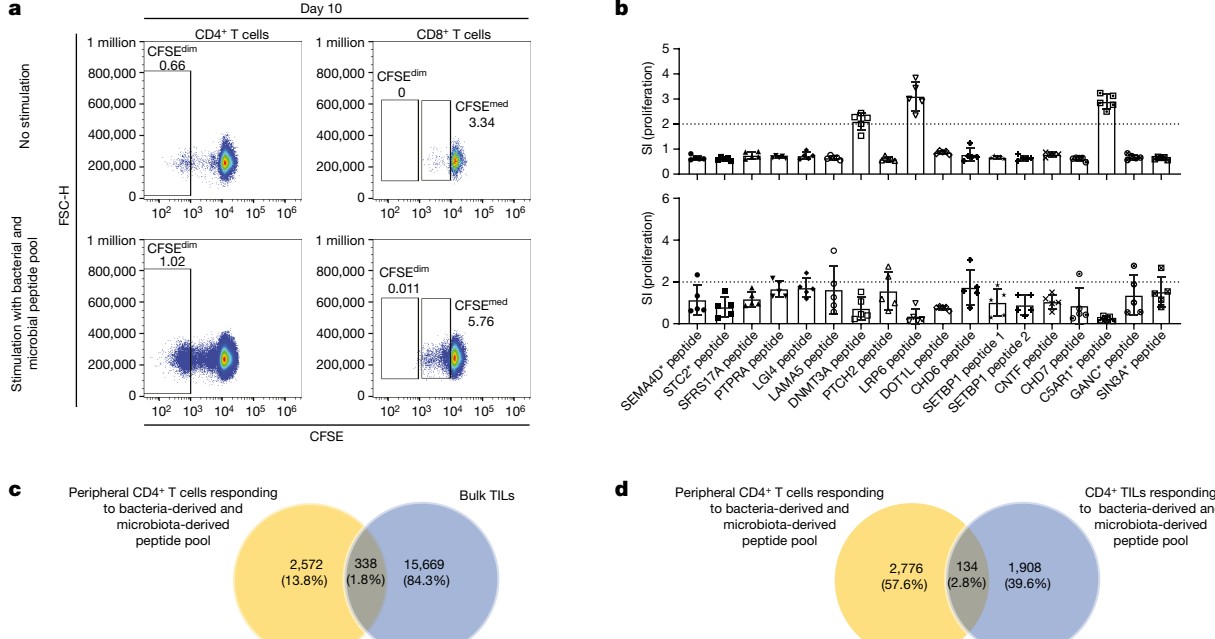

**Fig. 5 | Peripheral memory T cells recognize several tumour antigens after activation with bacteria-derived and microbiota-derived peptides.**
**a**, CD45RA[−] peripheral T cells were isolated and labelled with CFSE. Cells were then stimulated with the bacteria-derived and microbiota-derived peptide pool for 10 days. Next, 60 replicate wells were pooled and proliferative (CFSE[dim]) CD4[+] T cells were sorted and short-term expanded. **b**, Bacteria-reactive or microbiota-reactive peripheral CD4[+] T cells responded to several tumour antigens from the autologous tumour when stimulated with peptide presented by irradiated autologous PBMCs as APCs (top). Proliferation was measured after 5 days by [3]H-thymidine incorporation (data represent the means of five replicate wells ± s.e.m.). Peripheral blood memory T cells were isolated and

directly tested with tumour antigens (bottom). Proliferation was measured after 5 days by [3]H-thymidine incorporation (data represent the means of five replicate wells ± s.e.m.). **c,d**, Part of CFSE[dim] (that is, proliferating) peripheral CD4[+] T cells that responded to the bacteria-derived and microbiota-derived peptide pool was used for TCRVβ sequencing. The percentage and frequency of TCR overlap of unique productive TCRVβ sequences of bacteria-reactive and microbiota-reactive peripheral CD4[+] (CFSE[dim]) and TCRVβ sequences of bulk TILs (**c**), and the overlap of unique productive TCRVβ sequences of bacteria-reactive and microbiota-reactive peripheral CD4[+] (CFSE[dim]) and TCRVβ sequences of (CFSE[dim]) bacteria-reactive and microbiota-reactive TILs (**d**) are shown. Also see Extended Data Figs. 6 and 7.

proliferation and/or GM-CSF secretion (Fig. 4d). Moreover, short-term expanded CD4[+] TILs that responded to the bacteria-derived and microbiota-derived peptide pool showed a comparable cytokine and chemokine profile including several pro-inflammatory cytokines (IFNγ, TNF and granzyme B) after activation with both peptide pools (Extended Data Fig. 6c, left panel). When we compared TCR sequences of the CFSE[dim]CD4[+] T cells responding to the tumour vaccine pool with those responding to the bacteria-derived and microbiota-derived peptide pool, they share 3.7% of the TCRs, indicating that these T cells are cross-reactive against tumour and bacterial antigens (Fig. 4e). The fraction of TCRs that are found in both expanded T cell fractions makes up 1% of the entire TILs (Fig. 4f). From the above TIL-derived TCCs, we only identified TCC75 and TCC88 in the TCR data of both CFSE[dim] fractions, indicating that the fractions of cells that were isolated for TCR sequencing were too small.

## Stimulation of peripheral T cells

When considering bacteria-derived and microbiota-derived peptides for future vaccination approaches, it will be important to show that they induce tumour-specific immune responses in the peripheral immune system. To address whether T cells with reactivity against bacterial peptides from the TCC88 profiling also stimulate peripheral blood memory cells, we CFSE-labelled CD45RA[−] cells from peripheral blood mononuclear cells and activated these with the above pool of bacteria-derived and microbiota-derived peptides that were stimulatory for TILs. CD4[+] T cells responded substantially stronger than CD8[+] T cells to this peptide pool (Fig. 5a). Proliferating CFSE[dim]CD4[+] T cells

were then sorted and short-term expanded with PHA and IL-2 (Fig. 5a). CFSE[mid]CD8[+] T cells were not tested further.

We next addressed whether the sorted and short-term expanded peripheral memory CD4[+] T cells, which had been pre-selected by a pool of bacteria-derived and microbiota-derived peptides, responded to the peptides derived from tumour-specific antigens and tumour-associated antigens of the autologous tumour. These cells recognized three tumour antigens via proliferation, whereas unselected peripheral memory T cells did not (Fig. 5b). In addition to antigen-specific proliferation, they produced various cytokines and chemokines in response to the LRP6 peptide in the selected population compared with the unselected cells (Extended Data Fig. 6c, right panel). To address whether these peripheral blood CD4[+] T cells can be found in TILs, we compared the TCR sequences of bacteria-derived and microbiota-reactive peripheral CD4[+] memory T cells with bulk TILs (Fig. 5c) and TILs selected by stimulation with the bacteria-derived and microbiota-derived peptide pool (Fig. 5d). Of the peripheral blood-derived bacteria-responsive and microbiota-responsive T cells, 1.8% are also present in bulk TILs (Fig. 5c). Furthermore, 2.8% of the T cells are present in both the selected peripheral T cells and TILs responding to the bacteria-derived and microbiota-derived peptide pool (Fig. 5d).

To return to our starting observation with IPdBPs, we examined whether T cells with specificity for these also exist in peripheral blood memory T cells and whether they respond to autologous tumour antigens. We stimulated CFSE-labelled peripheral memory CD4[+] T cells with all 37 IPdBPs, sorted the proliferating cells and short-term expanded them (Extended Data Fig. 7a). These bacteria-activated memory T cells recognized the same three tumour antigens, which are recognized

by peripheral T cells that had been selected with bacteria-derived and microbiota-derived peptides via proliferation and/or GM-CSF cytokine expression as indicators for T cell activation (Extended Data Fig. 7b). Furthermore, the cytokine and chemokine profile of immunopeptidome-reactive peripheral CD4[+] T cells stimulated with the tumour peptide (LRP6) as well as the pool of IPdBPs showed that T cells secreted not only GM-CSF but also several other pro-inflammatory cytokines including IFNγ, TNF and granzyme B (Extended Data Fig. 7c). To address whether there was an overlap between peripheral blood memory T cells responding to IPdBPs and the bulk TILs, we sequenced the TCRVβ repertoires of both populations and found that indeed 0.5% of the TCR sequences were shared (Extended Data Fig. 7d). This indicates that T cells responding to bacterial antigens, which are presented in the tumour, can also be found in the peripheral immune compartment. As another means to show that peripheral blood CD4[+] T cells that had been activated by bacteria and microbiota or IPdBPs recognized the same tumour antigens, we compared the TCR sequences of these two populations. These data support the cross-reactivity results. Of the TCR sequences, 3.4% were shared between the two populations, and most of these TCRs were clonally expanded in each group (Extended Data Fig. 7e). Finally, to assess whether the bacteria-derived and microbiota-derived peptides that were recognized by TCC88 and TILs from patient 1635WI were only stimulatory for T cells of this patient or also for TILs from another patient with glioblastoma, we tested TIL reactivity from another patient with glioblastoma (GBM-40), and indeed these TILs recognized several bacteria-derived and microbiota-derived peptides from patient 1635WI, albeit less strongly (Extended Data Fig. 8a,b).

These data show that (1) T cells with reactivity against bacterial peptides that are presented on HLA class II molecules, that is, the immunopeptidome, within the tumour exist in the peripheral blood, (2) memory T cells that recognize bacteria and microbiota that had been identified by analysing a tumour neoantigen-reactive TCC can be found in the periphery and TILs, and (3) a fraction of T cells from both (1) and (2) also responds to tumour antigens (Extended Data Fig. 9a,b).

## Discussion

Our study examines the role of bacteria for tumour antigen-specific immune responses in glioblastoma, a tumour with poor prognosis and high unmet medical need[36]. We build on recent observations about the involvement of bacteria and particularly gut microbiota and how they may shape immune responses[1,2,37–39]. Findings of bacterial communities in tumours including glioblastoma[20], bacterial peptides in the immunopeptidome of tumours[21] and the influence of commensal bacteria and a bacteriophage on stimulating tumour-specific immune responses[12,13] motivated our studies. We first confirmed data by Nejman et al.[20] that bacterial sequences are found in glioblastoma and by Kalaora et al.[21] regarding bacteria-derived and microbiota-derived peptides in the HLA class II immunopeptidomes of patients with melanoma[21], but here in glioblastoma. Peptides and bacterial strains, from which IPdBPs derived, varied between 5 and more than 50. Hence, compared with tumour-derived human peptide sequences, which usually are a few thousands[24] (Fig. 1c, top), the abundance of bacterial peptides in the HLA class II immunopeptidomes was low. This may be due to low bacterial burden in tumour cells[40] and/or on tumour-infiltrating APCs where bacterial peptides may occupy only a minority of HLA class II molecules and fall below the detection limit[21,41]. Consistent with Nejman et al.[20], we found the phyla Proteobacteria and Firmicutes to be most prevalent in glioblastoma. Although we first focused on IPdBPs, both analyses identified bacterial antigens from Enterobacterales, Burkholderiales, Bacillales, Clostridiales and others at the order-level phylotypes[20]. In a patient, whom we had previously treated with a personalized vaccination using tumour neoepitope peptide cocktails, we tested whether IPdBPs

are recognized by TILs. Compared with the mutated SIN3A* peptide, one of the vaccine peptides, TILs responded very weakly to 3 out of 37 IPdBPs via proliferation; however, secretion of GM-CSF and TNF as well as CXCL10, which enhances cytotoxic CD8[+] T cell trafficking to tumours[42], support the reactivity. CXCL10 has been used to predict responses to anti-PD1 treatment in several tumours[43]. When examining more pro-inflammatory SIN3A*-reactive TCCs, two out of ten recognized single (TCC57) or multiple (TCC2D3) IPdBPs, demonstrating that T cells with cross-reactivity against tumours and IPdBPs can be found in the TILs. With an unbiased antigen discovery approach, we had previously identified targets for autoantigen-specific, virus-specific and bacteria-specific CD4[+] TCCs, and all of these recognized multiple antigens[26–29,32]. Therefore, we addressed, now starting from tumour-infiltrating T cells, whether TIL-derived TCCs respond to viral-derived, bacterial-derived, microbiome-derived, human protein-derived or tumour antigen-derived peptides. TIL-derived, SIN3A*-specific TCC88 served as a prototypic example to search for target antigens among sequences of all human proteins, viruses and bacteria with tropism for humans, gut microbiota, the translated proteome of the autologous primary and recurrent tumours and the tumour immunopeptidome. The sizes of these databases ranged from 16,000 (immunopeptidome) to more than 3.4 billion (human gut microbiota) decamer peptides (Fig. 3a). We synthesized 542 peptides from different sources with predicted high stimulatory scores for TCC88 (Fig. 3a). TCC88 responded to 109 peptides, and, among these, reactivity was broadest to bacterial and microbiota-derived peptides, but also to many tumour proteome-derived peptides (Fig. 3a). Of 226 bacterial and microbiota peptides, TCC88 recognized 63, from which 34 stimulated TCC88 equally well or better than the cognate antigen. Besides *B. ovatus* and *Bacillus* sp., other bacteria-derived and microbiota-derived peptides were derived from completely different bacterial species than IPdBPs (Extended Data Fig. 9c and Supplementary Table 15). These data show that tumour-infiltrating TCCs may cross-recognize not only single bacterial, viral or phage-derived peptides[12,13,44,45] but also large numbers of peptides from both pathogenic bacteria and gut microbiota-derived commensals. Similar to previous data with autoreactive TCCs in MS, we did not identify amino acid patterns or HLA-binding or TCR-binding motifs that allow to distinguish stimulatory from non-stimulatory peptides[46] (Extended Data Fig. 5b) to explain the broad cross-reactivity[47]. The primary HLA restriction element of TCC88 is DRB3*02:02, and similar to virus-specific and autoreactive TCCs that we had previously described[31,32], TCC88 is cross-restricted, that is, capable of recognizing peptides also in the context of other autologous HLA-DR molecules[31,48]. Regarding its function, TCC88 showed in vitro cytotoxicity against an autologous glioblastoma cell line and a multifunctional pro-inflammatory phenotype with strong secretion of TNF, IFNγ, granzyme B, GM-CSF and others. Although the response to SIN3A* and two bacterial peptides is almost identical, the relatively higher secretion of IL-10 upon stimulation with the bacterial peptides may indicate that the latter could both activate, but also dampen, intratumoural T cell responses.

These data raised important questions including whether TCC88 is an unusually cross-reactive clone among the TILs and whether the bacteria-derived and microbiota-derived peptides recognized by TCC88 could also stimulate bulk TILs. Indeed, both were the case. Of the 63 bacteria-derived and microbiota-derived peptides that were recognized by TCC88, 23 also stimulated bulk TILs, and several of these were also recognized by other SIN3A*-specific, TIL-derived TCCs, underscoring the broad reactivity of CD4[+] TCC against bacterial and microbial peptides. Furthermore, the diversity of the TIL response against bacteria and microbiota is supported by the fact that there are SIN3A*-specific TCCs (TCC68 and TCC75) that responded to bacteria-derived and microbiota-derived peptides that had been identified as targets for TCC88, whereas other clones such as TCC8F7 did not recognize them (Extended Data Fig. 9a).

Hence glioblastoma-infiltrating TILs can recognize multiple peptides from bacteria and gut microbiota, and the immunopeptidome analyses reveal that such peptides might be locally presented in the tumour (Extended Data Fig. 9a). If one wanted to use this knowledge for tumour vaccination in the future, it is of interest to examine whether tumour-derived, bacteria-derived and microbiota-derived peptides also stimulate peripheral blood memory T cells. Indeed, upon activation of peripheral blood memory T cells with bacteria-derived and microbiota-derived peptides derived either from the target search for TCC88 or from the tumour immunopeptidome not only identified reactivity against bacterial targets from both sources but also against tumour antigens (Extended Data Fig. 9a). Moreover, comparing the TCR repertoires of bulk peripheral blood memory T cells that had been pre-stimulated with either bacteria-derived and microbiota-derived peptides or with several tumour-derived antigens implies that cross-reactivity, which was shown at the clonal level for TCC88, probably exists for more T cells within these T cell populations (Extended Data Fig. 9b). Of note, one tumour antigen, a peptide from LRP6, was recognized by bulk T cells after pre-stimulation with both bacterial and microbial peptide and IPdBP peptide pools. LRP6 is a co-receptor in the WNT signalling pathway that supports the proliferation of tumour cells, and several lines of evidence have shown that inhibition of LRP6 leads to suppression of tumour cell proliferation and enhances chemosensitivity of cancer cells[49]. Hence, the reactivity of peripheral blood memory T cells overlaps with that of TILs.

Our data highlight several important points. The immunopeptidome analyses suggest that bacteria and gut microbiota can exist within glioblastoma and that peptides from these are presented by HLA molecules on tumour cells and probably also on local APCs. Thus, we confirm previous data and extend these findings by showing that some IPdBPs are weakly recognized by both individual tumour-specific TCCs and bulk CD4$^+$ TILs. Furthermore, systematic unbiased epitope discovery of tumour-specific TCCs identified novel bacteria-derived and microbiota-derived antigens, which were highly stimulatory for clones and bulk TILs. Remarkably, we confirm that individual TCCs recognize not only single bacterial or phage-derived peptides[13,44] but are also broadly cross-reactive with multiple bacteria-derived and microbiota-derived peptides. Such cross-reactivity patterns may explain the response of patients with cancer with certain gut microbiota composition to immune checkpoint inhibitors. Vétizou and colleagues found that *B. fragilis* induced a T$_H$1 phenotype, which led to restoration of treatment responsiveness to CTLA4 blockade in tumour-bearing germ-free mice. Mice transplanted with faeces of CTLA4-responsive patients with melanoma with high *B. fragilis* abundance also responded to the blockade[2]. Moreover, *Ruminococcus bromii* is prevalent in the gut of patients with melanoma who respond to anti-PD1 therapy[38], and *Faecalibacterium* of the Ruminococcaceae family was proposed to favour such responses[38]. Other bacteria such as *Alistipes* spp. and *Eubacterium* spp. are enriched in the gut microbiota of patients with non-small-cell lung carcinoma responding to anti-PD1 therapy[1]. *Alistipes indistinctus* has been shown to restore the response to checkpoint blockade in mice[1]. Using our unbiased approach, we found peptides from *B. fragilis*, *R. bromii*, *Alistipes* and *Eubacterium* genus that stimulated TCC88 (Supplementary Table 15). This observation provides further hints at T cell cross-reactivity as a possible mechanism of effective immunotherapies due to certain gut microbiota composition. Our data indicate that not only commensals but also peptides from certain pathogenic bacteria may be exploited for strong stimulation of T cells against cancer cells (Supplementary Table 15). The reactivity of peripheral blood memory T cells against both bacteria-derived and microbiota-derived as well as tumour peptides may be used for vaccination approaches that combine tumour-derived and bacterial and microbial peptides, use mRNAs encoding such combinations of tumour/self and foreign antigens or even adoptive T cell therapies with TCCs such as TCC88 or TCC2D3 or their TCRs. The recent observation

that bacteria can be involved in tumour metastasis[40] suggests that T cell activation against bacteria and tumour cells may not only be used early in the disease but even at later stages after cancer metastasis.

Limitations of our study include that we could not examine the 'autologous' gut microbiota of patients, their lung microbiome, which appears to have an important role in brain homing of pro-inflammatory T cells[50], and also that the most detailed studies on both tumour immunopeptidome and its annotation to bacteria and microbiota, unbiased target discovery for TIL-derived TCC and their cross-reactivity against bacteria and tumour antigens mainly derive from one patient. However, we identified TIL reactivity against some of the bacteria-derived and microbiota-derived peptide targets from patient 1635WI already in TILs from another patient with glioblastoma (Extended Data Fig. 8). Furthermore, the three different methodologies, that is, 16S rRNA sequencing, immunopeptidomics and unbiased target discovery for a TIL-derived T cell clone, identify largely non-overlapping bacterial and gut microbiota-derived species and peptides. All suggest an involvement of bacteria and gut microbiota in the biology of the tumour and/or immune defence against it; however, future studies will need to examine their roles in more depth.

To explore the above therapeutic applications further, strong emphasis would have to be laid on rapid analysis of the tumour immunopeptidome for bacterial and microbial sequences and/or, even better, the unbiased target discovery for TIL-derived clones, and de-risking potential reactivity against vital organs should be incorporated. Our findings support that this is, in principle, possible and that intratumoural bacteria and gut microbiome communities probably have a role in tumour defence mechanisms and could be exploited systematically in the context of immune checkpoint inhibitors and personalized cancer therapies[1,2,37–39].

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

## Methods

### Patient information and materials

Patient 1635WI, who was treated with a personalized neopeptide vaccination, was a 55-year-old male diagnosed with glioblastoma. After resection, he was treated with standard radiochemotherapy and additionally with the immune checkpoint inhibitor pembrolizumab (anti-PD1) and the anti-VEGF monoclonal antibody bevacizumab. He was interested in participating in an individual treatment attempt with a personalized peptide vaccination therapy[25]. He was fully informed about the potential risks of such a vaccination, which, by definition, is an individual medical treatment ('compassionate use') and not subject to special regulations for medical research according to the Therapeutic Products Act (TPA) or Swiss Federal Human Research Act (HRA). Consequently, individual medical treatments are not approved by the Ethics Committee or Swiss Agency for Therapeutic Products (Swissmedic), as they do not constitute clinical trials for the purpose of systematically gaining knowledge. The patient gave written informed consent to receive the individualized vaccination.

The cohort of 18 additional patients with glioblastoma includes the primary tumour of 16 patients and 2 recurrent tumours from 2 patients, all from the University Hospital Zurich, Zurich, Switzerland. Tumour tissues were immediately frozen after surgery and stored at −80 °C until immunoaffinity purification of HLA–peptide complexes. Autologous tumour cells were also isolated and frozen from several samples ($n = 6$) and immunopeptidomes were analysed (Supplementary Table 1). Written informed consent was obtained from each patient in accordance with the local ethical requirements (KEK-ZH-Nr. 2015-0163). All included patients underwent elective brain tumour surgery at the Department of Neurosurgery at the University Hospital Zurich and were diagnosed with glioblastoma (IDH1 wild type) by a board-certified neuropathologist. No other inclusion or exclusion criteria were applied so we do not expect other biases. Sample size was determined based on the availability of biological materials. To generate autologous tumour cell lines, tumour tissue was washed thoroughly using PBS. Next, they were cut into small pieces and treated with digestion mix including RPMI-1640 medium (Sigma-Aldrich) containing 10% FCS (Eurobio), 1 mg ml$^{-1}$ of collagenase IV (Sigma-Aldrich) and 0.1 mg ml$^{-1}$ of DNase I (Sigma-Aldrich), and incubated for 42 min at 37 °C. The reaction was stopped by adding PBS with 2 mM EDTA (Invitrogen) and the lysate was passed through a filter mesh. Next, tumour cells were isolated using Percoll (Sigma-Aldrich) gradient. Isolated cancer cells were passaged up to 30 times to make sure no other cells would survive before they were snap frozen and used for the immunopeptidome analyses. All cells were regularly tested for mycoplasma contamination and were negative.

### DNA extraction and genotyping

DNA was extracted from PBMCs, tumour cells and tissues of patients with glioblastoma using DNeasy Blood & Tissue Kits (Qiagen). The genotyping of HLA class I and HLA class II was done by high-resolution HLA sequence-based typing (Histogenetics) (Supplementary Table 2).

### PBMCs, cell lines and TCCs

PBMCs of patient 1635WI were isolated from a leukapheresis, and allogenic PBMCs were isolated from buffy coats by Ficoll (Eurobio) density gradient centrifugation.

BLS-DRB1*03:01 (kindly gifted by W. W. Kwok, Benaroya Institute, WA, USA), BLS-DRB1*04:02, BLS-DRB3*02:02 and BLS-DRB4*01:01 are EBV-infected B cells from a patient with BLS type II that are transfected with single HLA-DR molecules of the patient with glioblastoma. Thus, each BLS-DR transfectant expresses a single HLA-DR molecule. In brief, various HLA-DRB complementary DNAs were inserted after the CMV promoter in a pLenti-CMV vector (kindly provided by M. Scharl, University of Zurich). Plasmid vectors carrying different HLA-DRBs were used to prepare lentiviruses using pCMV-dR8.91 (packaging plasmid) and pMD2.G (VSV-G envelope plasmid) (both kindly gifted by M. Scharl). Next, BLS cells expressing HLA-DRA1*01:01 (kindly provided by W. W. Kwok) were transduced with various HLA-DRB lentiviruses and cells were selected using blasticidine after 7 days. BLS cells were maintained and expanded using RPMI-1640 medium (Sigma-Aldrich) containing 10% heat-inactivated FCS (Eurobio), 2 mM L-glutamine (Thermo Fisher Scientific), 100 U ml$^{-1}$ penicillin (Corning) and 100 µg ml$^{-1}$ streptomycin (Corning). BLS cell lines were regularly examined for expression of HLA-DR via FACS. Bulk TILs and isolated TCCs including TCC88 and other TCCs were expanded using irradiated allogenic PBMCs (45 Gy) in T cell medium containing IMDM (GE Healthcare), 5% heat-inactivated human serum (Blood Bank Basel, Switzerland), 1 µg ml$^{-1}$ PHA, 20 U ml$^{-1}$ of human IL-2 (h-IL-2), 2 mM L-glutamine (Thermo Fisher Scientific), 100 U ml$^{-1}$ penicillin (Corning) and 100 µg ml$^{-1}$ streptomycin (Corning). The medium was changed every 3–4 days and new h-IL-2 was added each time.

### Proliferation assay

All peptides were synthesized with acetylated N terminus and amidated C terminus (Peptides & Elephants) (Supplementary Table 10). Proliferation of TCCs and bulk TILs was measured using the $^3$H-thymidine incorporation (Hartmann Analytic) assay after 3 and 5 days, respectively. Isolated TCCs ($2 \times 10^4$ cells per well) were co-cultured with irradiated BLS cells (300 Gy) ($5 \times 10^4$ cells per well) in 200 µl of X-VIVO 15 medium (Lonza) and seeded in 96-well U-bottom plates (Greiner Bio-One). Bulk TILs ($5 \times 10^4$ cells per well) were co-cultured with irradiated autologous PBMCs (45 Gy) ($2 \times 10^5$ cells per well) in 200 µl of X-VIVO 15 medium (Lonza) and seeded in 96-well U-bottom plates (Greiner Bio-One). Peptides were then used at a final concentration of 10 µM to stimulate the TCCs or bulk TILs. Results are shown as SI, that is, the ratio of count per minute in the presence of the peptides divided by the negative control (no peptide). SI ≥ 2 is considered positive.

For testing peripheral blood memory T cells, CD45RA$^-$ cells were isolated using human CD45RA microbeads (Miltenyi). Negatively selected CD45RA$^-$ cells were seeded ($2 \times 10^5$ cells per well) in 200 µl of X-VIVO 15 medium (Lonza) in 96-well U-bottom plates (Greiner Bio-One). Peptides were then added at a final concentration of 10 µM and incubated for 7 days before measuring their proliferation via $^3$H-thymidine incorporation.

To isolate antigen-reactive T cells from the TILs or peripheral T cells, CFSE (Sigma-Aldrich) was used to label the cells. In brief, cells were incubated with CFSE (final concentration of 0.5 µM) for 3 min at room temperature. Next, the labelling was immediately stopped using a serum-containing medium and washed before cells were seeded ($5 \times 10^4$ cells per well) in the presence of irradiated autologous PBMCs (45 Gy) ($2 \times 10^5$ cells per well) in 200 µl of X-VIVO 15 medium (Lonza) and the peptide of interest. After 10 days, cells were pooled, washed once with X-VIVO 15 medium (Lonza) and stained with live-dead yellow (Invitrogen) as well as surface markers and cells sorted with a SH800S Cell Sorter (Sony).

### Sorting bulk T cells

TILs expressing specific TCRBVs were isolated from bulk TILs using a panel of 22 fluorescently labelled TCRBV antibodies that detect 27 different TCRBVs. Twelve TCRBV antibodies were labelled with PE and 10 with FITC (Beckman Coulter). Five million TILs that had been expanded once with the above protocol were thawed and washed once with PBS before incubation with human IgG (1:25) (Sigma-Aldrich) for 30 min to block Fc fragments and reduce nonspecific antibody staining. After another wash, cells were stained with CD4 (1:100) (APC anti-human CD4, eBioscience), CD8 (1:100) (Pacific blue anti-human CD8, BioLegend), TCRBV-PE (1:25) (Beckman Coulter) and TCRBV-FITC (1:25) (Beckman Coulter) antibodies (Supplementary Table 12) for 30 min. Finally, cells were sorted, seeded at $5 \times 10^4$ cells per well in T cell medium without h-IL-2 and rested for 1 day before stimulation with $2 \times 10^5$ cells per well of irradiated autologous PBMCs (45 Gy) and peptide (10 µM).

Furthermore, TILs were labelled with CFSE and $5 \times 10^4$ cells per well were stimulated with irradiated autologous PBMCs ($2 \times 10^5$ cells per well) and pools of 22 glioblastoma-associated peptides that had been used in an individualized vaccination of patient 1635WI[25] or 23 bacteria-derived and microbiota-derived peptides, which had been identified by an unbiased target search for TCC88 (Supplementary Table 13) (see below). Both peptide pools have a final concentration of approximately 25 µM for all the peptides in the pool, and the DMSO concentration was always lower than 0.5%.

Regarding isolation of peripheral memory T cells, CD45RA$^-$ cells were sorted using human CD45RA microbeads (Miltenyi) and labelled with CFSE. Next, $2 \times 10^5$ cells per well were seeded in 200 µl of X-VIVO 15 medium (Lonza) in 96-well U-bottom plates (Greiner Bio-One) in the presence of the bacteria-derived and microbiota-derived peptide pool or the IPdBP pool. The IPdBP pool contains 37 HLA class II-derived bacterial peptides at a final concentration of 20 µM for all the peptides in the pool (Supplementary Table 13). After 10 days, cells were Fc-blocked using human IgG (1:25) (Sigma-Aldrich) and labelled with live-dead yellow (1:1,000) (Invitrogen), PerCP-Cy5.5 anti-human CD3 (1:100) (BioLegend), APC anti-human CD45RA (1:50) (BioLegend), APC-Cy7 anti-human CD4 (1:100) (BioLegend) and Pacific blue anti-human CD8 (1:100) (BioLegend). CD45RA$^-$CD3$^+$CD4$^+$CFSE$^{dim}$ cells were then sorted and $2 \times 10^4$ cells per well were seeded in 96-well U-bottom plates (Greiner Bio-One) with irradiated allogenic PBMCs, PHA and h-IL-2 as described above. The medium was changed every 3–4 days and new h-IL-2 was added each time. If necessary, each well was split into two. After 14 days, cells were cryopreserved.

### T cell cloning and characterization

To generate CD4$^+$ TCCs, bulk TILs were labelled with CFSE and stimulated with SIN3A* peptide (10 µM). After 10 days, live proliferative (CFSE$^{dim}$), CD3$^+$ (1:100) (PerCP/Cy5.5 anti-human CD3 antibody, BioLegend) and CD4$^+$ (1:100) (APC-Cy7 anti-human CD4 antibody, BioLegend) T cells were single-cell sorted into 96-well U-bottom plates (Greiner Bio-One). Cells were then expanded using irradiated allogenic PBMCs in the presence of PHA and h-IL-2 in T cell medium. The medium was changed every 3–4 days and h-IL-2 was added each time until day 12. Cells were then subjected to rapid expansion to expand TCCs. TCCs ($5 \times 10^4$ cells) were transferred into T25 flasks (TPP) and were expanded using $3 \times 10^7$ irradiated allogenic PBMCs (45 Gy) in T cell medium containing anti-CD3 monoclonal antibody (OKT3) (Ortho Biotech). After 1 day, h-IL-2 was added at 50 U ml$^{-1}$; at day 5, the content of the flask was harvested, spun down and resuspended in new T cell medium containing 50 U ml$^{-1}$ of h-IL-2. At day 8, cells were split into new T25 flasks and 20 U ml$^{-1}$ of h-IL-2 was added. At day 16, cells were harvested, washed and cryopreserved in freezing medium including 90% heat-inactivated FCS and 10% DMSO (Applichem).

TCCs were characterized using primers for TCRVβ chains and further sequencing of amplicons[31]. In addition, the LEGENDplex Multi-Analyte Flow Assay kit (BioLegend) was used to identify the functional phenotype of TCC by measuring multiple cytokines in the supernatant following activation with anti-CD2/CD3/CD28 antibody-loaded MACSibead (Miltenyi). The HLA-DR restriction of TCCs was defined by co-culturing TCCs with the four aforementioned irradiated HLA-DR transfected BLS cells (300 Gy), presenting different concentrations of the cognate antigen and measuring proliferation after 3 days using $^3$H-thymidine incorporation.

### Cytokine measurements

Supernatants from each experiment were harvested a day before adding thymidine and frozen at −20 °C. Cytokines were measured using ELISA for human IFNγ ELISA MAX (BioLegend), human GM-CSF Duo-Set ELISA (R&D Systems) and bead-based assays. A Synergy H1 hybrid multi-mode reader (Agilent) was used to measure the absorbance. The LEGENDplex Multi-analyte Flow Assay kit (BioLegend) is a 13-plex human T$_H$ cytokine panel, which includes IFNγ, TNF, IL-2, IL-4, IL-5, IL-6, IL-9, IL-10, IL-13, IL-17A, IL-17F, IL-21 and IL-22. Staining was conducted as suggested by the manufacturer. LSR Fortessa Flow Cytometer (BD Biosciences) was used to measure cytokines, and data were analysed using FlowJo (Tree Star). The Human XL Cytokine Luminex Performance Assay Fixed Panel (R&D Systems) contains 45 analytes covering a broad range of various chemokines and cytokines. Bio-Plex 200 systems (Bio-Rad) was used to measure cytokine and chemokine secretion as suggested by the manufacturer.

### ps-SCL and biometrical analysis

A library of L-amino acid decapeptide ps-SCL mixtures was prepared as mentioned before[30]. TCC88 is a CD4$^+$ TCC isolated from the TILs and is restricted by HLA-DRB3*02:02. Irradiated BLS-DRB3*02:02 cells (300 Gy) were incubated with 200 mixtures from the library at 200 and 100 µg ml$^{-1}$ concentration. TCC88 was then co-cultured with these cells. Each well contained $2 \times 10^4$ TCC88 and $5 \times 10^4$ BLS cells. After 72 h, supernatants were harvested and tested for IFNγ secretion by ELISA (BioLegend). Next, a scoring matrix was generated via log$_{10}$ transformation of median IFNγ values of three independent experiments[26]. Next, biometrical analysis[29] was used to score all 10-mers from several large databases, that is, all human proteins (Swiss-Prot-20201202), the translated transcriptomes of the primary and recurrent autologous glioblastoma tumour (patient 1635WI), all viruses, which are annotated for infecting humans in UniProt (Swiss-Prot-20210305), as well as all bacteria (Swiss-Prot-20201202) and human gut microbiota[34]. To assess the validity of the in silico predictions of stimulatory peptides for TCC88, consecutive 10-mer peptides with the highest predicted scores in each database were synthesized and tested for their capacity to stimulate TCC88. Peptides were selected based on their predicted stimulatory scores and not other factors such as their source. Peptide scores ranged between 80% and 100% of the maximum theoretical score for all databases beside the tumour immunopeptidome (annotated by human proteins), which contained only a few peptides with a score over 80%. Therefore, we also synthesized several peptides within an extended score range between 60% and 80%. Peptides were tested at a final concentration of 10 µM on irradiated BLS-DRB3*02:02 and co-cultured with TCC88 as mentioned before. $^3$H-thymidine incorporation (Hartmann Analytic) was used to measure proliferation.

### T cell cytotoxicity assay

In brief, TCC88 ($2 \times 10^4$ cells per well) was stimulated using the tumour antigen SIN3A* and *Victivallis* gut bacterium HGM3 peptides (2 µM) separately on irradiated autologous PBMCs ($1 \times 10^5$ cells per well) for 72 h in X-VIVO 15 medium (Lonza). Next, the cells were pooled and washed once with X-VIVO 15 medium (Lonza). Autologous glioblastoma cells were stained with PKH26 Red Fluorescent Cell Linker Kit (Sigma-Aldrich) for 3 min and immediately washed three times with serum-containing medium. Target cells were then resuspended in X-VIVO 15 medium (Lonza) and seeded at $2 \times 10^4$ cells per well. Activated and rested TCC88 cells were co-cultured at 1:1 and 5:1 effector to target ratios in duplicates. Next, the plate was centrifuged at 80$g$ for 1 min to ensure cell–cell contact. After 24 h, supernatants were removed and cells were detached using accutase (Thermo Fisher Scientific), washed and stained with LIVE/DEAD Fixable Near-IR Dead Cell Stain Kit (Thermo Scientific), and the percentage of PKH26$^+$Near-IR$^+$ dead tumour cells was assessed using BD LSR Fortessa. HLA-DR expression of cancer cells was also measured using PE-Cy7 anti-human HLA-DR antibody (1:50) (BioLegend). To calculate the specific T cell-mediated cytotoxicity, target cells were separately seeded without TCC88 and their death (%) was considered as unspecific background death.

### TCRVβ sequencing

To compare the TCRVβ sequence of TILs responding to the vaccine peptide pool with those responding to the bacteria-derived and

microbiota-derived peptide pool, CFSE-labelled TILs were seeded at $5 \times 10^4$ cells per well in a 96-well U-bottom plates (Greiner Bio-One) and stimulated with the corresponding peptide pools incubated with irradiated autologous PBMCs ($2 \times 10^5$ cells per well). After 10 days, between $5 \times 10^3$ and $10 \times 10^3$ live CD3$^+$CD4$^+$CFSE$^{dim}$ cells were sorted from each sample. Moreover, to isolate peripheral T cells responding to bacteria-derived and microbiota-derived peptides or the IPdBP pool, CFSE-labelled cells were sorted. Sorted cells from both experiments were spun down and supernatants were removed. The pellet was snap-frozen immediately. The DNeasy Blood & Tissue Kit (Qiagen) was used to extract the DNA, and amplification and sequencing of TCRVβ CDR3 was performed using the ImmunoSEQ platform (Adaptive Biotechnologies). For the TCR data analysis, non-productive TCRVβ sequences were excluded, and shared unique productive TCRVβ sequences were specified based on identical CDR3 sequence as well as V and J chains. Representation of data was done using VENNY 2.1 (BioinfoGP, CNB-CSIC)[51].

## IFNγ fluorospot assay

TILs ($2 \times 10^4$ cells per well) were seeded with $5 \times 10^4$ irradiated (300 Gy) autologous EBV-transformed B cell line cells per well as APCs primed with bacteria-derived and microbiota-derived peptides (10 μM final concentration). Cells were seeded in X-VIVO 15 medium (Lonza) in human IFNγ/IL-10/IL-17A pre-coated fluorospot plates (FSP-010703-10, Mabtech) and were incubated for 44 h at 37 °C. Before seeding, fluorospot plates were washed three times using PBS and then blocked with RPMI-1640 medium (Sigma-Aldrich) containing 10% FCS (Eurobio) as instructed by the manufacturer. After 44 h of incubation, plates were washed using PBS and incubated with detection antibody, anti-IFNγ monoclonal antibody 7-B6-1-BAM, for 2 h. Next, plates were washed using PBS and incubated with anti-BAM-490 fluorophore conjugate for 1 h. Both detection antibody and fluorophore conjugate were diluted with PBS containing 0.1% BSA (Sigma-Aldrich) as instructed by the manufacturer. Next, plates were washed using PBS and incubated with fluorescence enhancer (Mabtech) for 15 min. Plates were emptied by flicking and kept in the dark for 24 h to completely dry. Finally, spot analysis was performed using the automated Mabtech IRIS fluorospot reader (Mabtech).

## Immunopeptidome analyses

To analyse the immunopeptidomes of the tumour tissues and cell lines, frozen tissues and frozen cell pellets were homogenized in CHAPS-containing lysis buffer to perform standard immunoaffinity chromatography[24]. In brief, tumour tissues were washed thoroughly using PBS. Next, they were cut into small pieces and treated with digestion mix including RPMI-1640 medium (Sigma-Aldrich) containing 10% FCS (Eurobio), 1 mg ml$^{-1}$ of collagenase IV (Sigma-Aldrich) and 0.1 mg ml$^{-1}$ of DNase I (Sigma-Aldrich), and were incubated for 42 min at 37 °C. The reaction was stopped by adding PBS with 2 mM EDTA (Invitrogen) and the lysates were passed through a filter mesh. Next, tumour cells were isolated using Percoll (Sigma-Aldrich) gradient. HLA class II molecules were isolated using the HLA-DR-specific antibody (L243)[52] as well as the pan-HLA class II-specific antibody Tü39 (ref. 53) (mixed 1:1, at least 1 mg or 1 mg per 1 g of tissue, both produced in-house at the Department of Immunology, Interfaculty Institute of Cell Biology, University of Tübingen) and subsequent elution of the HLA-bound peptides. Eluated peptides were analysed by nanoflow high-performance liquid chromatography (Dionex UltimateTM 3000 Series liquid chromatography system) and tandem mass spectrometry (MS; Orbitrap Fusion Lumos, Thermo Fisher Scientific). Data-dependent acquisition of data was always conducted in technical triplicates[24]. The raw MS data files were converted into 'mgf' using ThermoRawFileParser[54] and analysed on MASCOT software with a non-redundant eubacteria UniProtKB/Swiss-Prot protein database (12 December 2020 with 334,492 entries) and search parameters were 0.5-Da fragment mass

tolerance and 5 ppm precursor with no enzyme. All spectra were analysed manually. We only considered a spectrum valid if we clearly identified a sequence of four B-ions or Y-ions. The samples were also run on Scaffold (Scaffold version 5.2, Proteome Software) software and the 'protein prophet' reported false discovery rates between 2.4 and 5.4. All the acquired sequences were again re-evaluated by comparing them with UniPort human proteins, viral proteins, bacteriophage protein database[55] and genomic transposable elements (data not shown)[56]. Synthetic IPdBPs were measured with the same MS/MS method and device (Orbitrap Fusion Lumos, Thermo Fisher Scientific) and their MS/MS spectra were compared with those of the immunopeptidome peptides. As our synthetic peptides were synthesized with modifications at the N-terminal (acetylation) and C-terminal (amidation) sites, spectra often look quite different in terms of intensities and the presence or absence of B-ion and Y-ion series. The modification changed the charge of the peptides and the character of ionization patterns. The reason for synthesizing peptides with the above-end modifications stems from a convention that we previously established based on the ps-SCLs search algorithm. The latter decamer peptide mixtures, which are used for the unbiased antigen discovery studies, have N-terminal acetylation and C-terminal amidation, and consequently, we always synthesized the predicted peptides with the same ends. The peptides for the sequence validation of the IPdBPs should have been synthesized with free ends. Despite this limitation, peptide sequencing identified the exact same amino acid sequences for 37 IPdBPs via the synthetic peptides. The comparison of MS/MS spectra of ten synthetic peptides per original spectra of IPdBPs are shown in Supplementary Table 16. Universal Spectrum Explorer (USE) was used to annotate the spectra and Proteome Discoverer (Thermo Fischer) was utilized to do the research in the raw files.

The core binding motif of peptides to DRB1*03:01, DRB1*04:02, DRB3*02:02 and DRB4*01:01 was predicted using the NetMHCII 2.3 server[57], and iceLogo[58] (https://iomics.ugent.be/icelogoserver/) was used for graphical representation of the core binding motifs.

## 16S rRNA gene sequencing

16S rRNA gene sequencing of the tumour tissues was conducted by Microsynth AG. In brief, fresh-frozen tissues from patients were sent to Microsynth AG. Tissues were cut manually using a sterile scalpel, homogenized using Fastprep 24 (45 s, speed 6.5) and total DNA was extracted using ZymoBiomics DNA Mini Kit D4300 (Zymo Research). Negative extraction controls were included. Enrichment PCR of the whole 16S gene (primer pair 27F and 1492R) was performed, followed by amplicon library generation (Nextera library preparation protocol) using the locus-specific primer pair 341F (5′-CCT ACG GGN GGC WGC AG-3′) and 805R (5′-GAC TAC HVG GGT ATC TAA TCC-3′) and sequencing on a MiSeq platform 2 × 250 bp v2 (Illumina).

Reads were demultiplexed by separating reads into individual FastQ files, quality controlled and trimmed of Illumina adaptor sequences using locus-specific bcl2fastq software version v2.20.0.422, FastQC version 0.11.8 (https://www.bioinformatics.babraham.ac.uk/projects/fastqc/) and cutadapt v3.2 (http://journal.embnet.org/index.php/embnetjournal/article/view/200), respectively. Trimmed forward and reverse reads of each paired-end read were merged to in silico reform the sequenced molecule considering a minimum overlap of 15 bases using the software USEARCH (version 11.0.667)[59]. Merged reads were denoised using the UNOISE algorithm implemented in USEARCH to form operational taxonomic units, discarding singletons and chimaeras in the process. The resulting operational taxonomic unit abundance table was then filtered for possible barcode bleed-in contaminations and against the negative controls using the UNCROSS algorithm. Operational taxonomic unit sequences were compared with the reference sequences of the RDP 16S database[60]. In addition, we used the previously reported data on the tumour microbiome[20] with in-depth contamination filtering to further remove potential bacterial contaminants.

## RNA sequencing

RNA sequencing of the tumour tissue (1635WI) and analysis were conducted as previously reported[25]. In brief, RNA sequencing was conducted using the HiSeq 4000 System (Illumina) at the Functional Genomics Center Zurich. Quantile normalized gene counts were compared with The Cancer Genome Atlas (TCGA) glioblastoma cohort.

## Statistical analyses

All analysis were conducted using GraphPad Prism 8.0. Data are expressed as mean ± s.e.m. Proliferation results from incorporation of $^3$H-thymidine are always shown with SI mean ± s.e.m. and, as mentioned before, SI ≥ 2 is considered significant[26,27,32].

## Reporting summary

Further information on research design is available in the Nature Portfolio Reporting Summary linked to this article.

## Data availability

The TCRVB sequencing results reported in this paper are available and can be accessed via the immuneACCESS database of Adaptive Biotechnologies (https://clients.adaptivebiotech.com/pub/naghavian-2023-n). The MS immunopeptidomic raw data have been deposited to the ProteomeXchange Consortium via the PRIDE[61] partner repository with the dataset identifier PXD036811. Gene expression data for the TCGA glioblastoma cohort were downloaded from the Broad GDAC Firehose (Broad Institute of MIT and Harvard; https://doi.org/10.7908/C16T0M0N). Data were also acquired from UniProt human proteins (Swiss-Prot-20201202; https://www.uniprot.org/proteomes?query=(taxonomy_id:9606), UniProt viruses (Swiss-Prot-20210305; https://www.uniprot.org/uniprotkb?query=viruses%20AND%20(virus_host_id:9606), UniProt bacteria (Swiss-Prot-20201202; https://www.uniprot.org/proteomes/?facets=superkingdom%3ABacteria&query=%2A) and for human gut microbiota[34]. The RNA sequencing raw data have been deposited to the European Nucleotide Archive with the accession code PRJEB56507 (ref. 25). Source data are provided with this paper.

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

**Acknowledgements** Study support came from a European Research Council grant (ERC-2013-ADG-340733) and a Swiss National Science Foundation grant (SNF 320030_153213). We thank I. Jelcic, T. Lorenzini, M. Wawrzyniak and L. Grob for their help; A. Lutterotti, R. Planas, P. Tomas Ojer, T. Weiss, P. Roth, H. Moch, E. Rushing, S. Dettwiler, B. Schrörs, J. H. Shin, R. McKay, C. J. Wu and E. F. Greiner who participated in the personalized vaccination[25]; and C. Falkenburger, U. Wulle, P. Hrstić, N. Bauer and B. Pömmerl for excellent technical support. Figures 1a and 2a and Extended Data Fig. 3b were created with BioRender (https://biorender.com).

**Author contributions** R.N. designed, performed and analysed experiments (proliferation assays, cell sorting, cytotoxicity testing, TCR sequencing analysis, T cell cloning, cytokine and ELISA analysis) and contributed to writing the manuscript. W.F. performed immunopeptidome annotations with bacteria and microbial databases. J.W. participated in experimental design and performed T cell cloning. P.O., N.C.T. and Y.Z. provided protein databases, the tool for peptide searching and performed target searches. M.C.N. provided tumour tissue and cancer cells for immunopeptidome analyses, and participated in designing the immunopeptidome and cytotoxicity experiments and discussing the manuscript. L.K.F. and H.-G.R. performed immunopeptidome analyses and commented on the manuscript. M.Wacker and J.S.W. provided the MS/MS fragmentation of synthetic peptides and the analyses. G.M. provided patient-derived cancer cells for the cytotoxicity experiments and participated in designing the assay. S.D.B. assisted with bacterial taxonomy and 16S rRNA sequencing. M.Weller provided tumour tissue and patient-derived cancer cells and immunopeptidome data, participated in designing patient cohort and logistics, and reviewed the manuscript. L.R. provided tumour tissue, participated in designing the patient cohort and logistics, and reviewed the manuscript. M.M. provided additional glioblastoma-derived TILs and tumour tissues, and participated in designing the experiment. P.-E.B. and S.A. contributed with annotation of the immunopeptidome with non-coding sequences. Y.Q. performed expansion of cells and testing with autoantigen peptide pools. M.S. provided expertise in epitope mapping and interpreting the scoring matrix data. R.M. supervised and designed the overall study, interpreted the data and wrote the manuscript.

**Funding** Open access funding provided by University of Zurich.

**Competing interests** R.N., W.F. and M.S. are currently employees of Cellerys AG. R.M. received unrestricted grants from Biogen, Novartis, Roche, Third Rock; has advisory roles and lectures for Roche, Novartis, Biogen, Merck, Genzyme, Neuway, CellProtect, Third Rock and Teva; is listed as an inventor on a NIH-held patent of daclizumab in MS and University of Zurich-held patents on JCV VP1 for vaccination against PML, JCV-specific neutralizing antibodies to treat PML, antigen-specific tolerization with peptide-coupled cells and novel autoantigens in MS, the use of designer neoantigens for glioblastoma (inventor R.M.), and of peptides and TCRs shown in this study for the personalized treatment of glioblastoma (inventors R.N., J.W. & R.M.); and is a co-founder of Abata, and a co-founder and employee of Cellerys AG. L.K.F. is an employee of Immatics Biotechnologies. M.C.N. has received a research grant from Novocure, and honoraria for consulting or lectures from WISE, MSD and Novocure. M.Weller has received research grants from Quercis and Versameb, and honoraria for lectures or advisory board participation or consulting from Bayer, Curevac, Medac, Merck (EMD), Novartis, Novocure, Orbus, Philogen, Roche and Sandoz.

**Additional information**
**Correspondence and requests for materials** should be addressed to Roland Martin.

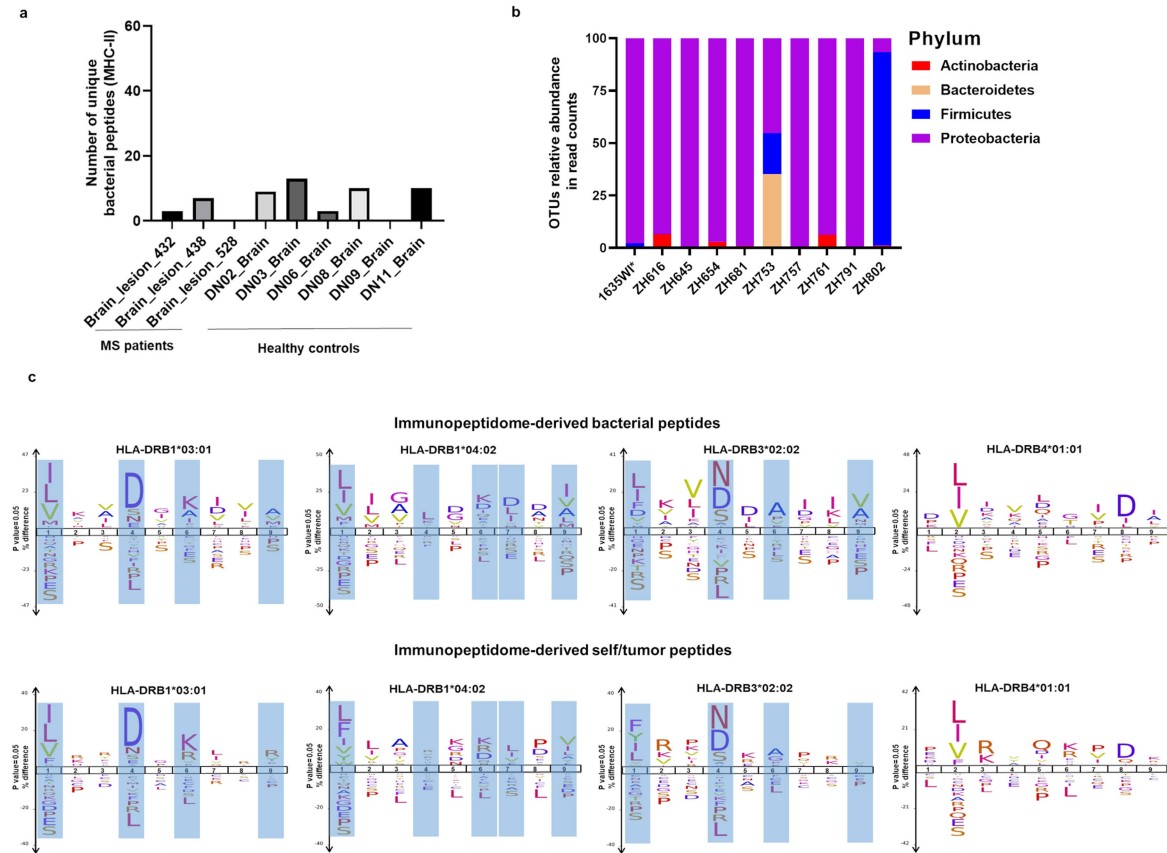

**Extended Data Fig. 1 | Confirming the presence of bacteria in tumor tissues (see also Fig. 1, Extended Data Fig. 2 and Supplementary Table 4). a**, HLA-II immunopeptidome analyses for bacterial peptides in control including brain lesions of 3 MS patients and brain tissue of 6 healthy donors. A total of 55 bacterial peptides was identified from all the samples. Identical bacterial peptides in the control cohort were removed from the GBM tumor-specific bacterial peptides. **b**, 16S rRNA gene sequencing confirmed bacteria in tumor tissues of 10 GBM patients, from whom immunopeptidome analyses were available. OTUs relative abundance in read counts were shown at the phylum

level for each tumor (Supplementary Tables 5 and 6). *16S rRNA data from patient 1635WI includes both primary and recurrent tumors of the patient in triplicates. OTUs relative abundance was calculated based on mean OTU of the triplicates divided by total OTUs in each tumor. **c**, Deduced core binding motif all HLA-DR-derived peptides annotated to human/tumor proteins and UniProt bacterial peptides from primary and recurrent tumors (patient 1635WI) were compared using iceLogo (p = 0.05, two-tailed independent *t*-test by IceLogo[58]). Light blue squares show the known anchor positions for corresponding HLA-DR molecules[62].

**HLA-DR mRNA expression in tumor tissue (1635WI)**

HLA-DRA

HLA-DRB1

HLA-DRB5

**HLA-DR expression of cancer cell line isolated from GBM tumor tissue**

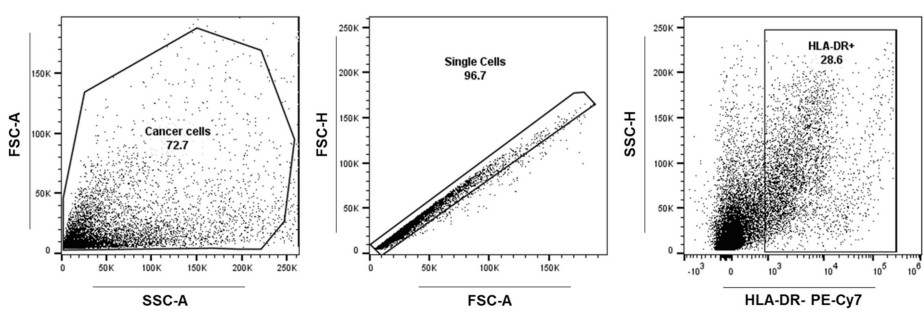

GFAP/CD31/CD3/CD68/HLA-DR

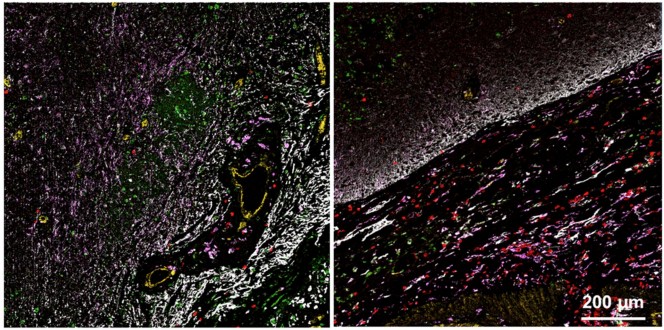

200 µm

**Tumoral areas**

**Extended Data Fig. 2 | HLA-II expression of tumor tissue and cancer cell line. a**, RNA-seq analysis of the tumor tissue (1635WI) was conducted by TCGA-RNAseqv2 pipeline (https://webshare.bioinf.unc.edu/public/mRNAseq_TCGA/UNC_mRNAseq_summary.pdf)[25]. Normalized gene counts of HLA-DR were compared with The Cancer Genome Atlas (TCGA) GBM cohort. White square represents normal expression of the gene in the tumor (Supplementary Table 14). **b**, Cancer cells isolated from the tumor tissue (patient 1635WI) and expanded *in vitro* were stained with HLA-DR antibody. **c**, Paraffin-embedded recurrent tumor tissue (patient 1635WI) was stained for GFAP, CD31, CD3, CD68 and HLA-DR[25]. HLA-DR is present on the surface of immune cells (CD3+), endothelial cells (CD31) and GFAP+ astrocytes (6 different regions in n = 1 section of the tumor were used for this analysis). Scale bar: 200 µm.

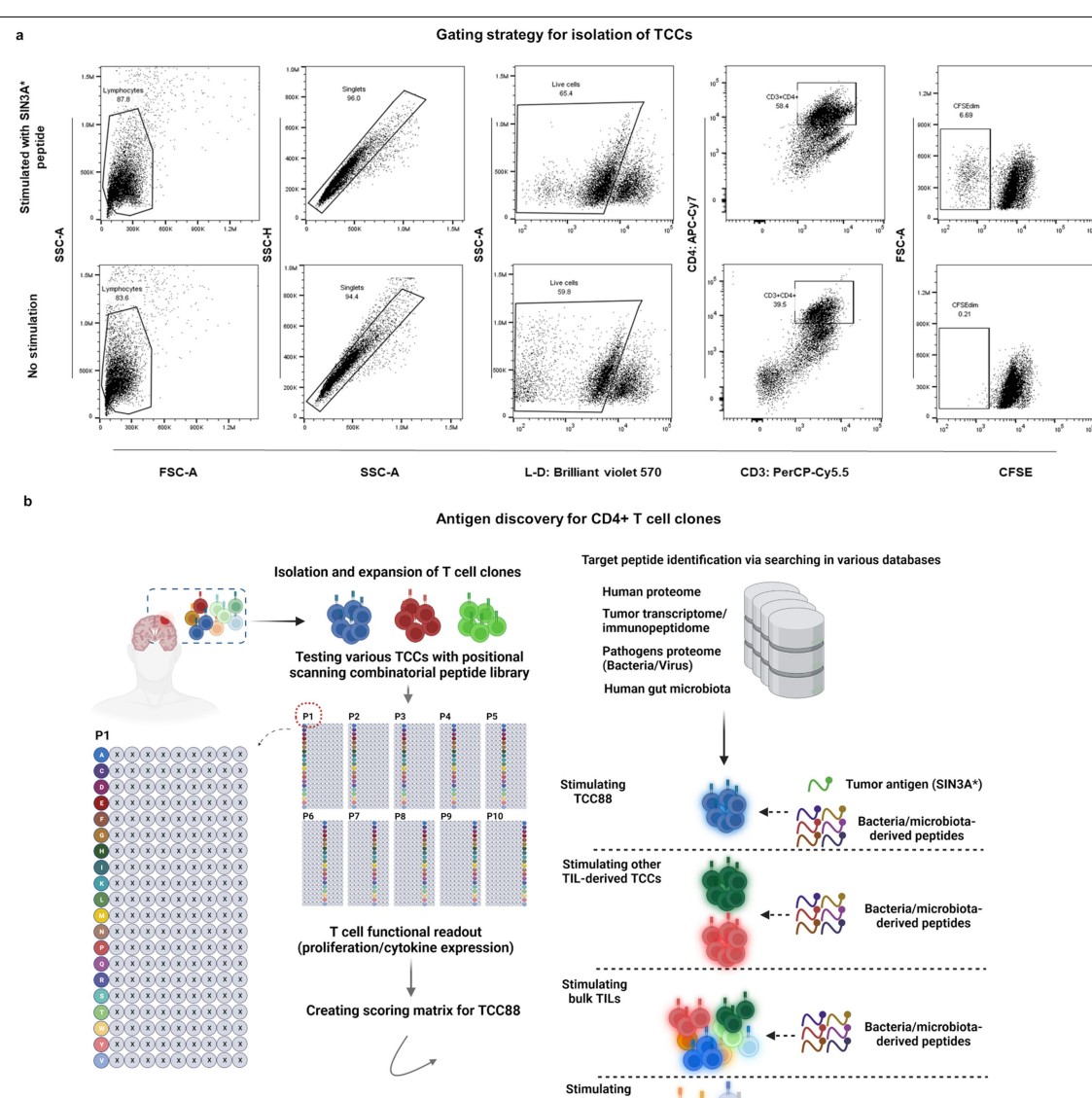

**Extended Data Fig. 3 | T cell cloning using SIN3A* tumor antigen and graphical summary of TCC isolation, response of TCC88 to ps-SCL, and depiction of database screening for antigen discovery approach (see also Figs. 1 and 2 and Extended Data Fig. 4). a**, Gating strategy for isolation of TCCs from TILs responding to SIN3A* peptide. 60 replicate wells of CFSE-labeled TILs were pooled after 10 days stimulation with SIN3A* tumor antigen. Cells were stained using live-dead, CD3, CD4 and were next single-cell sorted. **b**, Schematic depiction of the steps for discovering target antigens for a TCC. TCCs are tested with positional scanning combinatorial peptide libraries (200 mixtures of decamer peptides) presented by BLS cells expressing a single HLA-DR molecule, which had been previously established as the restriction element for each TCC. Proliferative or cytokine responses of TCC to each of the decamer mixtures, i.e. 200 compounds are measured. Data are then used to create a scoring matrix. Antigens with predicted high stimulatory potential are identified by searches of several databases including human proteome, autologous tumor transcriptome and peptidome, pathogenic bacteria and viruses as well as human gut microbiota. Candidate peptides originating from pathogenic bacteria and gut microbiota strongly stimulated TCC88. Further, these peptides were used to stimulate other tumor-reactive TCCs as well as bulk TILs and peripheral memory CD4+ T cells. The schematics in panel **b** were created using BioRender (https://biorender.com).

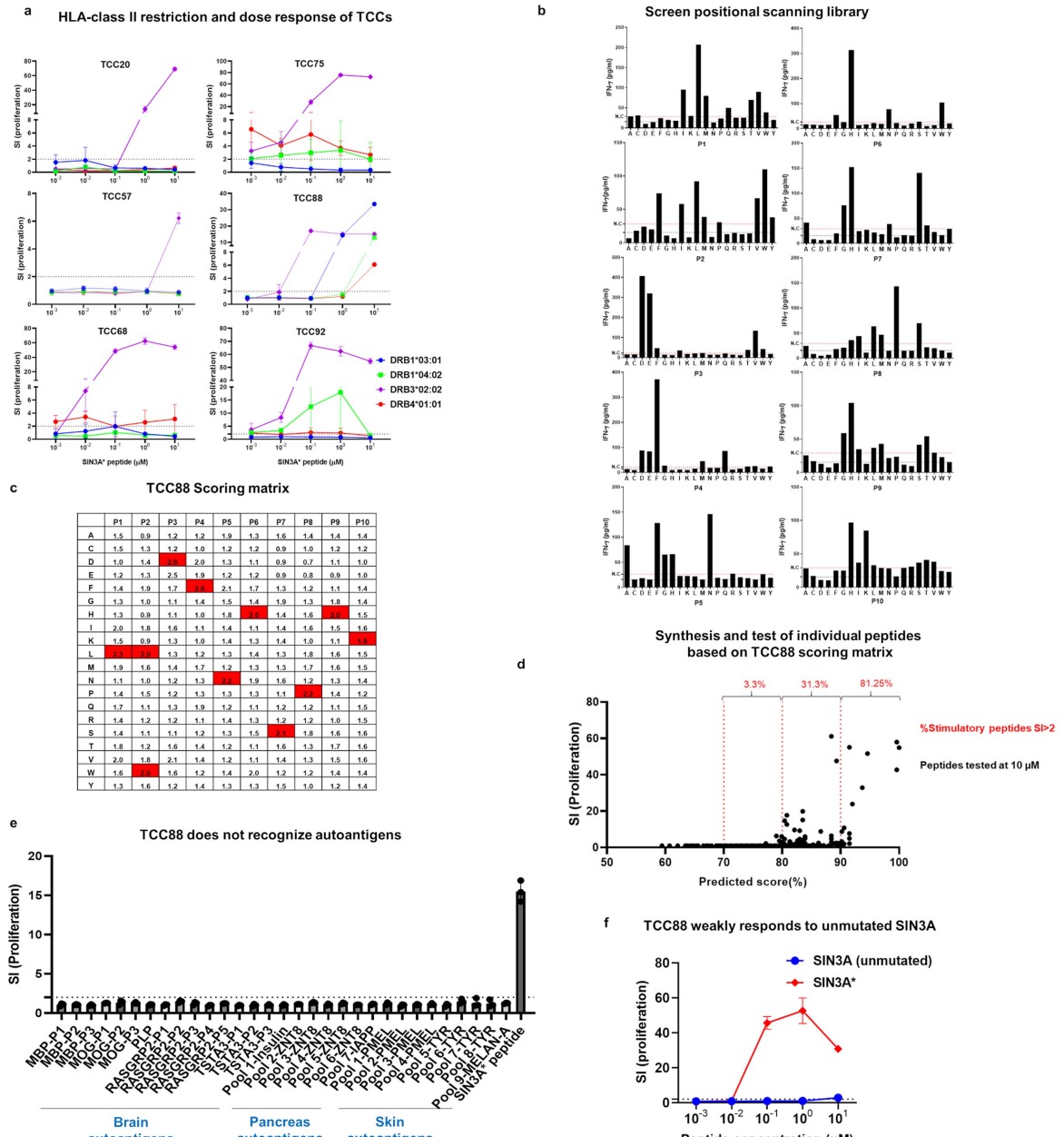

**Extended Data Fig. 4 | (see also Fig. 3). HLA-DR restriction and response to autoantigens of TCC88. a**, Several of the isolated TCCs were stimulated with irradiated BLS cells expressing single HLA-DR molecules of patient 1635WI (BLS-DRB1*03:01, BLS-DRB1*04:02, BLS-DRB3*02:02 and BLS-DRB4*01:01) as APCs and dose titration of SIN3A* peptide (10, 1, 0.1, 0.01 and 0.001 μM). Proliferation was measured using ³H-thymidine incorporation after 3 days. Data represents mean SI of 5 replicate wells ± SEM). Several TCCs did not respond to peptides presented on any of HLA-DR molecules (data not shown). **b-c**, Median IFN-γ response of three independent experiments (transformed to log base 10 values) of TCC88 stimulated via ps-SCL was measured and depicted for 20 L-amino acids fixed in all positions of a 10-mer peptide. **c**, IFN-γ values were depicted in a table format (scoring matrix) and represent the relative stimulatory potency of each L-amino acid in different positions of the decamers. **d**, Relationship between predicted score and stimulatory index (SI) was established by testing potential cognate antigens within the predicted score range of 60–100% of the maximal score with TCC88. **e–f**, TCC88 was tested with pools of overlapping peptides (including 260 peptides) covering brain, pancreas and skin autoantigens (Supplementary Table 11) including peptide pools covering myelin basic protein (MBP pools), myelin oligodendrocyte glycoprotein (MOG pools), myelin proteolipid protein (PLP pool contains 5 immunodominant PLP peptides), RAS guanyl-releasing protein 2 (RASGRP2 pools), GDP-L-fucose synthase (TSTA3 pools), MELAN-A, tyrosinase (TYR), premelanosome protein (PMEL), insulin, SLC30A8 (ZnT8) and islet amyloid polypeptide (IAPP). TCC88 was stimulated with peptide-pulsed irradiated BLS-DRB3*02:02 cells for 3 days. Proliferation was measured using ³H-thymidine incorporation (data represents mean value of 3 replicate wells ±SEM). **f**, Response of TCC88 to unmutated SIN3A peptide was compared to SIN3A* at different concentrations including 10, 1, 0.1, 0.01 and 0.001 μM.

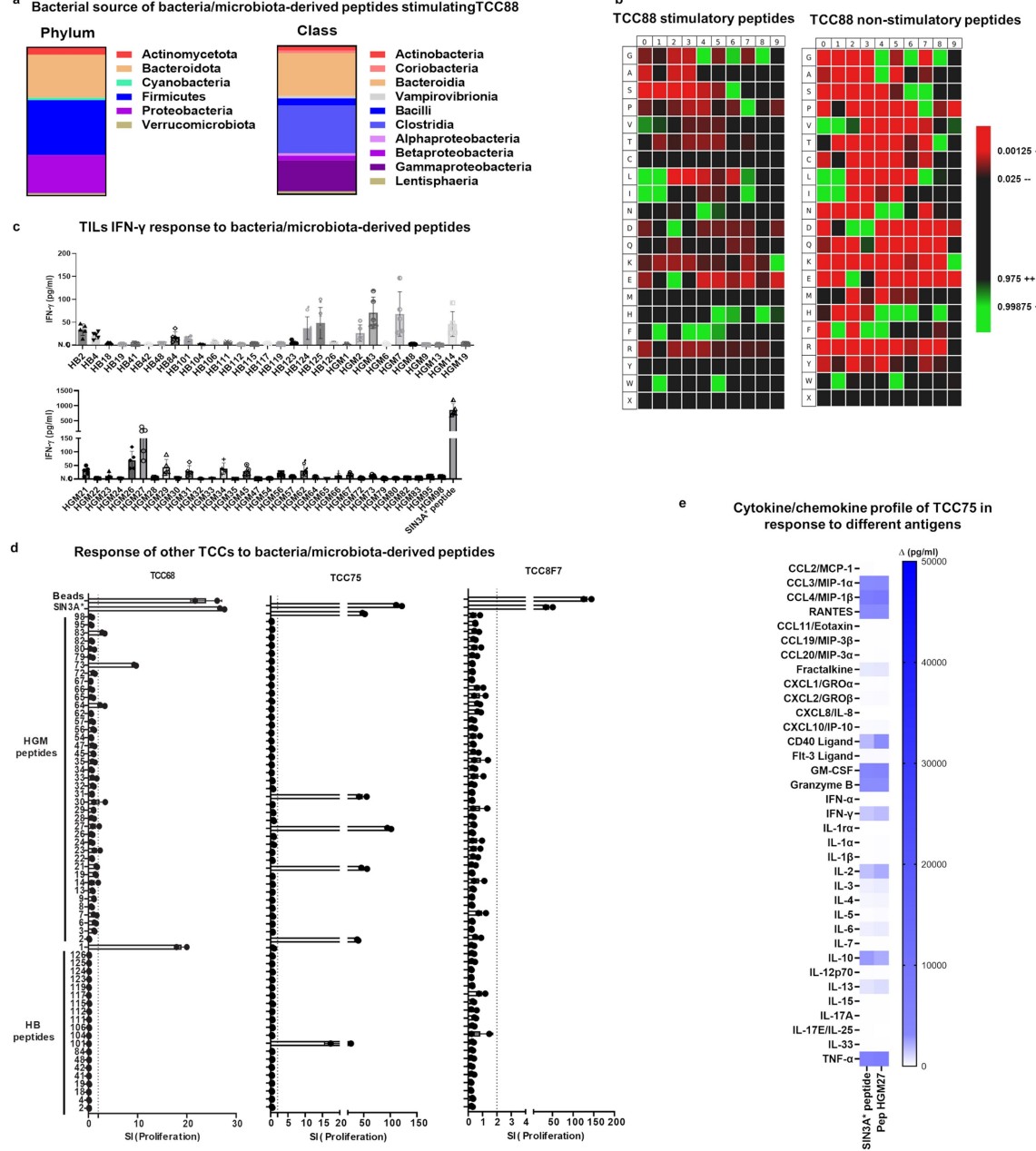

**a**, Bacterial source of bacteria/microbiota-derived peptides stimulating TCC88

**b**, TCC88 stimulatory peptides. TCC88 non-stimulatory peptides

**c**, TILs IFN-γ response to bacteria/microbiota-derived peptides

**d**, Response of other TCCs to bacteria/microbiota-derived peptides

**e**, Cytokine/chemokine profile of TCC75 in response to different antigens

**Extended Data Fig. 5 | Response of TILs and TCCs to bacteria/microbiota-derived peptides (see also Figs. 3 and 4). a**, The source bacterium of bacteria/microbiota-derived peptides that stimulate TCC88 is shown at the phylum and class levels. Source proteins of the peptides were used to check within UniProt databases and taxonomy data was then extracted. **b**, Heatmap result analyzing all peptides that were stimulatory or non-stimulatory for TCC88 with the aim to examine if amino acid motifs are present in stimulatory peptides (Green, black and red represent higher, average and lower prevalence of distinct amino acids in the positions of the tested 10-mer peptides. IceLogo[58] utilizes the z-score to calculate amino acid specific p-values at each position via an error function[58]). **c**, Supernatants were collected from bulk TILs stimulated with bacteria/

microbiota-derived peptides after 5 days. IFN-γ secretion of TILs was measured by ELISA (data represents mean value of 5 replicate wells ± SEM). **d**, Several TIL-derived TCCs were stimulated with bacteria/microbiota-derived peptides and irradiated autologous PBMCs as APCs. Proliferation was measured by [3]H-thymidine incorporation after 3 days (data represents mean value of 2 replicate wells). **e**, Supernatants were collected after stimulation of TCC75 with SIN3A* and HGM27 peptides for 3 days. Cytokine/chemokine secretion of TCC75 was examined for the different antigens and depicted in a heatmap. Specific amount of each cytokine/chemokine (pg/ml) was calculated by subtracting the cytokines in the negative control cultures (no peptide control).

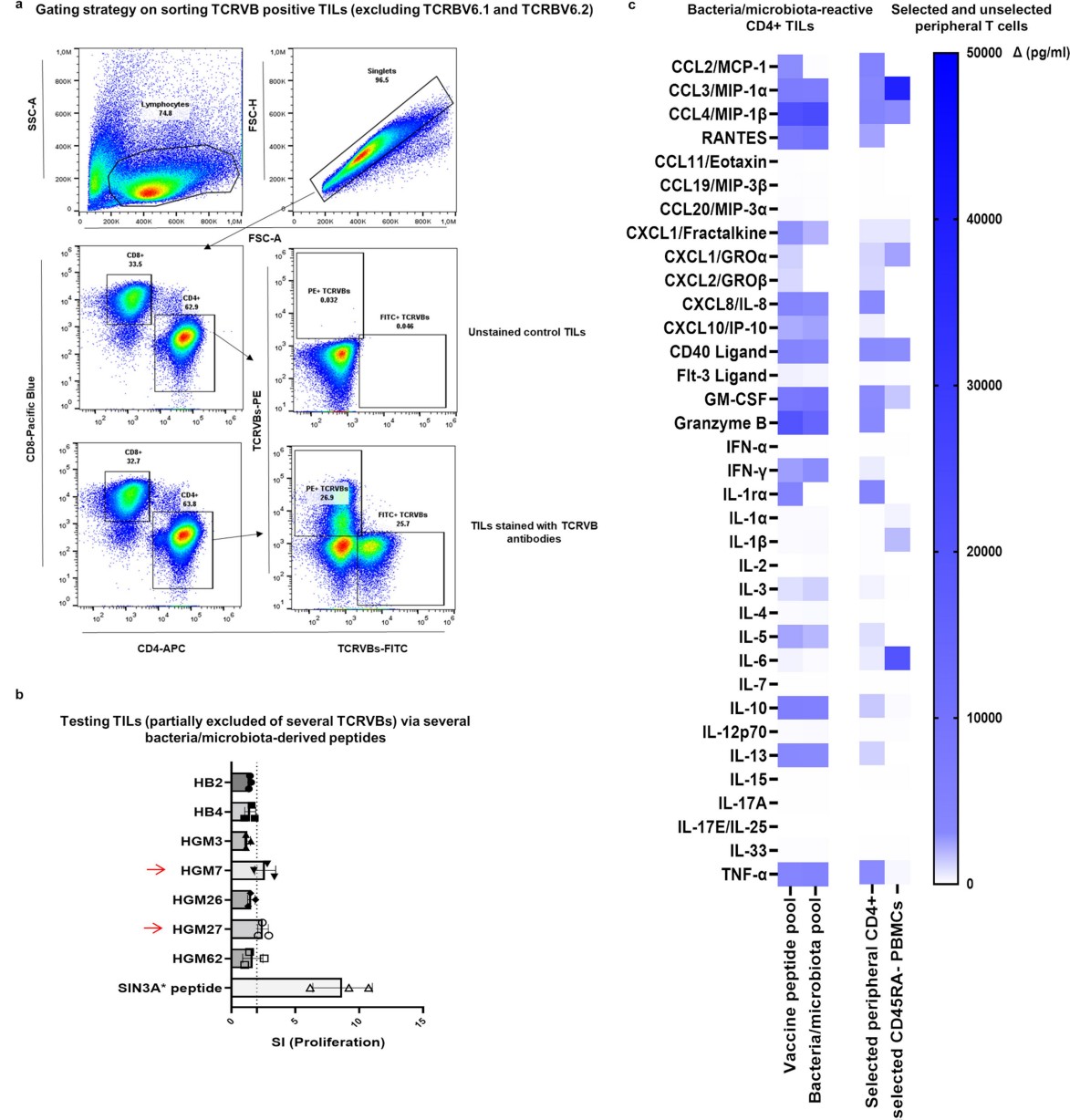

**a** Gating strategy on sorting TCRVB positive TILs (excluding TCRBV6.1 and TCRBV6.2)

**b** Testing TILs (partially excluded of several TCRVBs) via several bacteria/microbiota-derived peptides

**c** Bacteria/microbiota-reactive CD4+ TILs / Selected and unselected peripheral T cells

**Extended Data Fig. 6 | TILs depleted of TCC88 and also other TCCs respond to bacteria/microbiota-derived peptides (see also Figs. 3 and 4). a**, Gating strategy for bulk TILs using CD4, CD8, PE- and FITC-TCRVβ antibodies. Two populations of CD4⁺ PE-TCRVβ⁺ and CD4⁺ FITC-TCRVβ⁺ were sorted and later mixed together. Unstained TILs were used as control to demonstrate specific staining of TILs with TCRVβ antibodies of interest. **b**, Sorted CD4⁺ PE and FITC TCRVβ⁺ were mixed and rested. Cells were then stimulated with irradiated autologous PBMCs pulsed with several bacteria/microbiota-derived peptides at 10 μM and proliferation was measured after 3 days using ³H-thymidine incorporation (data represents mean value of 3 replicate wells ± SEM).

**c**, Supernatants were tested for secretion of a broader range of cytokines/chemokines. Cytokine/chemokine secretion of bacteria/microbiota-reactive TILs was measured after stimulation with vaccine and bacteria/microbiota-derived peptide pools (left panel). Cytokine/chemokine secretion of bacteria/microbiota-reactive peripheral CD4⁺ T cells (selected peripheral CD4⁺) and CD45RA⁻ peripheral T cells (unselected peripheral CD4⁺) in response to a tumor antigen (LRP6) was measured (right panel). Specific amount of each cytokine/chemokine (pg/ml) was calculated by subtracting the cytokines in the negative control cultures (no peptide control).

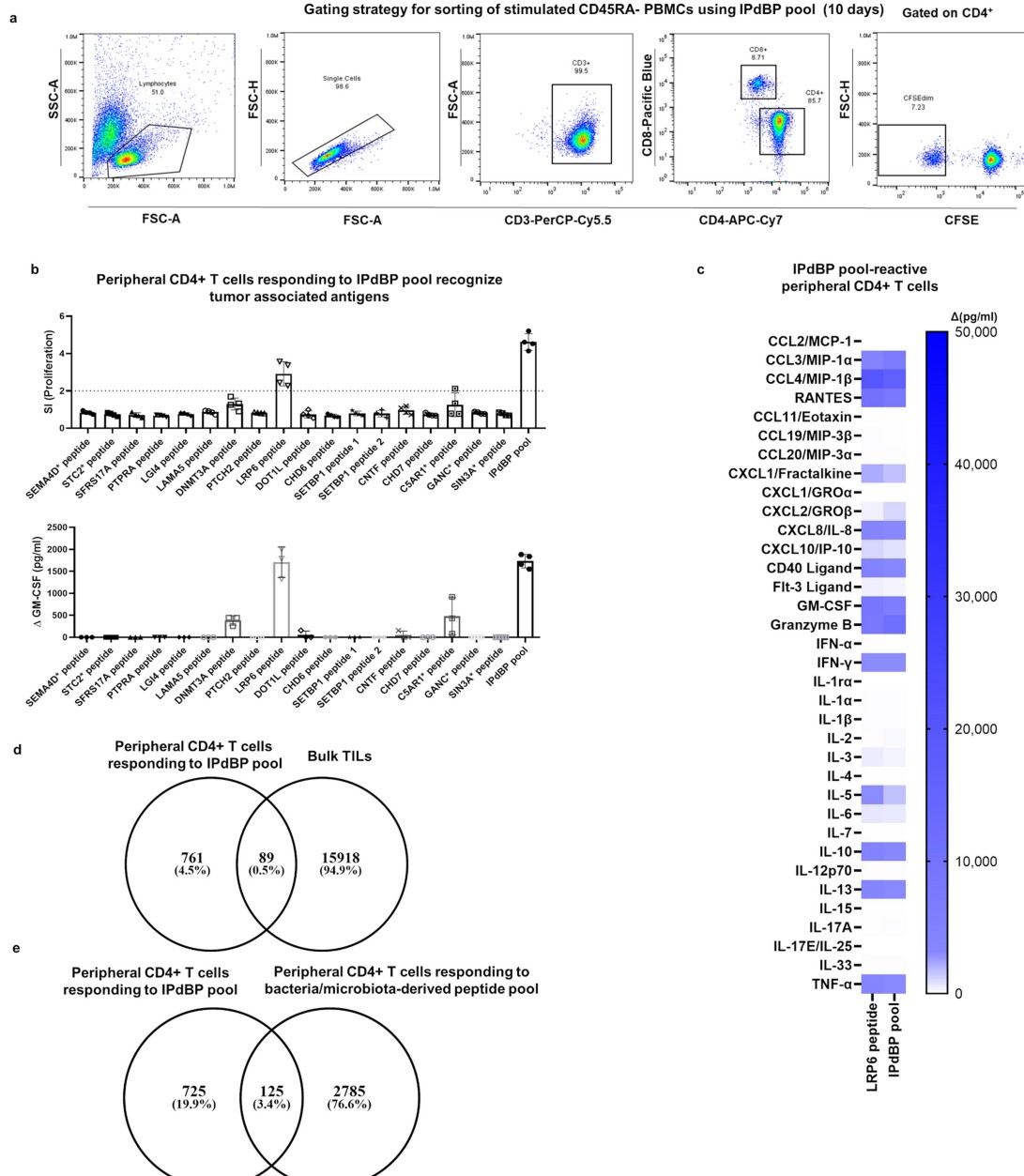

**Extended Data Fig. 7 | Stimulatory effect of IPdBPs on peripheral memory T cells (see also Fig. 5). a**, CD45RA⁻ PBMCs from patient 1635WI were CFSE-labeled and stimulated with a pool of IPdBPs (see Supplementary Table 13). 60 replicate wells were pooled after 10 days and stained with CD3, CD4 and CD8. Proliferating (CFSE^dim) CD4⁺ T cells were sorted and short-term expanded. **b**, Expanded and IPdBP-pre-stimulated peripheral CD4⁺ T cells were tested with 10 µM of autologous tumor peptides derived from over-expressed or mutated proteins (Supplementary Table 13). Proliferation was measured using ³H-thymidine incorporation (data represents mean value of 4 replicate wells ±SEM). GM-CSF ELISA was used as another read-out for T cell activation. GM-CSF was measured by ELISA and is depicted in (pg/ml) after subtraction of the negative control. The peptide pool that was used to pre-stimulate peripheral

blood CD4⁺ T cells (IPdBP pool) served as a positive control (see material and methods section). **c**, Cytokine/chemokine response of peripheral memory CD4⁺ T cells that had been pre-stimulated with the pool of IPdBPs and short-term expanded *in vitro*, to the autologous tumor peptide (LRP6) as well as the IPdBP pool was measured. Specific amount of each cytokine/chemokine (pg/ml) was calculated by subtracting the cytokines in the negative control cultures (no peptide control). **d**–**e**, Percentage and frequency overlap of unique productive TCRVβ sequences of CFSE^dim peripheral memory CD4⁺ T cells responding to IPdBP pool with TCRVβ sequences of bulk TILs. Overlap of unique productive TCRVβ sequences of IPdBP-reactive peripheral CD4⁺ (CFSE^dim) (**d**) and bacteria/microbiota-reactive peripheral CD4⁺ (CFSE^dim) (**e**).

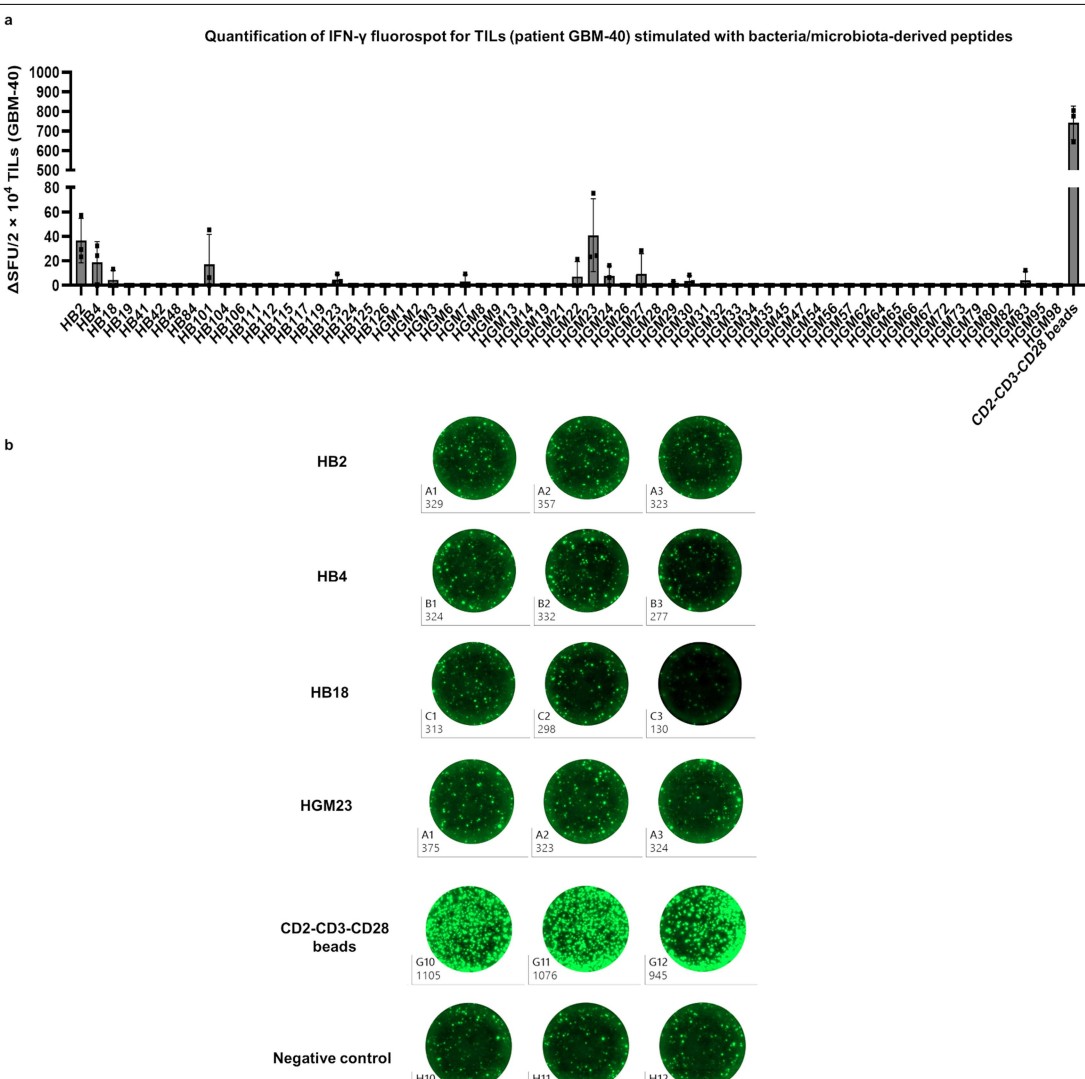

**Extended Data Fig. 8 | IFN-γ fluorospot testing of expanded TILs from patient GBM-40. a**, 64 bacteria/microbiota-derived peptides that were stimulatory for TILs from patient 1635WI were tested with TILs from GBM-40 patient. TILs were seeded with irradiated autologous BLCLs primed with bacteria/microbiota-derived peptides and in triplicates. ΔSFU (spot forming units) for each peptide was calculated after subtraction of the mean number of spots in negative control (data represents mean value of 3 replicate wells ± SEM). **b**, Positive and negative responses of TILs to several bacteria/microbiota-derived peptides are shown on the lower panel. CD2-CD3-CD28 beads are used as positive control.

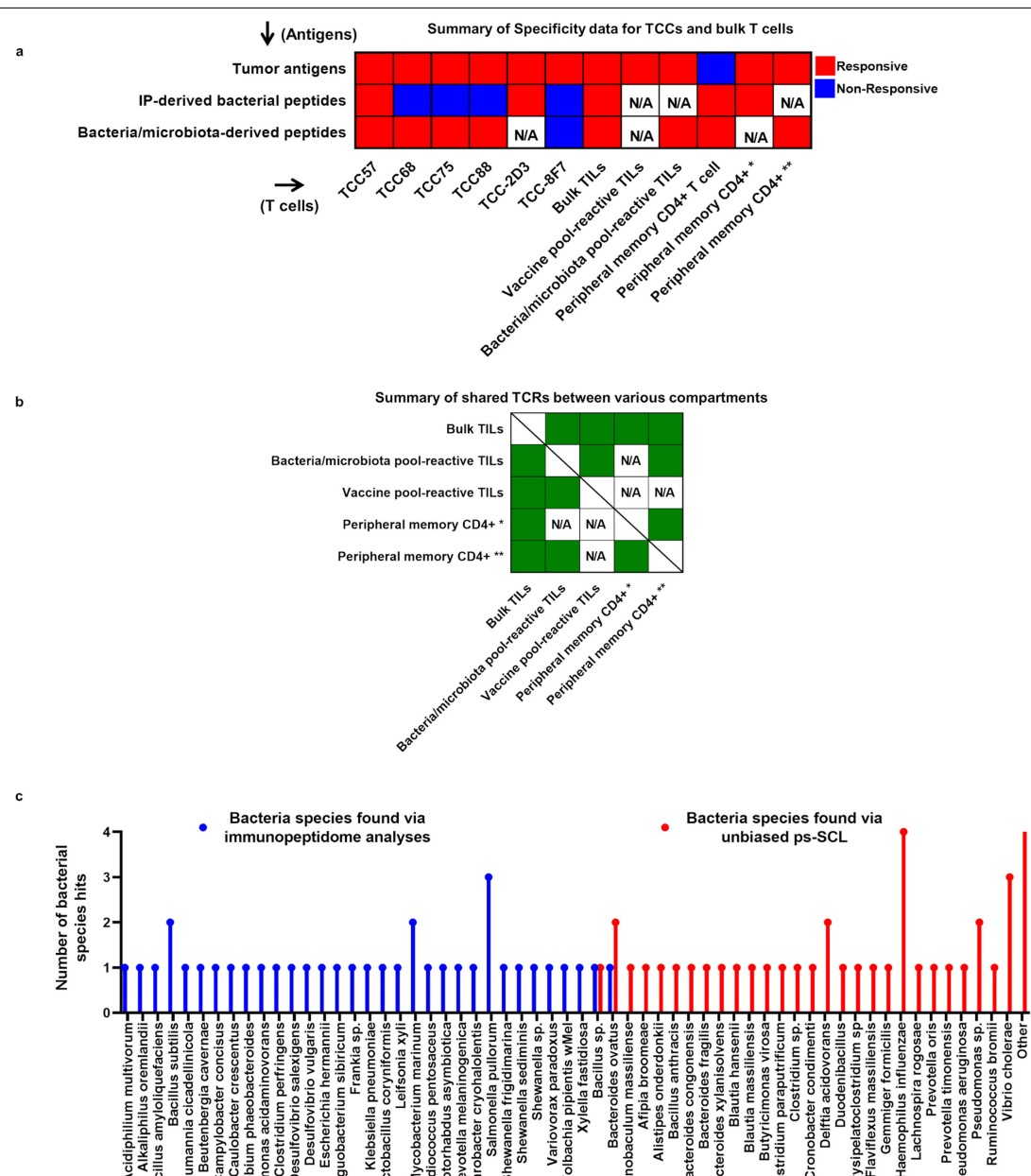

**Extended Data Fig. 9 | Summary of T cell specificities, TCR sharing and comparison of bacterial species identified via immunopeptidome analyses and ps-SCL screening of TCC88. a**, Overview of proliferative responses of all tested TCCs and bulk T cell populations to different antigens. Stimulatory antigens are depicted in red, non-stimulatory antigens in blue (N/A means that T cells were not tested with the respective antigens, * Peripheral memory CD4⁺ T cells selected by IPdBPs, ** Peripheral memory CD4⁺ T cells selected by bacteria/microbiota-derived peptides). **b**. Overview of shared

TCRs between T cells from different compartments. Green displays the presence of shared TCRs (N/A means that samples were not examined for presence of shared TCRs, * Peripheral memory CD4⁺ T cells selected by IPdBPs, ** Peripheral memory CD4⁺ T cells selected by bacteria/microbiota-derived peptides). **c**. Comparison of peptides from bacterial species identified via HLA-II immunopeptidome analysis versus those derived from unbiased ps-SCL screening of TCC88. Bars of blue and red for *Bacillus sp.* and *Bacteroides ovatus* mean that these species were found via both approaches.

# Reporting Summary

## Statistics

For all statistical analyses, confirm that the following items are present in the figure legend, table legend, main text, or Methods section.

| n/a | Confirmed | |
|---|---|---|
| ☐ | ☒ | The exact sample size (*n*) for each experimental group/condition, given as a discrete number and unit of measurement |
| ☐ | ☒ | A statement on whether measurements were taken from distinct samples or whether the same sample was measured repeatedly |
| ☐ | ☒ | The statistical test(s) used AND whether they are one- or two-sided<br>*Only common tests should be described solely by name; describe more complex techniques in the Methods section.* |
| ☐ | ☒ | A description of all covariates tested |
| ☐ | ☒ | A description of any assumptions or corrections, such as tests of normality and adjustment for multiple comparisons |
| ☐ | ☒ | A full description of the statistical parameters including central tendency (e.g. means) or other basic estimates (e.g. regression coefficient) AND variation (e.g. standard deviation) or associated estimates of uncertainty (e.g. confidence intervals) |
| ☐ | ☒ | For null hypothesis testing, the test statistic (e.g. $F$, $t$, $r$) with confidence intervals, effect sizes, degrees of freedom and $P$ value noted<br>*Give P values as exact values whenever suitable.* |
| ☒ | ☐ | For Bayesian analysis, information on the choice of priors and Markov chain Monte Carlo settings |
| ☒ | ☐ | For hierarchical and complex designs, identification of the appropriate level for tests and full reporting of outcomes |
| ☒ | ☐ | Estimates of effect sizes (e.g. Cohen's *d*, Pearson's *r*), indicating how they were calculated |

*Our web collection on statistics for biologists contains articles on many of the points above.*

## Software and code

Policy information about availability of computer code

| Data collection | No software was used for data collection |
|---|---|
| Data analysis | GraphPad Prism 8, IceLogo (https://iomics.ugent.be/icelogoserver/) , FlowJo_v10.8.1, VENNY2.1, Scaffold version 5.2, Proteome Software, MASCOT software (https://proteomicsresource.washington.edu/mascot/cgi/login.pl), Universal Spectrum Explorer (https://www.proteomicsdb.org/use/), Proteome Discoverer (Thermo Fischer), bcl2fastq v2.20.0.422, FastQC v 0.11.8, cutadapt v3.2, USEARCH v. 11.0.667, NetMHCII 2.3, Biorender (https://biorender.com/) |

For manuscripts utilizing custom algorithms or software that are central to the research but not yet described in published literature, software must be made available to editors and reviewers. We strongly encourage code deposition in a community repository (e.g. GitHub). See the Nature Portfolio guidelines for submitting code & software for further information.

## Data

Policy information about availability of data

All manuscripts must include a data availability statement. This statement should provide the following information, where applicable:
- Accession codes, unique identifiers, or web links for publicly available datasets
- A description of any restrictions on data availability
- For clinical datasets or third party data, please ensure that the statement adheres to our policy

The TCRVB sequencing results reported in this paper are publicly available via the immuneACCESS database of Adaptive Biotechnologies: (https://

## Human research participants

Policy information about studies involving human research participants and Sex and Gender in Research.

| | |
|---|---|
| Reporting on sex and gender | Sex and gender were collected upon self-reports. Since our study deeply focuses on understanding the immunogenicity of commensal and pathogenic bacteria antigens on many T cell clones isolated from a patient underwent a neoantigen vaccination therapy, sex and gender was not considered in the study design. |
| Population characteristics | In total, 20 patients (13 males, 7 females) were included. The mean age was 66.75 years (range 46-86). All patients were IDH1 wildtype. These information are available in supplementary tables 1, 2. |
| Recruitment | All included patients underwent elective brain tumor surgery at the Department of Neurosurgery at the University Hospital Zurich and were diagnosed with glioblastoma (IDH1 wildtype) by a board-certified neuropathologist. No other inclusion or exclusion criteria were applied so that we do not expect other biases of any kind. Freshly resected human tissue and blood samples were obtained from the Department of Neurosurgery at the University Hospital Zurich. Written informed consent was obtained from each patient in accordance with the local ethical requirements (KEK-ZH-Nr. 2015-0163). Patient 1635WI was interested in participating in an individual treatment attempt with a personalized peptide vaccination therapy (Wang et al., submitted 2022). He was fully informed about the potential risks of such a vaccination, which  by definition is an individual medical treatment ("compassionate use") and not subject to special regulations for medical research according to the Therapeutic Products Act (TPA) or Swiss Federal Human Research Act (HRA). Consequently, individual medical treatments are not approved by the Ethics Committee or Swiss Agency for Therapeutic Product (Swissmedic), as they do not constitute clinical trials for the purpose of systematically gaining knowledge. The patient gave written informed consent to receive the individualized vaccination. |
| Ethics oversight | Written informed consent was obtained from each patient in accordance with the local ethical requirements, Zurich, Switzerland (KEK-ZH-Nr. 2015-0163). |

Note that full information on the approval of the study protocol must also be provided in the manuscript.

# Field-specific reporting

Please select the one below that is the best fit for your research. If you are not sure, read the appropriate sections before making your selection.

☒ Life sciences ☐ Behavioural & social sciences ☐ Ecological, evolutionary & environmental sciences

For a reference copy of the document with all sections, see nature.com/documents/nr-reporting-summary-flat.pdf

# Life sciences study design

All studies must disclose on these points even when the disclosure is negative.

| | |
|---|---|
| Sample size | Sample size was determined based on the availability of biological materials |
| Data exclusions | No data was excluded. |
| Replication | All experiments including immunopeptidome, bulk sorted T cells in response to vaccine, bacteria/microbiota-derived peptide pools, TCC88's response to ps_SCL were conducted in at least 3 replicates and were all also reproducible. Cytotoxicity assay, HLA-DR expression of tumor cells and staining of the tumor tissue were done in duplicates as we had limited access to certain material. TCCs large screening with bacteria/microbiota-derived peptides were also conducted as duplicates. All experiments were reproducible. |
| Randomization | Randomization is not relevant for this study. The study was not a controlled clinical trial, in which an intervention had been compared to a control (either placebo or verum), but rather used biomaterials (i.e. tumor tissue) from glioblastoma patients, who had received surgery. |
| Blinding | Blinidng is not relevant for this study. The study was not a controlled clinical trial, in which an intervention had been compared to a control (either placebo or verum), but rather used biomaterials (i.e. tumor tissue) from glioblastoma patients, who had received surgery. |

# Reporting for specific materials, systems and methods

We require information from authors about some types of materials, experimental systems and methods used in many studies. Here, indicate whether each material, system or method listed is relevant to your study. If you are not sure if a list item applies to your research, read the appropriate section before selecting a response.

## Materials & experimental systems

| n/a | Involved in the study |
|---|---|
| ☐ | ☒ Antibodies |
| ☐ | ☒ Eukaryotic cell lines |
| ☒ | ☐ Palaeontology and archaeology |
| ☒ | ☐ Animals and other organisms |
| ☒ | ☐ Clinical data |
| ☒ | ☐ Dual use research of concern |

## Methods

| n/a | Involved in the study |
|---|---|
| ☒ | ☐ ChIP-seq |
| ☐ | ☒ Flow cytometry |
| ☒ | ☐ MRI-based neuroimaging |

## Antibodies

Antibodies used

PerCP/Cy5.5 anti-human CD3 antibody, HIT3a clone, Biolegend, cat.300328;
APC-Cy7 anti-human CD4 antibody, OKT4 clone, Biolegend, cat. 317418;
Anti-CD3 monoclonal antibody, OKT3 clone, ThermoFisher, cat. 16-0037-81;
APC anti-human CD4, RPA-T4 clone, Biolegend, cat. 300514;
Pacific blue anti-human CD8, SK1 clone, Biolegend, cat. 344718;
APC anti-human CD45RA, HI100 clone, Biolegend, cat. 304112;
PE-Cy7 anti-human HLA-DR, L243 clone, Biolegend, cat. 307616;
CellTrace™ CFSE Cell Proliferation Kit, for flow cytometry, ThermoFisher, cat. C34554;
CD45RA MicroBeads, human, Miltenyi Biotec, cat. 130-045-901;
ELISA MAX™ Standard Set Human IFN-γ, Biolegend, cat. 430101;
LEGENDplex Human T Helper Cytokine Panels, Biolegend, cat. 740001;
LIVE/DEAD™ Fixable Yellow Dead Cell Stain Kit, for 405 nm excitation, ThermoFisher, cat. L34968;
LIVE/DEAD™ Fixable Aqua Dead Cell Stain Kit, for 405 nm excitation, ThermoFisher, cat. L34957;
Human GM-CSF DuoSet ELISA, rndsystems, cat. DY215;
Human XL Cytokine Luminex® Performance Assay 44-plex Fixed Panel, rndsystems,cat. LKTM014;
LIVE/DEAD Fixable Near-IR Dead Cell Stain Kit, ThermoFisher, cat. L10119;
Recombinant Anti-HLA-DR antibody, clone TAL 1B5, abcam, cat. ab20181;
CD68 Monoclonal Antibody, clone KP1, eBioscience, cat. 14-0688-82;
CD3ε, clone D7A6E, Cell Signaling technology, cat. 85061;
CD31/PECAM-1 Antibody, clone JC/70A, Novus Biologicals, cat. NB600-562;
Recombinant Anti-GFAP antibody, clone EP672Y, abcam, cat. ab220820;
Anti-Human IgG (Fc specific), Sigma-Aldrich, cat. I2136;
TCRBV1-PE, clone BL37.2, Beckman coulter, cat. IM2355;
TCRBV2-PE, clone MPB2D5, Beckman coulter, cat. IM2213;
TCRBV4-PE, clone WJF24, Beckman coulter, cat. IM3602;
TCRBV5.3-PE, clone 3D11, Beckman coulter, cat. IM2002;
TCRBV7.1-PE, clone ZOE, Beckman coulter, cat. IM2287;
TCRBV9-PE, clone FIN9, Beckman coulter, cat. IM2003;
TCRBV12-PE, clone VER2.32.1, Beckman coulter, cat. IM2291;
TCRBV13.1-PE, clone IMMU 222, Beckman coulter, cat. IM2292;
TCRBV14-PE, clone CAS1.1.3, Beckman coulter, cat. IM2047;
TCRBV18-PE, clone BA62.6, Beckman coulter, cat. IM2049;
TCRBV20-PE, clone ELL1.4, Beckman coulter, cat. IM2295;
TCRBV23-PE, clone AF23, Beckman coulter, cat. IM2004;
TCRBV3-FITC, clone CH92, Beckman coulter, cat. IM2372;
TCRBV5.1-FITC, clone IMMU 157, Beckman coulter, cat. IM1552;
TCRBV5.2-FITC, clone 36213, Beckman coulter, cat. IM1482;
TCRBV8-FITC, clone 56C5.2, Beckman coulter, cat. IM1233;
TCRBV11-FITC, clone C21, Beckman coulter, cat. IM1586;
TCRBV13.6-FITC, clone JU74.3, Beckman coulter, cat. IM1330;
TCRBV16-FITC, clone TAMAYA1.2, Beckman coulter, cat. IM1560;
TCRBV17-FITC, clone E17.5F3.15.13, Beckman coulter, cat. IM1234;
TCRBV21.3-FITC, clone IG125, Beckman coulter, cat. IM1483;
TCRBV22-FITC, clone IMMU 546, Beckman coulter, cat. IM1484;

Validation

All antibodies were validated by the manufacturer and data is available at the manufacturer's website as indicated below:
https://www.biolegend.com/it-it/products/percp-cyanine5-5-anti-human-cd3-antibody-5613
https://www.biolegend.com/fr-fr/products/apc-cyanine7-anti-human-cd4-antibody-3658
https://www.thermofisher.com/antibody/product/CD3-Antibody-clone-OKT3-Monoclonal/16-0037-81
https://www.biolegend.com/en-us/products/apc-anti-human-cd4-antibody-823?GroupID=BLG7755
https://www.biolegend.com/fr-ch/products/pacific-blue-anti-human-cd8-antibody-6509
https://www.biolegend.com/ja-jp/products/apc-anti-human-cd45ra-antibody-684
https://www.biolegend.com/nl-be/products/pe-cyanine7-anti-human-hla-dr-antibody-2862
https://www.thermofisher.com/order/catalog/product/C34554
https://www.miltenyibiotec.com/CH-en/products/cd45ra-microbeads-human.html#130-045-901
https://www.biolegend.com/fr-ch/products/human-ifn-gamma-elisa-max-standard-2226

# Eukaryotic cell lines

Policy information about cell lines and Sex and Gender in Research

| | |
|---|---|
| Cell line source(s) | BLS cells expressing HLA-DRA1*01:01 and BLS-DRB1*03:01 were kindly gifted from William W. Kwok, Benaroya institute BLS-DRB1*04:02, BLS-DRB3*02:02 and BLS-DRB4*01:01 were created in-house. All BLS cell lines in this study were originally derived from a female. |
| Authentication | Gifted cell lines were authenticated by the suppliers. BLS cells were subjected to antibiotic selection in culture to deplete cells that have lost their HLA-DR expression. BLS cells expressing different HLA-DRB molecules were then authenticated using HLA-DR antibody for expression of HLA-II molecules. All BLS cells were expressing HLA-DR molecule. No further authentication has been conducted. |
| Mycoplasma contamination | All cells were regularly tested for mycoplasma contamination and were negative. |
| Commonly misidentified lines (See ICLAC register) | No commonly misidentified lines were used in this study |

# Flow Cytometry

## Plots

Confirm that:

☒ The axis labels state the marker and fluorochrome used (e.g. CD4-FITC).

☒ The axis scales are clearly visible. Include numbers along axes only for bottom left plot of group (a 'group' is an analysis of identical markers).

☒ All plots are contour plots with outliers or pseudocolor plots.

☒ A numerical value for number of cells or percentage (with statistics) is provided.

## Methodology

| | |
|---|---|
| Sample preparation | CFSE-labeled bulk T cells were seeded with irradiated PBMCs loaded with peptides. After incubation, cells were washed and stained with live-dead dye and human IgG for 30 minutes. Next, cells were washed and stained with the detection antibodies for 30 minutes in 4 degree. Finally, cells were washed and used for FACS or sorting. |

| Instrument | LSR Fortessa Flow Cytometer (BD Biosciences) was used to measure cytokines and sorting was conducted using a SH800S Cell Sorter (Sony). |
|---|---|
| Software | Data were analyzed using FlowJo (Tree Star). |
| Cell population abundance | Known cell numbers based on the counted and seeded cells in each well were used for TILs, CD45RA- PBMCs and T cell clones |
| Gating strategy | Cells were always gated for live, proliferating (CFSEdim) and further gated based on the staining |

☒ Tick this box to confirm that a figure exemplifying the gating strategy is provided in the Supplementary Information.

