## [Peer Review File · Nature]

Manuscript Title: Microbial Peptides Activate Tumor-Infiltrating Lymphocytes in Glioblastoma

Reviewer Comments & Author Rebuttals

Reviewer Reports on the Initial Version:

Referees' comments:

Referee #1 (Remarks to the Author):

The authors analyzed the tumor immunopeptidome of 19 glioblastoma (GB) patients for the presence of HLA-DR-associated bacterial peptides. Next, they scrutinized the capacity of TILs and TIL derived CD4+ T cell clones in one of the patients for reactivity against HLA-bound bacterial peptides, tumor antigens, viruses, bacteria, and gut commensals. For their demonstration, the authors used a scoring matrix that they developed earlier, and several large databases including all human proteins, the translated transcriptome of the primary and recurrent autologous glioblastoma as well as all viruses, which are annotated for infecting humans as well as UniProt bacteria and human gut microbiota.

TCC88 clone directed against a GB neoantigen, not only recognized multiple other GB-derived peptides, but also numerous bacterial and gut commensals-derived targets with a very robust avidity compared with GB peptides. 57.7% of all stimulatory peptides were derived from bacteria and gut microbiota, and 29.3% from TAAs derived from the autologous glioblastoma. The TIL-derived, SIN3A*-specific TCC88 recognized 21 bacterial and 42 gut microbiota-derived peptides while 34 of these were better recognized at a single antigen concentration than the mutated tumor peptide (with stronger secretion of CXCL10 or IL-10). TCC88 strongly cross-recognizes bacteria/microbiota-derived peptides, mainly derived from Firmicutes, Proteobacteria, Bacteroidota phyla. The reactivity against tumor (LAMA5, LRP6, SIN3A*)- and bacterial antigens is shared by T cell clones but also bulk TILs. When they compared TCR sequences of the CFSEdim CD4+ T cells responding to the tumor vaccine pool with those responding to the bacteria/microbiota pool, CD4+ shared 3.7% of the TCRs indicating. The fraction of TCRs that are found in both expanded T cell fractions represented up to 1% of the entire TILs. Moreover, peripheral blood CD4+ T cells that had been activated with bacteria/microbiota- or IPdBP recognized the same tumor associated -antigens by secreting proinflammatory and Th1 cytokines and chemokines. T cells responding to bacterial antigens, which are presented in the tumor, can be found in the peripheral immune compartment. The authors compared the TCR sequences of these two populations and demonstrated that this cross-reactivity encompassed 3.4% of the TCR sequences.

These findings suggest that commensal-based tumor vaccination may be a very promising approach to circumvent peripheral tolerance to self -antigens, the very limitation of current vaccination strategies.

Major comments

1/ Visualization of intratumoral bacteria (GB) would be of great impact in GB: PCR, culturomics and electron microscopy are doable and can be done on frozen or fresh GB, as well as in IHC for LPS and

LTA (although anti-LTA staining may not be that reliable). FISH with specific probe sets would also be an added value.

2/ TCC characterization or TILs as for gut- derived hallmarks: CCR9, a4b7, IL-17 or Foxp3 or any Tr17 fingerprint? Despite the tremendous work performed by the authors, single cell transcriptomics may enable tracing or fate mapping of these bacteria-specific T cells primed in the GALT. Can the authors find whether cross-reactivity comes from TAA-primary recognition or the other way around, using the gut molecular fingerprint of these TCCs?

3/ Relevance for patient prognosis: I admit this is a prospective and difficult question. Since the satellite paper is summarizing the results of a vaccine clinical trial, can the authors infer from the time to progression in all these patients, whether bacteria-versus neoantigen -specific T cell responses (or cross-reactivity between both) is of prognosis value?

Minor comments

Line 168: Enterobacterales, and Campylobacterales are “bacterial orders”, not “families” as stated.

Referee #2 (Remarks to the Author):

In this manuscript the authors show that bacteria-specific peptides are presented on glioblastoma cells and recognized by glioma-infiltrating lymphocytes and peripheral blood-derived memory CD4+ T cells, which cross-react with several tumor antigens. The authors further show that a glioma-infiltrating CD4+ cell clone recognizes a broad spectrum of peptides from pathogenic bacteria, commensal gut microbiota and also glioblastoma-related tumor antigens. The authors conclude that bacterial pathogens and bacterial gut microbiota can be involved in specific immune recognition of tumor antigens.

Intratumoral bacterial antigens have been described in several tumor entities including gliomas. In melanoma bacterial peptides have been shown to be presented on HLA-DR. It is also known that pathogen-specific T cell may cross react with tumor antigens. Using immune peptidomes of 19 glioblastoma samples and 6 derived cell lines the authors annotate 5-56 bacterial but not viral peptides per sample. The authors identify one neoantigen-specific glioma-infiltrating T cell clone for further unbiased testing and showed HLA-DR-restricted reactivity to multiple peptides from different sources including the autologous tumor transcriptome, the human proteome, viruses, bacteria as well as gut microbiota, indicating high promiscuity / cross reactivity of this T cell clone.

This is an interesting manuscript confirming previous observations in glioblastoma and other types of tumors. There are several points / unanswered questions that dampen my enthusiasm for this manuscript.

1. The authors describe the response of bulk TILs to glioma-derived bacterial peptidome as weak. In fact, I only see 2 out of 40 peptidomes induce a proliferative response over background. Is this

significant? In any case, this questions the relevance of bacterial peptidomes un shaping the anti-glioma T cell response. In Fig. 1e the positive control is lacking.

2. After stimulation with a glioma-derived neoepitope tha authors isolated T cell clones, only two out of eight TCC responded to one and several IPdBPs, respectively. The heatmap does not allow to determine the strength of activation compared with the bulk T cells and the positive control.

3. The unbiased screening approach identified several bacterial peptides that induce activation of 3 TCCs and bulk TILs. It is, however, not clear, whether these peptides are also contained in the initial 40 peptidomes and if so in which.

4. The authors infer from activation and sorting of T cells and a small fraction of overlapping TCRs that these must be cross-reactive. However, it is not shown that the TCRs are indeed cross-reactive. To this end TCRs should be cloned an tested individually for differential affinity and avidity of tumor-specific versus bacterial peptides.

5. A major question is whether the reactivity of peripheral T cells to both, tumor-derived and bacterial peptides is specific for the tumor disease. Analysis of healthy donors are lacking. Also, it is not clear whether the bacterial peptides are also present / presented in normal brain and / or CNS tissue with a disrupted blood-brain barrier.

6. The authors infer from the fact that the immunopeptidome of tumor-derived cell lines also contains bacterial peptidomes that the bacteria must be intracellular. However, this is not shown. It is unclear whether the intracellular presence of these bacteria result in processing and presenting of the identified epitopes.

Referee #3 (Remarks to the Author):

Naghavian and colleagues set out to assess whether CD4+tumor-infiltrating lymphocytes (TILs) recognize intratumorally bacterial pathogens and gut microbiota in glioblastoma. This work is based on the premise that a subset of the intra-tumoral bacteria can invade eukaryotic cells, be processed and presented on HLA-II molecules impacting the immune response. Using an antigen discovery approach, the authors demonstrate cross-reactivity of a TIL derived T cell clone recognizing a broad spectrum of peptides including tumor neoantigen and bacterial peptides, and further show that these bacterial peptides are able to stimulate bulk TILs and neoantigen-derived T cell clones, showing the functional relevance of the tumor microenvironment.

The concepts underlying this manuscript are of high interest to both immunologists and proteomic scientists; however, overall the data presented by the authors do not sufficiently support their conclusions. Further clarifications/additions should be made as described below.

1. In order to identify bacterial peptides, the authors perform HLA peptidomics on 19 glioblastoma samples and tumor cell lines using bacterial proteomes from UniprotKB/Swiss prot. The use of such big databases for searching the peptides can affect the validity of the findings. Also, as it includes not only the proteomes of the bacteria that were actually identified in the tumor samples, it can result in

many false positives. The authors should definitively identify the bacteria residing in each patients' tumor in order to analyze their MS data with it, so they could claim that these are peptides presented by the patient's tumors. Identification of the species in each patient by 16s sequencing will enable a patient-specific analysis, this method is extensively used in microbiome research and fundamentally required here to prove their identifications.

2. The authors claim that the identified peptides are derived from tumor cells, but the authors have not ruled out the possibility that APCs in the tumor microenvironment could take up the pathogen antigens and present these peptides on their HLA-II molecules. The authors should experimentally demonstrate that the infected tumor cells are the only source of these pathogen-derived antigens presented in the tumor microenvironment or alternatively change their claims. This is an important distinction since the antigen presentation pathway between phagocytosing APCs would be different than tumor cells, and therefore there will be differences in the presented antigens. Altering the language would help clarify this, additionally to clarify this possibility, an experimental suggestion would be to isolate the APCs from tumor cells and perform HLA-II peptidomics on both populations, emphasizing the presence of tumor pathogen peptides.

3. If the authors suggest that the bacterial HLA-II peptides are presented by the tumor cells, they should provide evidence showing the glioblastoma tumors express HLA-II, as not all tumor cells do.

4. There are clarifications that need to be made and missing information in the materials and methods section which are fundamental to the understanding of the experiments performed. Specifically, the information regarding the growth of the tumor-derived cell lines used for HLA-II peptidomics, as well as the number of replicates. Most likely bacterial species will not survive within tumor cells cultured for a number of passages. Additionally, in the immunopeptidome analyses section, there is missing information regarding the peptide search, was the MS data searched with both human and pathogen proteomes together or only with the pathogen database? What was the FDR? And how did the authors control for the FDR analysis when using such a big dataset?. Removing redundant sequences from the pathogen database can help with this. Were there any filtrations steps of the identified peptides or quality controls for the data, such as the removal of peptides which sequence is identical between the pathogen and human peptides, ambiguous I/L identifications, peptides that might result from non-coding regions and pseudogenes?. Such steps will ensure that these peptides are indeed derived from pathogens and not the result of other human sources or ambiguity. Additionally, the list of pathogen species used for the HLA peptidomics analysis is missing.

5. The authors should validate the peptides identified by HLA peptidomics by comparing their MS/MS spectra fragmentation to that of synthetic peptides, used as standard validation for peptide identification. Additionally, the authors present in Supplementary Table 5 the MS spectra only from patient 1635WI, the bacterial peptide spectra from the other patients are missing. All RAW data as the authors mentioned in the material and method section, as well as proteome databases and search results should be made publically available by depositing them in public repositories.

6. Do the authors see these bacterial antigens in any previously published HLA-peptide datasets?

7. The authors should remove control bacterial species, bacterial species that were shown to be found in normal tissue, thus would not expect to be found in the tumor.

8. For recurrent peptides between primary and recurrent tumor in patient 1635WI, authors should mark these peptides in supplementary table 4, additionally they should describe if their origin is from an identical bacterial species, and if there were recurrent peptides that originated from a number of different species.

9. In Extended data Figure 1, did the authors test the TCC88 proliferation after incubation with the SIN3A unmutated peptide? this should be added to the figure as a comparison.

10. In lines 224-233 the authors describe the application of an antigen discovery methodology for targeted peptide recognition, have the peptides chosen by the authors to be tested in the proliferation assay of TCC88, been identified as binders to the 1635WI alleles using prediction servers such as NetMHCIIpan? Are they strong binders? Could the authors provide the readers with a comparison between the two?

11. In lines 256-262 the authors state that 57.7% of all stimulatory peptides, predicted with their stimulatory scores and tested with clone TCC88 were derived from bacteria and gut microbiota. However, most of the peptides tested by the authors were derived from these types making it hard to rule out autoreactivity, specifically when only 4 peptides from the human proteome were tested—a very unbalanced comparison. It is surprising that this clone tested with the predicted bacterial peptides did not respond to the immunopeptidome-derived bacterial peptides (Fig. 2d), how do the authors explain this?

12. In extended data fig. 1e, the authors test 135 peptides from brain and myelin autoantigens. In order to rule out auto reactivity, specifically, when suggesting that such clones could be potentially used for adoptive cell therapy a wider and a more in-depth study should be performed. The test of HLA presented peptides from non malignant tissues, should be used, for example, those that appear in the HLA ligand atlas Marcu et al. 2020.

13. In Figure 3d-f, the authors were not consistent with the peptides chosen for activation assays of TCC88 and have not provided an explanation in the text, why was only HGM3 chosen for the killing assay in Fig. 3f and this was not confirmed with the additional DRB3 microbiota peptides HGM62 and HGM27 in which there functional avidity is similar in figure 3d?

14. In line 300 it is not clear why TCC68 and TCC75 were chosen for bacterial peptide reactivity?, Based on Fig. 1b there were a lot of other TCC that sowed high SIN3A* antigen activation additionally it raises the question why were these TCC not reacting to the HLA-II presented bacterial peptides in figure 2d.

15. Changing the presentation of the TCC proliferation graphs in to a heatmap with an intensity-colored scale will clarify figure 4, and combining Fig. 4b and c, would make this figure clearer.

Additional comments:

- The excel supplemental table numbers are missing, making it hard to find supplementary tables. There is no table of contents in these files or legend.

- A comprehensive comparison of the bacteria phylum previously associated with response to immunotherapy treatment, such as: Gopalakrishnan., 2018, Matson., 2018, Routy, 2018, should be added and should also be discussed In the discussion section.

- What are the limitations to this study? Can the authors add to discussion?

Author Rebuttals to Initial Comments:

Point-by-point replies to the reviewers' comments:

Referee #1 (Remarks to the Author):

"The authors analyzed the tumor immunopeptidome of 19 glioblastoma (GB) patients for the presence of HLA-DR-associated bacterial peptides. Next, they scrutinized the capacity of TILs and TIL derived CD4⁺ T cell clones in one of the patients for reactivity against HLA-bound bacterial peptides, tumor antigens, viruses, bacteria, and gut commensals. For their demonstration, the authors used a scoring matrix that they developed earlier, and several large databases including all human proteins, the translated transcriptome of the primary and recurrent autologous glioblastoma as well as all viruses, which are annotated for infecting humans as well as UniProt bacteria and human gut microbiota.

TCC88 clone directed against a GB neoantigen, not only recognized multiple other GB-derived peptides, but also numerous bacterial and gut commensals-derived targets with a very robust avidity compared with GB peptides. 57.7% of all stimulatory peptides were derived from bacteria and gut microbiota, and 29.3% from TAAs derived from the autologous glioblastoma. The TIL-derived, SIN3A*-specific TCC88 recognized 21 bacterial and 42 gut microbiota-derived peptides while 34 of these were better recognized at a single antigen concentration than the mutated tumor peptide (with stronger secretion of CXCL10 or IL-10). TCC88 strongly cross-recognizes bacteria/microbiota-derived peptides, mainly derived from Firmicutes, Proteobacteria, Bacteroidota phyla. The reactivity against tumor (LAMA5, LRP6, SIN3A*)- and bacterial antigens is shared by T cell clones but also bulk TILs. When they compared TCR sequences of the CFSEdim CD4⁺ T cells responding to the tumor vaccine pool with those responding to the bacteria/microbiota pool, CD4⁺ shared 3.7% of the TCRs indicating. The fraction of TCRs that are found in both expanded T cell fractions represented up to 1% of the entire TILs. Moreover, peripheral blood CD4⁺ T cells that had been activated with bacteria/microbiota- or IPdBPs recognized the same tumor associated -antigens by secreting proinflammatory and Th1 cytokines and chemokines. T cells responding to bacterial antigens, which are presented in the tumor, can be found in the peripheral immune compartment. The authors compared the TCR sequences of these two populations and demonstrated that this cross-reactivity encompassed 3.4% of the TCR sequences.

These findings suggest that commensal-based tumor vaccination may be a very promising approach to circumvent peripheral tolerance to self -antigens, the very limitation of current vaccination strategies."

Major comments

"1/ Visualization of intratumoral bacteria (GB) would be of great impact in GB: PCR, culturomics and electron microscopy are doable and can be done on frozen or fresh GB, as well as in IHC for LPS and LTA (although anti-LTA staining may not be that reliable). FISH with specific probe sets would also be an added value."

Authors' reply: We agree that the suggested analyses would be of great importance to detect bacteria in the tumor. To address this point, we have considered several of the recommended methods.

We discussed the comments of the reviewer with several experts and attempted to perform some of the suggested assays, and the following points emerged from these. We conducted 16S rRNA FISH for bacteria on the FFPE tumor tissue slides. We found microorganisms (morphologically concordant with aerobic spore formers) outside the tissue in the first three samples that we deparaffinated and FISHed. We confirmed this finding in a fourth section that we did not deparaffinate, but directly investigated with DAPI mounting medium only. Again, we found microorganisms in the paraffin outside the tissue. To us, this indicates a sample contamination during paraffination/sectioning on the slides, and we therefore did not pursue this line of examination further.

Next, we performed 16S rRNA sequencing from 10 tumors to assess if not only bacterial/microbial peptides were demonstrable in the immunopeptidomes of glioblastoma, but bacterial DNA could also be shown. This was indeed the case for all tumors. One important point that became clear from the sequencing studies is that the abundance of bacterial sequences is very low. We examined fresh frozen tissues of 10 GBM patients, from whom we analyzed the immunopeptidome including ZH616, ZH645, ZH654, ZH681, ZH753, ZH757, ZH761, ZH791, ZH802 and 1635WI. Moreover, non-template controls in empty tubes were included as controls and treated and processed like the samples. The latter would amplify any contamination from handling the tissue and thus identify contaminations. As already mentioned above, the bacterial load in the GBM tumor samples was very low. Nevertheless, we were able to detect bacteria in the tumor samples of all 10 patients (**Extended Data Fig. 1b** and **Supplementary Tables 5 and 6**).

The 16S rRNA sequencing-derived results were then compared with bacterial sources of the IPdBPs in each patient. We found at least 1 exact species match in 16S rRNA and immunopeptidomes in 6 out of 10 patients (**Supplementary Table 7**). Since, the resolution to identify bacteria down to the species level is limited due to high similarities of 16S rRNA sequences between certain bacteria¹, we annotated the data one level higher, i.e. at the phylo-type level, Genus, and found matches in 8 patients. These data suggest the presence of both bacterial DNA as well as processed and presented peptides in GBM tumors (**Supplementary Table 7**). Since the presence of bacterial DNA only indicates that the tumor contained bacterial DNA, but does not provide information as to which peptide from a bacterial protein would be processed and loaded onto autologous HLA-DR molecules, it is not too surprising that the two sources do not fully match. Furthermore, it can indicate heterogeneity of bacteria in different parts of the tumor², which were used for 16S rRNA sequencing and immunopeptidome analyses. Important to note, the relative abundance of bacterial peptides in the HLA class II immunopeptidomes was also low. This may be due to several reasons including the low burden of bacteria in the tumor tissue³ and/or on tumor-infiltrating APCs with the result that only a minority of HLA class II molecules are occupied by bacteria-derived peptides and furthermore because the sensitivity for detection with the LC-MS/MS-based proteomics method is limited.

Regarding the presence of bacteria in the tumor, we would like to emphasize that the number of IPdBPs in the tumor yielded only a fraction of peptide sequences compared to the annotation for human- and tumor-specific proteins (23 for bacterial and 4464 for human and tumor proteome HLA-II peptides from the recurrent tumor of patient 1635WI). This is in line with the abovementioned low abundance of bacterial DNA and also indicates that methodologies of bacteria-specific FISH and/or electron microscopy-based detection will be difficult to establish. Furthermore, although the identification of IPdBPs is in principle very promising, they were only (very) weakly stimulatory for TILs, when proliferation was used as a readout, slightly more stimulatory based on cytokine detection, and, in addition, two of the isolated TCCs (**Fig. 1d and 2d**) more robustly recognized two and multiple IPdBPs.

The 16S rRNA sequencing together with the immunopeptidome analyses and testing of these peptides indicated that bacteria are indeed in the tumor, processed and presented in the context of autologous HLA-class II molecules. Further and probably stronger evidence for a role of bacterial/microbial antigens (as target antigens, but not for their presence in the tumor) came from the unbiased identification of target epitopes of TIL-derived T cell clones (TCC) using the combination of positional scanning combinatorial peptide libraries (ps-SCL) and dedicated bioinformatics to query very large sequence databases. With this approach, we identified completely different bacterial peptides than by immunopeptidomics, and, very surprisingly, the majority of targets for TCC88 were derived from infectious and commensal bacteria (**Supplementary Table 15, Extended Data Fig. 7c**). These peptides stimulated TCCs, TILs and PBMCs of the patient much more efficiently showing at multiple levels that T cell cross-reactivity to strong foreign antigens exists and may be exploited for tumor therapies in the future on activating T cells using bacterial peptides other than those found in the immunopeptidome.

Extended Data Fig. 7c, Comparison of bacterial source species of the peptides found via HLA-II immunopeptidome analysis with those found by the unbiased target search for TCC88 using ps-SCL screening. Most of the bacterial species were either found only in the immunopeptidomes (blue) or unbiased target search (red). The adjacent double bars of blue and red for *Bacillus sp.* and *Bacteroides ovatus* indicate that peptides from these two species were found via both approaches.

Extended Data Fig. 1b, 16S rRNA gene sequencing of tumor tissues of 10 GBM patients, from whom immunopeptidome analyses were available, confirmed the presence of bacterial DNA in tumors. The relative abundance of operational taxonomic units (OTUs) in read counts is shown at the phylum level for each tumor (Supplementary Table 5, 6). * 16S rRNA data from patient 1635WI includes both primary and recurrent tumors in triplicates. The relative abundance of OTUs was calculated based on mean OTU of the triplicates divided by total OTUs in each tumor.

The above data from 16S rRNA sequencing data, its comparison with IPdBPs has been added to the body of the paper and the method section, and the data been discussed.

"2/ TCC characterization or TILs as for gut-derived hallmarks: CCR9, a4b7, IL-17 or Foxp3 or any Tr17 fingerprint? Despite the tremendous work performed by the authors, single cell transcriptomics may enable tracing or fate mapping of these bacteria-specific T cells primed in the GALT. Can the authors find whether cross-reactivity comes from TAA-primary recognition or the other way around, using the gut molecular fingerprint of these TCCs?"

Reply: This is a very interesting and important point. We looked for expression of CCR9, a4b7 and FOXP3 in 8 TCCs as well as the short-term expanded TILs since they were the source of the

Figure R1, FACS analysis of TILs and 8 TCCs including TCC57, TCC64, TCC68, TCC72, TCC75, TCC88, TCC-2D3 and TCC-8F7 for expression of FOXP3, a4b7 and CCR9 markers. Lymphocytes were gated for singlets, CD3+CD4+ T cells. Subsequently, FOXP3+ and FOXP3- cells were gated and histogram plots of a4b7 and CCR9 were generated. Antibody-stained beads were used as positive controls for a4b7 and CCR9.

isolated TCCs (**Figure R1**; please, note that a table of content of all materials that are only included in the response to the reviewers is included separately) (These data are not included in the manuscript). Based on previous phenotyping of TCCs, we knew that the majority show a mixed Th1&Th2 phenotype. We here now also confirm that the TCCs are CD3+CD4+FOXP3-CCR9-a4b7-. Only a small proportion of CD3+CD4+ expanded TILs are FOXP3+ and a4b7+, but all are CCR9-.

We further analyzed the RNA-seq data of the tumor tissue and found that a4b7 was expressed at normal and CCR9 at very low levels (white square mean normal expression of the gene in the tumor compared to the TCGA database of GBM cohort and blue diamond represents lower expression) (**Figure R2a and Supplementary Table 14**). We have also conducted single cell RNA-seq with the unexpanded TILs (patient 1635WI) and were not able to detect CCR9, a4b7 and

Figure R2, Analysis for gene expression of gut homing markers in tumor tissue and isolated TILs. a, Tumor RNA-seq analysis (1635WI) represents normal expression of ITGA4 and ITGB7 gene expression in the tumor tissue while displaying a low expression of CCR9 gene. b, Single cell RNA-seq analysis on 5000 unexpanded (unmanipulated) TILs signifies undetectable expression of gut and colon homing genes on CD3⁺ cells.

GPR15 (colon homing marker) gene expression (**Figure R2b**) (not included in the manuscript). These data indicate that, although the tissue-infiltrating T cells could have been activated in the gut and derive from this site, probably only a minority if any come from there. T cell activation could have occurred in the peripheral immune system or tissue sites whenever bacteria are transiently available.

"3/ Relevance for patient prognosis: I admit this is a prospective and difficult question. Since the satellite paper is summarizing the results of a vaccine clinical trial, can the authors infer from the time to progression in all these patients, whether bacteria-versus neoantigen-specific T cell responses (or cross-reactivity between both) is of prognosis value?"

Authors' reply: This is indeed an important question, particularly with respect to future developments of possible therapies. To address this question, we should have examined the T cell reactivity to tumor antigens and/or foreign antigens from several patients to be able to conclude the

relevance for patient prognosis. As we now also mention as a limitation at the end of the discussion, we performed the unbiased target discovery for TIL-derived TCCs and their cross-reactivity against bacteria and tumor antigens in detail only in one patient. However, we identified TIL reactivity against some of the bacteria/microbiota-derived peptide targets from patient 1635WI already in TILs from another glioblastoma patient (see below in Fig. R4 in reply to reviewer 2; data not included in the manuscript). As a preliminary step to address the reviewer's point, we compared survival and progression of the patients with the bacterial load in the tumor. These data did not show a correlation between the number of bacterial peptides (IPdBPs) and/or bacterial species (16s rRNA) with the survival of the patients. Moreover, the composition of bacteria in the tumor (16s rRNA) does also not show a discernable pattern with the patient's survival (**Figure R3**). This point should be addressed in larger patient numbers and systematically in the future. Because of the preliminary nature of the data in Fig. R3, we did not include it in the manuscript.

Figure R3, Side by side comparison of patients' survival/progression with the number of bacterial peptides as well as bacterial composition in the tumor.

Minor comments

"Line 168: Enterobacterales, and Campylobacterales are "bacterial orders", not "families" as stated."

Authors' reply: We thank the reviewer for noting this and have corrected the names accordingly.

Referee #2 (Remarks to the Author):

"In this manuscript the authors show that bacteria-specific peptides are presented on glioblastoma cells and recognized by glioma-infiltrating lymphocytes and peripheral blood-derived memory CD4⁺ T cells, which cross-react with several tumor antigens. They further show that a glioma-infiltrating CD4⁺ cell clone recognizes a broad spectrum of peptides from pathogenic bacteria, commensal gut microbiota and also glioblastoma-related tumor antigens. The authors conclude that bacterial pathogens and bacterial gut microbiota can be involved in specific immune recognition of tumor antigens.

Intratumoral bacterial antigens have been described in several tumor entities including gliomas. In melanoma bacterial peptides have been shown to be presented on HLA-DR. It is also known that pathogen-specific T cell may cross react with tumor antigens. Using immune peptidomes of 19 glioblastoma samples and 6 derived cell lines the authors annotate 5-56 bacterial but not viral peptides per sample. The authors identify one neoantigen-specific glioma-infiltrating T cell clone for further unbiased testing and showed HLA-DR-restricted reactivity to multiple peptides from different sources including the autologous tumor transcriptome, the human proteome, viruses, bacteria as well as gut microbiota, indicating high promiscuity / cross reactivity of this T cell clone.

This is an interesting manuscript confirming previous observations in glioblastoma and other types of tumors. There are several points / unanswered questions that dampen my enthusiasm for this manuscript."

"1. The authors describe the response of bulk TILs to glioma-derived bacterial peptidome as weak. In fact, I only see 2 out of 40 peptidomes induce a proliferative response over background. Is this significant? In any case, this questions the relevance of bacterial peptidomes in shaping the anti-glioma T cell response. In Fig. 1e the positive control is lacking."

Authors' reply: We thank the reviewer for the overall positive comments on the study. Regarding the reactivity against the immunopeptidome-derived peptides, we agree that the responses are very weak when proliferation is used as a readout and only slightly higher when cytokine secretion is measured. We have stated the weak responsiveness of the TILs to IPdBPs more clearly in the revised manuscript and mention that IPdBPs are unlikely to have shaped the intratumoral T cell response. Throughout the manuscript, we have now tried to emphasize better that the unbiased target search using the ps-SCL methodology was more efficient in identifying targets for TILs and also peripheral blood memory T cells. Furthermore, the latter method and testing a TIL-derived TCC identified more and also *different* bacterial peptides from infectious and commensal bacteria (**Extended Data Fig. 7c, Supplementary Table 15**), which were stimulatory for the TILs, TCCs and PBMCs.

To assess whether the bacterial/microbial peptides that were recognized by TCC88 and TILs from patient 1635WI were only stimulatory for T cells of this patient or also for TILs from other glioblastoma patients, we tested them with TILs from 3 additional patients and used IFN- γ Fluorospot testing as a readout. 2×10^4 TILs were co-cultured with 5×10^4 autologous irradiated, EBV-transformed B cell line cells (300 Gy) as antigen presenting cells. The final concentration of each peptide was 10 μ M, and cells were incubated for 44 hours at 37 °C. Interestingly, TILs isolated from 1 of the patients (TIL40) were activated by several of the peptides from infectious and gut commensal bacteria and secreted IFN- γ . Δ SFU (spot forming units) were calculated by subtracting the mean number of negative control spots from the spots in each well (**Figure R4**) (These data are not included in the manuscript).

Figure R4. IFN- γ Fluorospot testing of expanded TILs from 3 GBM patients. 64 bacteria/microbiota peptides that were stimulatory for TILs from patient 1635WI were tested with TIL27, TIL35 and TIL40 samples. TILs were stimulated with bacteria/microbiota-derived peptides and irradiated autologous BLCLs in triplicates. Δ SFU (spot forming units) were calculated by subtracting the mean number of negative control spots from the spots in each well. Positive response of TIL40 to several bacterial peptides are shown on the right panel. CD2-CD3-CD28 beads are used as positive controls.

These data show that, apart from the potential relevance of IPdBPs for anti-tumor immunity, strong foreign antigens can be found via the unbiased antigen discovery approach which starts from tumor-infiltrating T cells and cross-recognize bacterial/microbial as well as tumor antigens.

The adapted Fig. 1e including positive control (SIN3A* peptide) has been added (see below).

Fig 1e, Heatmap of cytokine/chemokine secretion of bulk TILs stimulated with three IPdBPs, which weakly stimulated proliferation. The secretion of each cytokine/chemokine (pg/ml) was calculated by subtracting the corresponding cytokine/chemokine amount (pg/ml) of the negative control. SIN3A* = positive control peptide. Co-culture of TILs with APCs without stimulus was used as negative control.

"2. After stimulation with a glioma-derived neoepitope that authors isolated T cell clones, only two out of eight TCC responded to one and several IPdBPs, respectively. The heatmap does not allow to determine the strength of activation compared with the bulk T cells and the positive control."

Authors' reply: Fig. 2d has been adapted, and the stimulatory indices (SI) of stimulatory peptides been added (see below). The comparison of the stimulatory strength of IPdBPs and positive control (SIN3A* peptide) for TCCs and bulk TILs shows that the strength of response of a single TCC such as TCC-2D3 is higher than the one of bulk TILs. This is expected since the TCC is a single uniform cell population, while the bulk TILs contain many specificities, and most of these are not directed against the antigens that were tested. While the response of TCC-2D3 to many IPdBPs

is not very strong with SIs just above 2 (Fig. 2d), several peptides are recognized with SIs >5 including 'Peptidome.Bac 5', which also weakly stimulates bulk TILs (Fig. 1d).

Fig. 2d, Response of tumor specific TCCs to IPdBPs

Fig. 1d, TILs weakly recognize IPdBPs (1635WI)

"3. The unbiased screening approach identified several bacterial peptides that induce activation of 3 TCCs and bulk TILs. It is, however, not clear, whether these peptides are also contained in the initial 40 peptidomes and if so in which."

Authors' reply: This is an important point, which we apparently did not depict sufficiently clearly in the manuscript. We have now tried to describe it better. The bacterial peptides found via the unbiased target discovery approach did not appear in the immunopeptidome-derived peptides, i.e. they are completely different bacterial peptides. By using the unbiased target discovery approach and starting from tumor-infiltrating T cell clones rather than from what is presented on tumor cells and/or APCs, we identified foreign agent-derived peptides besides the IPdBPs. We now present both sets of peptides side-by-side in **Supplementary Table 15**.

"4. The authors infer from activation and sorting of T cells and a small fraction of overlapping TCRs that these must be cross-reactive. However, it is not shown that the TCRs are indeed cross-reactive. To this end TCRs should be cloned and tested individually for differential affinity and avidity of tumor-specific versus bacterial peptides."

Authors' reply: The reviewer is correct that TCR overlap and demonstrating recognition of both bacterial/microbial- and tumor peptides does not allow to infer cross-reactivity, but gives at best a hint that cross-reactive T cells may be contained in the sorted bulk cell population. Proof of cross-reactivity can only be shown for the T cell clones, i.e. TCCs in fig. 3b and fig. 4b for TCC88 and 2 other TCCs that had been isolated from the TILs (an overview of the responses is shown in fig. 4c). These TCCs cross-recognize the tumor-specific peptide (SIN3A*) as well as bacterial peptides. In fig. 3d, we compared the affinity of TCC88 to bacterial peptides as well as SIN3A* peptide showing that they have very similar half-maximal stimulatory concentrations (EC50). After establishing cross-reactive TCCs from the TILs, we compared the TCRs of bulk TILs stimulated with a pool of bacterial peptides and separately with a pool of tumor-derived peptides (fig. 5d). There is a small fraction of TCRs that overlap between the two groups. Also, we found TCRs of TCC75 and TCC88 in the overlapping fraction, which, in this case, confirms that truly cross-reactive T cells are contained in both bulk populations after stimulation with either bacterial- or tumor-derived peptides.

The reviewer is also correct that another way to provide proof for cross-reactive TCRs would be to isolate a representative number of paired TCR α/β chain combinations and use these to transduce a suitable hybridoma cell line for subsequent antigen specificity testing. We are working on these techniques, but are far from having them established. However, we hope that the above demonstration of cross-reactive TCC, although we have only identified few of these, provides sufficient evidence for cross-reactivity between bacterial/microbial- and tumor antigens at the clonal level.

"5. A major question is whether the reactivity of peripheral T cells to both, tumor-derived and bacterial peptides is specific for the tumor disease. Analysis of healthy donors are lacking. Also, it is not clear whether the bacterial peptides are also present / presented in normal brain and / or CNS tissue with a disrupted blood-brain barrier."

Authors' reply: These are again important suggestions. In order to address these points, we have

Extended Data Fig. 1a, HLA-II immunopeptidome analyses for bacterial peptides in control cohort including brain lesions of 3 MS patients and 6 healthy donors. 55 bacterial peptides were identified from all the samples together. Identical bacterial peptides in the control cohort were depleted from GBM tumor specific bacterial peptides

analyzed and added the immunopeptidome data from brain lesions of multiple sclerosis (MS) patients and healthy brain tissue ⁴ (dataset identifier PXD019643). We identified 10 and 45 hits from the tissues of 3 MS patients and 6 healthy donors, respectively (**Extended data fig.1a** and **Supplementary Table 4**). 7 peptides from these control cohorts were exact matches of the peptides in GBM tumors, and we therefore then subtracted them from the bacterial peptides in GBM tumors and prepared new graphs (**Fig. 1b**). Using this step, we eliminated bacterial peptides that were not specific for GBM. Although the number of bacterial peptides in the control cohorts were low, the presence of any bacterial peptide in the brain tissues is surprising. Nejman et al. show that healthy tissue adjacent to the tumors also harbor bacteria ⁵. In this case, it is, however, not clear whether the presence of bacteria in healthy-appearing tissue surrounding the tumor is due to the presence of tumor cells or not. Since glioblastoma is known to be tissue invasive, it would be difficult to exclude this point by analyzing tissue surrounding glioblastoma. The observation of small numbers of bacterial peptides on HLA-class II molecules of brain tissue from 6 healthy donors might be due to the presence of APCs binding peptides that had passed the normally tight barrier after the donor's death (the healthy donors' brains were donated shortly after their death and immunopeptidome analyses conducted afterwards) ⁴. Altogether, there were only 7 peptides that were shared between the control cohorts and GBM tumors, and these were subtracted. Furthermore, using 16S rRNA gene sequencing of 10 tumor samples we show that 6 out of 10 GBM

patients display at least 1 exact species match compared with immunopeptidome analysis (**Extended Data Fig. 1b** and **Supplementary Table 7**).

"6. The authors infer from the fact that the immunopeptidome of tumor-derived cell lines also contains bacterial peptidomes that the bacteria must be intracellular. However, this is not shown. It is unclear whether the intracellular presence of these bacteria result in processing and presenting of the identified epitopes."

Authors' reply: We have indeed not stated clearly what we think. HLA-class II-presented peptides can either be loaded exogenously or - probably for the larger part of peptides - derive from peptides that stem from intracellular processing in endosomal compartments. Hence, the immunopeptidome-derived peptides could either stem from peptides that are present in the tumor environment, but outside of the cells, or from inside. The latter is more likely for antigen-presenting cells that are capable of processing exogenous antigens. Tumor cells are unlikely candidates for efficient processing and presentation. Since we cannot be sure whether immunopeptidome-derived peptides stem from HLA-class II molecules on tumor cells or APCs in the tumor, we can only mention the two possibilities. The demonstration of bacterial peptides in the immunopeptidomes of in vitro cultured tumor cells, suggests that these have been exogenously loaded onto class II molecules. There are examples of intracellular bacteria including *Enterococcus faecalis* and *Mycobacterium tuberculosis* in the immunopeptidome data from the isolated cancer cell lines of the patients. However, we do not know if these were degraded and intracellularly loaded in the tumor cells. The existence of bacterial peptides in the immunopeptidome of cancer cells suggests that part of the bacterial peptides derives from tumor cells, but these aspects should be examined in more detail in the future. Since the bacterial load is very low in the GBM tissue and the cancer cell lines, it is, however, very difficult to address. We agree with the reviewer regarding ambiguity of the efficiency of bacterial protein processing and presentation and tried to express this aspect more cautiously in the revised manuscript.

Referee #3 (Remarks to the Author):

"Naghavian and colleagues set out to assess whether CD4+tumor-infiltrating lymphocytes (TILs) recognize intratumorally bacterial pathogens and gut microbiota in glioblastoma. This work is based on the premise that a subset of the intra-tumoral bacteria can invade eukaryotic cells, be processed and presented on HLA-II molecules impacting the immune response. Using an antigen discovery approach, the authors demonstrate cross-reactivity of a TIL derived T cell clone recognizing a broad spectrum of peptides including tumor neoantigen and bacterial peptides, and

further show that these bacterial peptides are able to stimulate bulk TILs and neoantigen-derived T cell clones, showing the functional relevance of the tumor microenvironment. The concepts underlying this manuscript are of high interest to both immunologists and proteomic scientists; however, overall the data presented by the authors do not sufficiently support their conclusions. Further clarifications/additions should be made as described below."

"1. In order to identify bacterial peptides, the authors perform HLA peptidomics on 19 glioblastoma samples and tumor cell lines using bacterial proteomes from UniprotKB/Swiss prot. The use of such big databases for searching the peptides can affect the validity of the findings. Also, as it includes not only the proteomes of the bacteria that were actually identified in the tumor samples, it can result in many false positives. The authors should definitively identify the bacteria residing in each patients' tumor in order to analyze their MS data with it, so they could claim that these are peptides presented by the patient's tumors. Identification of the species in each patient by 16s sequencing will enable a patient-specific analysis, this method is extensively used in microbiome research and fundamentally required here to prove their identifications."

Autors' reply: We thank the reviewer for these important suggestions. We agree that using very large databases might result in false positive hits. Below, we will outline, which measures have been taken to control for this problem. As we explain now in more detail in the Material and Methods section, we have used the non-redundant bacteria UniProtKB/Swiss-Prot protein database (December 12, 2020 with 334,492 entries) and analyzed all the sequences manually. Moreover, all the sequences were then re-evaluated by comparing them with UniProt human proteins, UniProt viral proteins, and bacteriophage protein databases ⁶ as well as genomic transposable elements (TEs) (please also see the response to your 4th question below; data is not included in the manuscript) to ensure that the sequences are derived from no other sources.

Furthermore, 16S rRNA gene sequencing was conducted with fresh frozen tissues of 10 GBM patients, from whom we analyzed the immunopeptidome. These data are already mentioned in the response to reviewer 1, and we would therefore like to refer reviewer 3 to the detailed response above (please also see **Extended data Fig. 1b** and **Supplementary Tables 5 and 6**). Our data suggest that using the 16S rRNA gene sequencing information that had been annotated for bacteria as the only source for subsequent annotation of the immunopeptidome-derived sequences for bacteria limits the number of identified peptides. Using the large available databases for bacterial sequences avoids this limitation.

"2. The authors claim that the identified peptides are derived from tumor cells, but the authors have not ruled out the possibility that APCs in the tumor microenvironment could take up the pathogen antigens and present these peptides on their HLA-II molecules. The authors should

experimentally demonstrate that the infected tumor cells are the only source of these pathogen-derived antigens presented in the tumor microenvironment or alternatively change their claims. This is an important distinction since the antigen presentation pathway between phagocytosing APCs would be different than tumor cells, and therefore there will be differences in the presented antigens. Altering the language would help clarify this, additionally to clarify this possibility, an experimental suggestion would be to isolate the APCs from tumor cells and perform HLA-II peptidomics on both populations, emphasizing the presence of tumor pathogen peptides."

Authors' reply: As already mentioned above in the reply to reviewer #2, we completely agree with the reviewer that APCs in the tumor microenvironment can also be a source for bacterial peptides in the tumor immunopeptidome and that we should therefore be cautious with our conclusions/claims. We have tried to clarify this point now better in the discussion of the revised manuscript. The presence of bacterial peptides in the immunopeptidome of tumor cells indicates that cells in the tumor tissue may be responsible for part, but probably not all of the bacterial peptides. Furthermore, the immunopeptidome analyses of the glioblastoma cell lines lend at least some support that part of the peptides were derived from tumor cells and not APCs. Although our data demonstrate that IPdBPs are weak stimulators of TILs in GBM, the presence of bacterial peptides on glioblastoma-derived cell lines in vitro could be exploited for anti-tumor therapies in the future. We have adapted the text to express this point more clearly. Unfortunately, we did not have enough material for isolation of APCs from tumor and subsequent immunopeptidome analysis.

"3. If the authors suggest that the bacterial HLA-II peptides are presented by the tumor cells, they should provide evidence showing the glioblastoma tumors express HLA-II, as not all tumor cells do."

Authors' reply: Another important point. To address it, we first analyzed the RNA-seq of the tumor tissue (1635WI) for HLA-DR expression. Normalized gene counts of HLA-DRs were compared with The Cancer Genome Atlas (TCGA) GBM cohort (**Extended Data Fig. 2a**). These data suggest that HLA-DR is expressed at normal levels in the tumor tissue (white square represents normal expression of the gene in the tumor) (**Supplementary Table 14**).

Moreover, we had isolated cancer cells from the tumor tissue (1635WI) and expanded these in vitro. 28.6 % of these cells stained positive with the HLA-DR antibody (**Extended Data Fig. 2b**). Furthermore, IHC staining of paraffin-embedded recurrent tumor tissue (1635WI) was performed with antibodies for GFAP, CD31, CD3, CD68 and HLA-DR ⁷. HLA-DR is present on the surface of immune cells (CD3+), endothelial cells (CD31) and GFAP+ astrocytes (**Extended Data Fig. 2c**)(see below).

Extended Data Fig. 2: HLA-II expression of tumor tissue and glioblastoma cell line

a, RNA-seq analysis of the tumor tissue (1635WI) was conducted by TCGA-RNAseqv2 pipeline (https://webshare.bio-inf.unc.edu/public/mRNAseq_TCGA/UNC_mRNAseq_summary.pdf) ⁷. Normalized gene counts of HLA-DRs were compared with The Cancer Genome Atlas (TCGA) GBM cohort. White square represents normal expression of the gene in the tumor (**Supplementary Table 14**).

b, Cancer cells isolated from the tumor tissue (1635WI) and expanded in vitro were stained with HLA-DR antibody.

c, Paraffin-embedded recurrent tumor tissue (1635WI) was stained for GFAP, CD31, CD3, CD68 and HLA-DR⁷. HLA-DR is present on the surface of immune cells (CD3+), endothelial cells (CD31) and GFAP+ astrocytes.

"4. There are clarifications that need to be made and missing information in the materials and methods section which are fundamental to the understanding of the experiments performed. Specifically, the information regarding the growth of the tumor-derived cell lines used for HLA-II peptidomics, as well as the number of replicates. Most likely bacterial species will not survive within tumor cells cultured for a number of passages. Additionally, in the immunopeptidome analyses section, there is missing information regarding the peptide search, was the MS data searched with both human and pathogen proteomes together or only with the pathogen database? What was the FDR? And how did the authors control for the FDR analysis when using such a big dataset? Removing redundant sequences from the pathogen database can help with this. Were there any filtrations steps of the identified peptides or quality controls for the data, such as the removal of peptides which sequence is identical between the pathogen and human peptides, ambiguous I/L identifications, peptides that might result from non-coding regions and pseudogenes? Such steps will ensure that these peptides are indeed derived from pathogens and not the result of other human sources or ambiguity. Additionally, the list of pathogen species used for the HLA peptidomics analysis is missing."

Authors' reply: All immunopeptidome analyses have been conducted in triplicates. The MS/MS data were used to search in both human and non-redundant eubacteria UniProtKB/Swiss-Prot protein databases (December 12, 2020 with 334,492 entries), and all spectra were manually analyzed. We only considered a spectrum valid if we clearly identified a sequence of 4 B- or Y-ions. Further, bacterial peptides were re-evaluated by systematically searching them in UniProt human, UniProt bacteria, UniProt virus databases as mentioned before, and furthermore also bacteriophage⁶ and genomic transposable elements (gTEs). Any matches with databases besides bacterial proteins/peptides would have been removed from the list, but our analysis did not show identical matches of bacterial peptides with any of the above databases, but 18 peptides from the human proteome. We have kept these 18 peptides with identical sequences in human and bacteria, and they are now specified in the revised **supplementary table 3**. Interestingly, these peptides are derived from proteins that are identical in bacteria and human and highly conserved in phylogeny like HSP70 or energy metabolism-related proteins. Regarding isoleucine/leucine (I/L) ambiguity, we considered replacing all isoleucines and leucines interchangeably. This step affected only 23 out of 344 bacterial peptides, and all other peptides were only present in bacterial databases (now specified in **supplementary table 3**). Following the comment of the reviewer, we also searched for matches of the bacterial sequences in the database for nucleic acid sequences

of 4.5 million gTEs and also approximately 34,000 amino acid sequences from intact endogenous viral elements (gEVE database)⁸. These data are shown below (**Figure. R5**) but are not included in the manuscript.

Analysis was performed by processing data at different thresholds of coverage of the query (bacterial peptides), and blast hits are filtered based on 3 parameters:

- No mismatch allowed between query and subject,
- No gap allowed between query and subject,
- 100% identity between query and subject.

After applying these filters, we created different categories ranging from 60% of the peptide to the full sequence of the peptide (100%) that map to the subject with 100% identity. For example, for a peptide of 14 amino acids:

- 60% -> 8.4 -> 8 / 14 Amino acids from the peptide match the subject,
- 70% -> 9.8 -> 9 / 14
- 80% -> 11.2 -> 11 / 14
- 90% -> 12.6 -> 12 / 14
- 100% -> 14 -> 14 / 14 (Perfect match)

Only 1 unique bacterial peptide from cancer cell line (ZH681) was found with 100% match to a translated TE sequence and no sequence matched the gEVE.

Figure R5. Lists of IPdBP were analyzed for the presence of peptides from the non-coding regions. Left figure shows the comparison of IPdBP with translated gTEs. Number of peptide matches are shown above each column and the x axis display the percentage of peptide match (100% means identical sequences). Right figure represents the comparison of IPdBP with gEVes. Similarly, number of peptide matches are shown above each column and the x axis display the percentage of peptide match (100% means identical sequences).

Moreover, beside analyzing samples with MASCOT software, they were also analyzed using Scaffold (Scaffold version 5.2, Proteome Software), and the "Protein prophet"-reported false discovery rates (FDR) was between 2.4 and 5.4. Isolated cancer cell lines were passaged up to 30 times before they were snap frozen and used for the immunopeptidome analyses. The high number of passages assures the absence of contaminating cells. We have now explained this better in the M&M section.

"5. The authors should validate the peptides identified by HLA peptidomics by comparing their MS/MS spectra fragmentation to that of synthetic peptides, used as standard validation for peptide identification. Additionally, the authors present in Supplementary Table 5 the MS spectra only from patient 1635WI, the bacterial peptide spectra from the other patients are missing. All RAW data as the authors mentioned in the material and method section, as well as proteome databases and search results should be made publicly available by depositing them in public repositories."

Authors' reply: As suggested by the reviewer, we synthesized the bacteria/microbiota-derived IPdBPs of patient 1635WI and measured them with the same tandem mass spectrometry method and device (Orbitrap Fusion Lumos, Thermo Fisher Scientific) and their MS/MS spectra were compared with those of the immunopeptidome peptides. Since our synthetic peptides were unfortunately synthesized with modifications at the N- (acetylation) and C-termini (amidation), spectra often look different in terms of intensities and presence/absence of B -and Y- ion series. The end modification changed the charge of the peptides and sometimes the character of ionization patterns. The reason for synthesizing peptides with the above end modifications stems from a convention that we previously established based on the ps-SCLs search algorithm. The latter decamer peptide mixtures, which are used for the unbiased antigen discovery studies, have N-terminal acetylation and C-terminal amidation, and consequently, we always synthesize the predicted peptides with the same ends. The peptides for the sequence validation of the IPdBPs should have been synthesized with free ends. Despite this fault and limitation, peptide sequencing identified the exact same amino acid sequences for 37 out of original 40 IPdBPs via the synthetic peptides. The comparison of MS/MS spectra of 10 synthetic peptides / original spectra of IPdBPs are now shown in **supplementary table 16**.

If the reviewer feels that the above does not address the concerns sufficiently we will re-synthesize the peptides without end modifications and analyze these as well.

The MS spectra from patient 1635WI serve as an example of how the peptide spectra look like. The raw data and the analysis of the immunopeptidome from all GBM tumor tissues and control

materials are now available on ProteomeXchange Consortium via the PRIDE partner repository with the dataset identifier PXD036811.

Reviewer account details:

Username: reviewer_pxd036811@ebi.ac.uk

Password: **N2qC920g**

As mentioned before, the proteome databases are derived from the UniProtKB/Swiss-Prot protein database. The human, bacteria and viral protein databases are from UniProt and publicly available at <https://www.uniprot.org/>. Gut bacteria⁹ and bacteriophage⁶ databases are from the cited papers. The dataset for the non-coding sequences are also available in the cited paper⁸.

"6. Do the authors see these bacterial antigens in any previously published HLA-peptide datasets?"

Authors' reply: As previously mentioned, we show 18 peptides out of the total 344 IPdBPs that can be annotated to both human and bacteria in the revised **supplementary table 3**, 'identical sequences' tab. 13 of these peptides are present in the HLA ligand atlas (<https://hla-ligand-atlas.org/>) and were previously annotated to human proteins. The remaining 331 peptides are not present in HLA-peptide datasets (**Table R1**). We also compared the IPdBPs with HLA-derived bacterial peptides in melanoma¹ and found one shared bacterial peptide, *FRVPTANV*.

"7. The authors should remove control bacterial species, bacterial species that were shown to be found in normal tissue, thus would not expect to be found in the tumor."

Authors' reply: This point was also mentioned under point 5 of Referee #2. In order to control for this aspect, we have analyzed and added the immunopeptidome data from brain lesions of multiple sclerosis (MS) patients and healthy brain tissues⁴ (dataset identifier PXD019643). We explained the inclusion of these new data and how it was incorporated into the revised manuscript, and therefore would like to refer the reviewer to our response to point 5 of Referee #2 above. Bacterial species that were also found in control tissues have been removed.

"8. For recurrent peptides between primary and recurrent tumor in patient 1635WI, authors should mark these peptides in supplementary table 4, additionally they should describe if their origin is from an identical bacterial species, and if there were recurrent peptides that originated from a number of different species."

Authors' reply: We have now specified this in **supplementary table 8** and in the text. We found 4 identical peptides in the primary and recurrent tumors of patient 1635WI, and these are marked

now in yellow in the supplementary table 8. Moreover, when comparing bacterial peptides from all tumor tissues, there are several peptides that we found in the tumor tissues of several patients. This information is now also provided in the revised **supplementary table 3**, 'Grouped by peptide sequence' tab.

"9. In Extended data Figure 1, did the authors test the TCC88 proliferation after incubation with the SIN3A unmutated peptide? this should be added to the figure as a comparison."

Authors' reply: We have now added the response of TCC88 to the unmutated SIN3A peptide in the revised **Extended data Fig. 4f**. Based on our criteria ($SI \geq 2$), TCC88 barely recognizes the unmutated SIN3A at 10 μM ($SI=2$). We have also compared the dose responses of TCC88 to the mutated and unmutated SIN3A peptides. While TCC88 strongly responds to the mutated SIN3A* peptide, the response to the unmutated SIN3A is already negative at 1 μM (see below).

Extended Data Fig. 4f, Response of TCC88 to unmutated SIN3A peptide was compared to SIN3A* at different concentrations including 10, 1, 0.1, 0.01 and 0.001 μM .

"10. In lines 224-233 the authors describe the application of an antigen discovery methodology for targeted peptide recognition, have the peptides chosen by the authors to be tested in the proliferation assay of TCC88, been identified as binders to the 1635WI alleles using prediction servers such as NetMHCIIpan? Are they strong binders? Could the authors provide the readers with a comparison between the two?"

Authors' reply: Beside the initial IPdBPs, all other peptides that were chosen to test with TCC88 are based on our unbiased antigen discovery approach using positional scanning combinatorial peptide libraries in combination with a dedicated bioinformatics approach¹⁰. As mentioned in the manuscript, we used APCs (BLS transfectants) expressing only HLA-DRB3*02:02 from patient 1635WI, because this was the main restriction element for TCC88. Our scoring matrix, on which

the bioinformatics algorithm for the search of targets is based ¹⁰, was employed to identify peptides based on the recognition of TCC88 and the 200 positional scanning decamer libraries. The antigen identification methodology is based on our prior observations that each amino acid between TCR and HLA-DR contributes additively and independently to antigen recognition ¹⁰. This approach does not build on or include HLA binding predictions, and among the peptides that are identified for a given T cell clone are both well- and poorly binding peptides. All peptides with high predicted stimulatory scores from bacteria/gut microbiota, autologous tumor RNA-seq data that had been translated into protein sequences, and viruses are then synthesized. As stated above, peptide-binding predictions are not part of the identification algorithm.

To follow the reviewer's suggestion, we now predicted the binding affinity of peptides to all HLA-DR alleles of patient 1635WI using NetMHCIIpan-version 4 (**Table R2**; these data are not included in the manuscript). As can be seen in **Table R2**, very few peptides, and these came only from the immunopeptidome-derived peptides (annotated to bacterial sources) showed predicted weak (rank between 1 and 5%) or strong binding (rank 1% and below) to one of the HLA-DR alleles of the patient. From this reason, we have not included the data in the manuscript, but can do so, if the reviewer thinks that the information would be relevant for the readers.

"11. In lines 256-262 the authors state that 57.7% of all stimulatory peptides, predicted with their stimulatory scores and tested with clone TCC88 were derived from bacteria and gut microbiota. However, most of the peptides tested by the authors were derived from these types making it hard to rule out autoreactivity, specifically when only 4 peptides from the human proteome were tested—a very unbalanced comparison. It is surprising that this clone tested with the predicted bacterial peptides did not respond to the immunopeptidome-derived bacterial peptides (Fig. 2d), how do the authors explain this?"

Authors' reply: It appears that we did not address these very important points well throughout the manuscript. As shown in **figure 3a** and **supplementary table 10**, 18 peptides from the human proteome were chosen to test TCC88. The reason for testing 18 peptides was that only these peptides met our search criteria for choosing peptides, i.e. that they had at least 80% of the maximal theoretical score based on testing TCC88 with the peptide libraries and predicting targets for the clone. Of these peptides, 4 were able to stimulate the TCC ($SI \geq 2$). Furthermore, we tested the clone with 135 peptides from known autoantigens (MBP, MOG, PLP, RASGRP2 and TSTA3) (**Extended data fig. 4e** and **Supplementary Table 11**). Based on our prior experience with TCCs derived from patients with a bacterial infection (chronic nervous system Lyme borreliosis) ¹¹, influenza infection ¹², and multiple clones from multiple sclerosis, an autoimmune disease of the

central nervous system^{13–15}, autoantigens, i.e. antigens from unmutated self-proteins are usually recognized less well than foreign antigen-derived peptides, which is in agreement with the elimination of autoreactive T cells with high functional avidity by thymic selection.

Furthermore, we did not state clearly that we do not assume that the bacterial / microbiota-derived target antigens of a TCC like TCC88 have to be present in the tumor, but rather the tumor antigen, which the clone cross-recognizes. We were therefore not surprised that TCC88 did not recognize immunopeptidome-derived bacteria / microbiota-derived peptides well despite its broad reactivity against multiple other peptides from bacteria and microbiota. The peptide sequences that we identified for TCC88 using the unbiased target discovery approach are completely different from the bacterial peptides found via immunopeptidomics (**Supplementary Table 15**). We have tried to clarify these points better throughout the manuscript.

"12. In extended data fig. 1e, the authors test 135 peptides from brain and myelin autoantigens. In order to rule out auto reactivity, specifically, when suggesting that such clones could be potentially used for adoptive cell therapy a wider and a more in-depth study should be performed. The test of HLA presented peptides from non malignant tissues, should be used, for example, those that appear in the HLA ligand atlas Marcu et al. 2020."

Authors' reply: We agree with the reviewer that this is an important point if one were to consider applying the strategy for target antigen discovery of bacteria/microbiota-derived antigens and cross-recognized self-/tumor antigens for tumor therapy. Unwanted cross-reactivity with critical targets that might result in organ toxicity should be ruled out or at least be known. We address this aspect below by the following steps and arguments:

Beside our initial testing of TCC88 with 135 brain autoantigens, we decided to test TCC88 with peptide pools covering the full length of autoantigens from skin and pancreas (**Extended data fig 4e** and **Supplementary Table 11**). We tested TCC88's response to 125 peptides from skin and pancreas autoantigens including MELAN-A, Tyrosinase (TYR), Premelanosome protein (PMEL), Insulin, SLC30A8 (ZnT8) and islet amyloid polypeptide (IAPP). TCC88 did not respond to any of these peptides either (**see below Extended data fig. 4e**).

Moreover, since our antigen discovery algorithm does not build on HLA binding predictions, we used the scoring matrix of TCC88 to score all HLA-II-derived peptides from the tumor tissue which are annotated to normal human proteins. As we outlined in the manuscript and demonstrated in several prior papers, peptides with high predicted stimulatory scores are also very likely to be recognized by the clone^{10,13–15}. Such peptides did not have high predicted scores and were indeed not recognized by TCC88 (**Figure. 3a, Figure. R6**).

We synthesized and tested the top 80 unique peptides with highest scores from primary and recurrent tumors of the patient tested with TCC88 (**Table R3**). TCC88 did not respond to any of

these peptides. As can be seen in **Table R3**, only the top 6 peptides had predicted stimulatory scores > 80% of the maximal score. We furthermore added data about the tissue distribution of the respective proteins, which we obtained from the Human Protein Atlas. Almost each of the source proteins of these peptides are broadly expressed in the human body, i.e. in multiple organs, however, they were not stimulatory for TCC88.

Furthermore, we have previously described TCCs responsive to influenza hemagglutinin¹² after acute flu infection, to *Borrelia burgdorferi* (a clone from the CSF of a patient with chronic CNS Lyme disease¹¹), and T cell clones isolated from the CSF of a patient with multiple sclerosis in relapse¹⁶. In each of these cases, the respective T cell clones not only recognized infectious agents (Influenza A virus, *Borrelia burgdorferi*, Torque teno virus), but also multiple self-proteins and many of these not only from the CNS, but also with broad organ expression. Hence, the cross-reactivity with self-proteins is not rare and does not necessarily imply a high risk of harmful off-target effects.

The induction/unleashing of antibody- and/or T cell reactivity against self-tissues are frequent adverse effects of checkpoint inhibitor therapies¹⁷ and they are an accepted risk vis-a-vis the prognosis of the tumors that are being treated with these. The reviewer is right, however, that such risks in principle also hold for peptide vaccination therapies if they were to elicit not only anti-tumor responses, but also reactivity to healthy tissue. In the future, algorithms should be developed that de-risk such possible adverse events as much as possible, e.g. focus on cancer testis antigens or on antigens like Melan-A, which may cause the well-known side effect of treatment-induced vitiligo in melanoma vaccination, but are not life-threatening.

From our data with the glioblastoma TIL-derived TCC88, we consider this risk not very high, but much more data with more clones and from phase I clinical trials will be necessary to understand this aspect better and develop safeguards against it.

Extended Data Fig. 4e TCC88 was tested with peptide pools (including 260 peptides) covering full length brain, pancreas and skin autoantigens (Supplementary Table 11) autoantigens including myelin basic protein (MBP pools), myelin oligodendrocyte glycoprotein (MOG pools), myelin proteolipid protein (PLP pool contains 5 immunodominant PLP peptides), RAS guanyl-releasing protein 2 (RASGRP2 pools), GDP-L-fucose synthase (TSTA3 pools), MELAN-A, Tyrosinase (TYR), Premelanosome protein (PMEL), Insulin, SLC30A8 (ZnT8) and islet amyloid polypeptide (IAPP). Peptide pools were pulsed on irradiated BLS-DRB3*02:02 cells and co-cultured with TCC88 for 3 days. Proliferation was measured using ³H-thymidine incorporation (data represents mean value of 3 wells ±SEM).

Figure R6. Negative responses of TCC88 to HLA-II-derived peptides from primary and recurrent tumors that are annotated with normal human proteins, and which have the highest predicted stimulatory scores as calculated via the scoring matrix of TCC88. Proliferation was measured using ³H-thymidine incorporation after 3 days (data represents mean value of 2 wells ±SEM)(PEPT1.... = Peptidome-derived peptides of primary tumor; PERT1.... = Peptidome-derived peptides of recurrent tumor).

"13. In Figure 3d-f, the authors were not consistent with the peptides chosen for activation assays of TCC88 and have not provided an explanation in the text, why was only HGM3 chosen for the killing assay in Fig. 3f and this was not confirmed with the additional DRB3 microbiota peptides HGM62 and HGM27 in which there functional avidity is similar in figure 3d?"

Author's reply: Due to the slow in vitro growth of the tumor cell lines, we were limited with respect to testing a larger number of peptides. HGM3 was chosen since it has the same EC50 as the SIN3A* peptide for the clone. We have explained this rationale now in the revised manuscript.

"14. In line 300 it is not clear why TCC68 and TCC75 were chosen for bacterial peptide reactivity? Based on Fig. 1b there were a lot of other TCC that showed high SIN3A* antigen activation

additionally it raises the question why were these TCC not reacting to the HLA-II presented bacterial peptides in figure 2d."

Authors' reply: This is correct, and initially there were multiple TCCs that responded strongly to SIN3A* peptide, however, many could not be expanded sufficiently well. We could maintain several TCCs, which still show good reactivity to SIN3A* peptide and continued further experiments with these. Also, we tested several TCCs other than TCC68 and TCC75 for their reactivity to bacteria/microbiota-derived peptides, but, similar to TCC8F7, they did not respond to these peptides (**Figure R7**, data not shown in the manuscript).

Figure R7. Negative responses of several TCCs to bacteria/microbiota-derived peptides. Proliferation was measured using ³H-thymidine incorporation after 3 days (data represents mean value of 2 wells ±SEM).

"15. Changing the presentation of the TCC proliferation graphs into a heatmap with an intensity-colored scale will clarify figure 4, and combining Fig. 4b and c, would make this figure clearer."

Authors' reply: This probably did not become sufficiently clear from the figure legend. **Fig. 4c** indeed summarizes the responses of TILs and TCCs to bacteria/microbiota-derived peptides (combination of fig. 4a, 4b and fig3b) in an intensity-colored scale.

"Additional comments:

- The excel supplemental table numbers are missing, making it hard to find supplementary tables. There is no table of contents in these files or legend.
- A comprehensive comparison of the bacteria phylum previously associated with response to immunotherapy treatment, such as: Gopalakrishnan., 2018, Matson., 2018, Routy, 2018, should be added and should also be discussed In the discussion section.
- What are the limitations to this study? Can the authors add to discussion?"

Authors' reply: We apologize for these omissions and have now added a table of contents. We agree with the reviewer and therefore compared bacterial species that were found using ps-SCL and were stimulatory for TILs and TCCs (**Supplementary Table 15**) with the abovementioned studies and added this point also to the discussion. Regarding the limitations of the study, we also agree with the reviewer that they should be mentioned. We have added the following paragraph on the most important limitations of the work in the discussion.

Limitations of our study include that we could not examine the "autologous" gut microbiota of patients, their lung microbiome, which appears to play an important role in brain homing of proinflammatory T cells¹⁸, and also that the most detailed studies on both tumor immunopeptidome and its annotation to bacteria/microbiota, unbiased target discovery for TIL-derived TCC and their cross-reactivity against bacteria and tumor antigens mainly derive from one patient. However, we identified TIL reactivity against some of the bacteria/microbiota-derived peptide targets from patient 1635WI already in TILs from another glioblastoma patient (data not shown). Furthermore, the three different methodologies, i.e. 16S rRNA sequencing, immunopeptidomics and unbiased target discovery for a TIL-derived T cell clone, identify largely non-overlapping bacterial and gut microbiota-derived species and peptides. All suggest an involvement of bacteria and gut

microbiota in the tumor's biology and/or immune defense against it, however, future studies will need to examine their roles in more depth.

References:

1. Kalaora, S. *et al.* Identification of bacteria-derived HLA-bound peptides in melanoma. *Nature* **592**, 138–143 (2021).
2. Galeano Niño, J. L. *et al.* Effect of the intratumoral microbiota on spatial and cellular heterogeneity in cancer. *Nature* **611**, 810–817 (2022).

3. Fu, A. *et al.* Tumor-resident intracellular microbiota promotes metastatic colonization in breast cancer. *Cell* **185**, 1356-1372.e26 (2022).
4. Marcu, A. *et al.* HLA Ligand Atlas: a benign reference of HLA-presented peptides to improve T-cell-based cancer immunotherapy. *J Immunother Cancer* **9**, e002071 (2021).
5. Nejman, D. *et al.* The human tumor microbiome is composed of tumor type-specific intracellular bacteria. *Science* **368**, 973–980 (2020).
6. Camarillo-Guerrero, L. F., Almeida, A., Rangel-Pineros, G., Finn, R. D. & Lawley, T. D. Massive expansion of human gut bacteriophage diversity. *Cell* **184**, 1098-1109.e9 (2021).
7. Wang, J. *et al.* Vaccination with designed neopeptides induces intratumoral, cross-reactive CD4⁺ T cell responses in glioblastoma. *Clin Cancer Res* CCR-22-1741 (2022) doi:10.1158/1078-0432.CCR-22-1741.
8. Bonté, P.-E. *et al.* Single-cell RNA-seq-based proteogenomics identifies glioblastoma-specific transposable elements encoding HLA-I-presented peptides. *Cell Rep* **39**, 110916 (2022).
9. Almeida, A. *et al.* A unified catalog of 204,938 reference genomes from the human gut microbiome. *Nat Biotechnol* **39**, 105–114 (2021).
10. Zhao, Y. *et al.* Combinatorial Peptide Libraries and Biometric Score Matrices Permit the Quantitative Analysis of Specific and Degenerate Interactions Between Clonotypic TCR and MHC Peptide Ligands¹. *The Journal of Immunology* **167**, 2130–2141 (2001).
11. Hemmer, B. *et al.* Identification of candidate T-cell epitopes and molecular mimics in chronic Lyme disease. *Nat Med* **5**, 1375–1382 (1999).
12. Markovic-Plese, S. *et al.* High level of cross-reactivity in influenza virus hemagglutinin-specific CD4⁺ T-cell response: implications for the initiation of autoimmune response in multiple sclerosis. *J Neuroimmunol* **169**, 31–38 (2005).
13. Planas, R. *et al.* GDP-I-fucose synthase is a CD4⁺ T cell-specific autoantigen in DRB3*02:02 patients with multiple sclerosis. *Sci Transl Med* **10**, eaat4301 (2018).

14. Jelcic, I. *et al.* Memory B Cells Activate Brain-Homing, Autoreactive CD4+ T Cells in Multiple Sclerosis. *Cell* **175**, 85-100.e23 (2018).
15. Wang, J. *et al.* HLA-DR15 Molecules Jointly Shape an Autoreactive T Cell Repertoire in Multiple Sclerosis. *Cell* **183**, 1264-1281.e20 (2020).
16. Sospedra, M. *et al.* Recognition of Conserved Amino Acid Motifs of Common Viruses and Its Role in Autoimmunity. *PLOS Pathogens* **1**, e41 (2005).
17. Perdigoto, A. L., Kluger, H. & Herold, K. C. Adverse events induced by immune checkpoint inhibitors. *Curr Opin Immunol* **69**, 29–38 (2021).
18. Hosang, L. *et al.* The lung microbiome regulates brain autoimmunity. *Nature* **603**, 138–144 (2022).

Reviewer Reports on the First Revision:

Referees' comments:

Referee #1 (Remarks to the Author):

The authors did their best to clarify the remaining issues and offer a new vision on how to best guide peptide-based or T cell based immunotherapy of this challenging cancer type.

One sentence has been added but I believe that the data need to be shown because they efficiently feed their viewpoint. This is this one:

"However, we identified TIL reactivity against some of the bacteria/microbiota-derived peptide targets from patient 1635WI already in TILs from another glioblastoma patient (data not shown)".

Referee #2 (Remarks to the Author):

The authors sufficiently addressed my concerns.

Referee #3 (Remarks to the Author):

The authors perform an in-depth analysis of glioblastoma infiltrating CD4+ T cells identifying a T cell clone that recognizes a broad spectrum of peptides from pathogenic commensal gut microbiota and tumor antigens. The authors utilize an unbiased antigen discovery approach, HLA peptidome analysis, 16s rRNA sequencing and T cell response characterization helping the authors in supporting their claim regarding the specificity of this T cell response. I believe that the authors addressed the majority of my concerns with edits presented in the response to the reviewers' document.

Specifically, I appreciated the addition of experiments and discussion of 16S rRNA gene sequencing from the glioblastoma tumors and the addition of multiple sclerosis patients and healthy brain tissue immunopeptidome analysis.

Below are some additional comments that should be considered by the authors before final publication. I support this manuscript being published after minor edits, as I think this work is both novel and the experiments are thoughtfully executed.

Additional comments for consideration:

1. As the authors have not ruled out the possibility that the presented bacterial peptide origin is from APCs present in the tumor, an indication of which bacterial species identified by the 16s rRNA sequencing are known to be intracellular based in the literature could facilitate in the understanding of the origin of the bacterial peptides.

2. As the authors now include synthetic peptide analysis, for clarity is it possible to use a comparison plot of the synthetic vs. observed spectra? this may be a more digestible presentation of these data.

There are open source R packages to do this: OrgMassSpecR that have specific functions (SpectrumSimilarity()) that can be used to do this, additionally could the authors include the correlation score?

3. Additionally I would suggest a re-evaluation of potential bacterial contaminants from sample processing and their removal from the analysis, such as Propionibacterium acnes, in Supplementary Table 7. An previous in-depth contamination filtering has been reported in the literature.¹

References:

1. Nejman, D. et al. The human tumor microbiome is composed of tumor type-specific intracellular bacteria. Science 368, 973-980, doi:10.1126/science.aay9189 (2020).

Author Rebuttals to First Revision:

Referees' comments:

Referee #1 (Remarks to the Author):

The authors did their best to clarify the remaining issues and offer a new vision on how to best guide peptide-based or T cell based immunotherapy of this challenging cancer type.

One sentence has been added but I believe that the data need to be shown because they efficiently feed their viewpoint. This is this one:

"However, we identified TIL reactivity against some of the bacteria/microbiota-derived peptide targets from patient 1635WI already in TILs from another glioblastoma patient (data not shown)".

Authors' reply: We thank the reviewer for the valuable input during the revision. This data was previously provided in response to the 2nd reviewer's question number 1, which we did not include in the manuscript. We have now included the IFN- γ fluorospot assay of TILs from another glioblastoma patient (GBM-40) in the manuscript (**Extended Data Fig. 7**). Briefly, 2×10^4 TILs/well were seeded with 5×10^4 irradiated (300 Gy) autologous EBV-transformed B cell line cells (BLCLs) as antigen presenting cells primed with bacteria/microbiota-derived peptides (10 μ M final concentration). Δ SFU (spot forming units) for each peptide was calculated after subtraction of the mean number of spots in negative control. As

shown, several bacteria/microbiota-derived peptides were able to stimulate TILs from another glioblastoma patient (GBM-40) after 44 hours of incubation, albeit less strongly than in patient 1635WI.

Extended Data Fig. 7: IFN- γ fluorospot testing of expanded TILs from patient GBM-40

a, 64 bacteria/microbiota-derived peptides that were stimulatory for TILs from patient 1635WI were tested with TILs from patient GBM-40. TILs were seeded with irradiated autologous BLCLs primed with bacteria/microbiota-derived peptides in triplicates. Δ SFU (spot forming units) values for each peptide were calculated after subtraction of the mean number of spots in negative control. **b**, Positive and negative responses of TILs to several bacteria/microbiota-derived peptides are shown on the lower panel. CD2-CD3-CD28 beads are used as positive control.

Referee #2 (Remarks to the Author):

The authors sufficiently addressed my concerns.

Referee #3 (Remarks to the Author):

The authors perform an in-depth analysis of glioblastoma infiltrating CD4+ T cells identifying a T cell clone that recognizes a broad spectrum of peptides from pathogenic commensal gut microbiota and tumor antigens. The authors utilize an unbiased antigen discovery approach, HLA peptidome analysis, 16s rRNA sequencing and T cell response characterization helping the authors in supporting their claim regarding the specificity of this T cell response. I believe that the authors addressed the majority of my concerns with edits presented in the response to the reviewers' document. Specifically, I appreciated the addition of experiments and discussion of 16S rRNA gene sequencing from the glioblastoma tumors and the addition of multiple sclerosis patients and healthy brain tissue immunopeptidome analysis. Below are some additional comments that should be considered by the authors before final publication. I support this manuscript being published after minor edits, as I think this work is both novel and the experiments are thoughtfully executed.

Additional comments for consideration:

1. As the authors have not ruled out the possibility that the presented bacterial peptide origin is from APCs present in the tumor, an indication of which bacterial species identified by the 16s rRNA sequencing are known to be intracellular based in the literature could facilitate in the understanding of the origin of the bacterial peptides.

Authors' reply: We thank the reviewer for the very positive comments. Based on the suggestion we have searched the literature and also consulted with the infectious disease / microbiology expert on our team (S. Brugger) to specify for each bacterial species that had been identified by the 16s rRNA sequencing whether it is considered an intracellular or extracellular bacterium or can exist in both niches/environments - if this is known. According to him, there is a general change in dogma that classically extracellular bacteria such as *S. aureus* have now also been considered to have an intracellular lifestyle¹. To include this information in the manuscript, we have now added extensive information on each bacteria from the 16s rRNA data (**Supplementary Tables 5 and 6**). Whenever it is not known, which is the case for many of the bacteria, we have marked it as not available/not known. This being considered, we were able to identify bacteria that have been described to also having an intracellular

lifestyle including *Staphylococcus spp.* and *Pseudomonas spp.* If bacteria can live intracellularly, this point may indicate that they are more likely to derive from the tumor cells, however, it would not exclude antigen-presenting cells (APCs) as sources. The issue is complicated further by data showing that astrocytes, i.e. the cells glioblastoma arises from, are able to express MHC/HLA-class I and -class II, costimulatory molecules and proteases involved in antigen presentation ²⁻⁴. It was further documented that mouse and rat astrocytes in an inflammatory environment and after exposure to IFN- γ can present antigens to T cells ^{2,5}. Human astrocytes have been shown to play a role as APCs in the brains of Parkinson's disease patients ⁶, and even malignant glioma cells can present antigen to T cells, although their capacity to process complex antigens appears to be relatively poor when compared to professional APCs ⁷. To understand this point better in the future, it will be important to process fresh tumor tissue in a way that enriches tumor cells and APCs / immune cells separately and performs the 16S rRNA amplicon sequencing from both populations of cells. Further, a cell-free preparation of extracellular matrix of the tumor under both aerobic and anaerobic conditions will shed further light on which bacterial species exist in a tumor and its different areas, e.g. necrotic tissue, which is prominent in glioblastoma, versus vascularized areas.

This complex and important issue should be investigated in much more detail in the future to clarify what type of niche glioblastoma and its various tissue areas as well as the infiltrating monocytes/macrophages constitutes for extra- and intracellular bacteria. Our reading on the question if and to what extent glioma cells and astrocytes can process and present antigens in an inflammatory / tumor environment quickly revealed that both types of cells may be involved in the bacteria/ gut microbiota in the tumor. To discuss this point appropriately, one would need to add at least a couple of sentences, and, since the manuscript is already quite long, we prefer to avoid it.

2. As the authors now include synthetic peptide analysis, for clarity is it possible to use a comparison plot of the synthetic vs. observed spectra? this may be a more digestible presentation of these data. There are open source R packages to do this: OrgMassSpecR that have specific functions (Spectrum-Similarity()) that can be used to do this, additionally could the authors include the correlation score?

Authors' reply: We agree that mirror plotting the synthetic versus immunopeptidome peptides makes it clearer and also more easily readable. We have now used Universal Spectrum Explorer (USE) to plot them as a mirror display (Supplementary Table 16). However, due to the end-modifications of the synthetic peptides, which we explained in our previous response, and due to the different complexities of spectra (synthetic versus immunopeptidome), plotting the correlation score using the "OrgMassSpecR", as was suggested by the reviewer, is probably confusing unless all the aspects are laid out in detail in the manuscript. The calculated correlation scores for the peptides range from low to high scores. We therefore provide two examples of high and low correlation scores for the reviewer but would prefer not to include these data in the manuscript (but can of course include it). Please, note, that as we previously

mentioned, spectra of synthetic and immunopeptidome are matching, i.e. the sequences are the same. Left panel (**Figure. R1a, b**) displays the mirror plot created using USE and the right panel (**Figure. R1a, b**) displays the mirror plot and the correlation scores generated via “OrgMassSpecR”.

Figure R1. Mirror plots of 2 synthetic peptides versus IPdBPs with high (a) and low correlation scores (b). a, Left panel displays the mirror plot created via the USE software comparing the IPdBPs (top) vs synthetic peptide (bottom) and the right panel displays similar comparison conducted via the “OrgMassSpecR” R package. The correlation score for this peptide is 0.97. b, Similarly, the mirror plot of another peptide using the software and R package are displayed while the calculated correlation score for this peptide is 0.19.

3. Additionally I would suggest a re-evaluation of potential bacterial contaminants from sample processing and their removal from the analysis, such as *Propionibacterium acnes*, in Supplementary Table 7. An previous in-depth contamination filtering has been reported in the literature. 1

Authors’ reply: We agree that this step will reduce the likelihood that contaminating bacteria are considered. In order to further limit the potential bacterial contaminations in our data, we compared 16s rRNA data of the tumors with data provided in Nejman et al., and found several potential contaminants. These bacterial species including *Pelomonas aquatica*, *Enhydrobacter aerosaccus*, *Acinetobacter*

towneri, *Propionibacterium acnes*, *Staphylococcus epidermidis* and *Acinetobacter calcoaceticus* were then removed from the 16s rRNA data (Supplementary Tables 5, 6) as well as Supplementary Table 7. We have adapted the text and methods section accordingly.

References:

1. Belon, C. & Blanc-Potard, A.-B. Intramacrophage Survival for Extracellular Bacterial Pathogens: MgtC As a Key Adaptive Factor. *Frontiers in Cellular and Infection Microbiology* **6**, (2016).
2. Dong, Y. & Benveniste, E. N. Immune function of astrocytes. *Glia* **36**, 180–190 (2001).
3. Li, J. *et al.* Conservation and divergence of vulnerability and responses to stressors between human and mouse astrocytes. *Nat Commun* **12**, 3958 (2021).
4. Fontana, A., Fierz, W. & Wekerle, H. Astrocytes present myelin basic protein to encephalitogenic T-cell lines. *Nature* **307**, 273–276 (1984).
5. Williams, K. C. *et al.* Antigen presentation by human fetal astrocytes with the cooperative effect of microglia or the microglial-derived cytokine IL-1. *J Neurosci* **15**, 1869–1878 (1995).
6. Rostami, J. *et al.* Astrocytes have the capacity to act as antigen-presenting cells in the Parkinson's disease brain. *Journal of Neuroinflammation* **17**, 119 (2020).
7. Soos, J. M. *et al.* Malignant glioma cells use MHC class II transactivator (CIITA) promoters III and IV to direct IFN-gamma-inducible CIITA expression and can function as nonprofessional antigen presenting cells in endocytic processing and CD4(+) T-cell activation. *Glia* **36**, 391–405 (2001).

Reviewer Reports on the Second Revision:

Referees' comments:

Referee #1 (Remarks to the Author):

No more comments. Congratulations for this seminal demonstration.

Referee #3 (Remarks to the Author):

Thank you. All our concerns have been addressed. We have no more critiques.